# R-loop-dependent promoter-proximal termination ensures genome stability

Congling Xu[1,11], Chengyu Li[2,11], Jiwei Chen[1,11], Yan Xiong[1,11], Zhibin Qiao[1,3], Pengyu Fan[1,4], Conghui Li[5], Shuangyu Ma[6], Jin Liu[1], Aixia Song[1], Bolin Tao[1], Tao Xu[7], Wei Xu[8], Yayun Chi[9], Jingyan Xue[9], Pu Wang[10], Dan Ye[10], Hongzhou Gu[1], Peng Zhang[4], Qiong Wang[6], Ruijing Xiao[5], Jingdong Cheng[1], Hai Zheng[1], Xiaoli Yu[3], Zhen Zhang[3], Jiong Wu[9], Kaiwei Liang[5], Yan-Jun Liu[1], Huasong Lu[2✉] & Fei Xavier Chen[1,3✉]

The proper regulation of transcription is essential for maintaining genome integrity and executing other downstream cellular functions[1,2]. Here we identify a stable association between the genome-stability regulator sensor of single-stranded DNA (SOSS)[3] and the transcription regulator Integrator-PP2A (INTAC)[4–6]. Through SSB1-mediated recognition of single-stranded DNA, SOSS–INTAC stimulates promoter-proximal termination of transcription and attenuates R-loops associated with paused RNA polymerase II to prevent R-loop-induced genome instability. SOSS–INTAC-dependent attenuation of R-loops is enhanced by the ability of SSB1 to form liquid-like condensates. Deletion of *NABP2* (encoding SSB1) or introduction of cancer-associated mutations into its intrinsically disordered region leads to a pervasive accumulation of R-loops, highlighting a genome surveillance function of SOSS–INTAC that enables timely termination of transcription at promoters to constrain R-loop accumulation and ensure genome stability.

During transcription, nascent RNAs exiting the RNA polymerase II (Pol II) elongation complex can invade double-stranded DNA and rehybridize with template strands to form RNA–DNA duplexes known as R-loops[7]. R-loops are enriched at active promoters that contain high levels of paused Pol II[8–10] and contribute to replication stress and genome instability due to the vulnerability of the exposed single-stranded DNA (ssDNA) coding strands to mutagens and nucleases, while also blocking replication fork progression[11,12]. R-loops can also have beneficial regulatory roles in transcription, DNA repair and the immune response[13–16]. Moreover, dynamic control of R-loops contributes to the kinetics of transcriptional program switches during cell differentiation and reprogramming[17–19].

Biomolecular condensates formed through liquid–liquid phase separation (LLPS) have critical functions in various cellular processes, including transcriptional regulation, signal transduction and the DNA-damage response[20,21]. These membrane-less structures are typically enriched with proteins that contain repeated modular domains or long stretches of intrinsically disordered regions (IDRs). For example, the phase-separation behaviour of several R-loop regulatory factors has been reported to be linked to their IDRs[22].

Here we find that the transcription regulator INTAC regulates R-loop levels by associating with the ssDNA binding complex SOSS to form SOSS–INTAC. The SOSS–INTAC subunit SSB1, through ssDNA

recognition and a liquid-like condensate formation ability, localizes SOSS–INTAC at promoters and catalyses transcription termination to prevent aberrant R-loop accumulation to ensure genome stability.

## INTAC and SOSS form a stable complex

The 1.59 MDa INTAC complex, comprising 15 subunits of the RNA cleavage complex Integrator and the PP2A core enzyme (Extended Data Fig. 1a), regulates transcription by inducing the termination of promoter-proximally paused transcripts[4–6,23–27]. SOSS—a heterotrimeric DNA damage sensing and repair complex—contains INTS3 (also known as SOSS-A), the ssDNA-binding protein SSB1 (also known as SOSS-B1; or its paralogue SSB2 (encoded by *NABP1*, also known as SOSS-B2)), and INIP (also known as SOSS-C)[3,28,29] (Extended Data Fig. 1b). Given that both complexes contain INTS3, we posited that, together, they could mediate communication between transcription and genome stability machineries. To test this idea, we conducted immunoprecipitation (IP) followed by mass spectrometry analysis. Most subunits of SOSS and INTAC were retrieved after IP of INTS3 but not when using an IgG control (Fig. 1a). We next purified SSB1 and found that SSB1 interacts with other SOSS and INTAC subunits (Fig. 1a). Endogenous co-immunoprecipitation (co-IP) analysis confirmed that SSB1 associates with INTAC subunits (Fig. 1b).

[1]Fudan University Shanghai Cancer Center, Institutes of Biomedical Sciences, State Key Laboratory of Genetic Engineering, Shanghai Key Laboratory of Medical Epigenetics, Shanghai Key Laboratory of Radiation Oncology, Human Phenome Institute, Fudan University, Shanghai, China. [2]Zhejiang Provincial Key Laboratory for Cancer Molecular Cell Biology, Life Sciences Institute, Zhejiang University, Hangzhou, China. [3]Department of Radiation Oncology, Fudan University Shanghai Cancer Center, Fudan University, Shanghai, China. [4]Department of Thoracic Surgery, Shanghai Pulmonary Hospital, Tongji University School of Medicine, Shanghai, China. [5]Department of Pathophysiology, School of Basic Medical Sciences, Wuhan University, Wuhan, China. [6]Department of Histoembryology, Genetics and Developmental Biology, Shanghai Key Laboratory of Reproductive Medicine, Key Laboratory of Cell Differentiation and Apoptosis of Chinese Ministry of Education, Shanghai Jiao Tong University School of Medicine, Shanghai, China. [7]Inflammation and Immune Mediated Diseases Laboratory of Anhui Province, Anhui Institute of Innovative Drugs, School of Pharmacy, Anhui Medical University, Hefei, China. [8]Department of Orthopedic Oncology, Changzheng Hospital, Second Military Medical University, Shanghai, China. [9]Department of Breast Surgery, Fudan University Shanghai Cancer Center, Shanghai, China. [10]Huashan Hospital, Fudan University, Shanghai Key Laboratory of Medical Epigenetics, Molecular and Cell Biology Lab, Institutes of Biomedical Sciences, Shanghai Medical College of Fudan University, Shanghai, China. [11]These authors contributed equally: Congling Xu, Chengyu Li, Jiwei Chen, Yan Xiong. ✉e-mail: huasong_lu@zju.edu.cn; feixchen@fudan.edu.cn

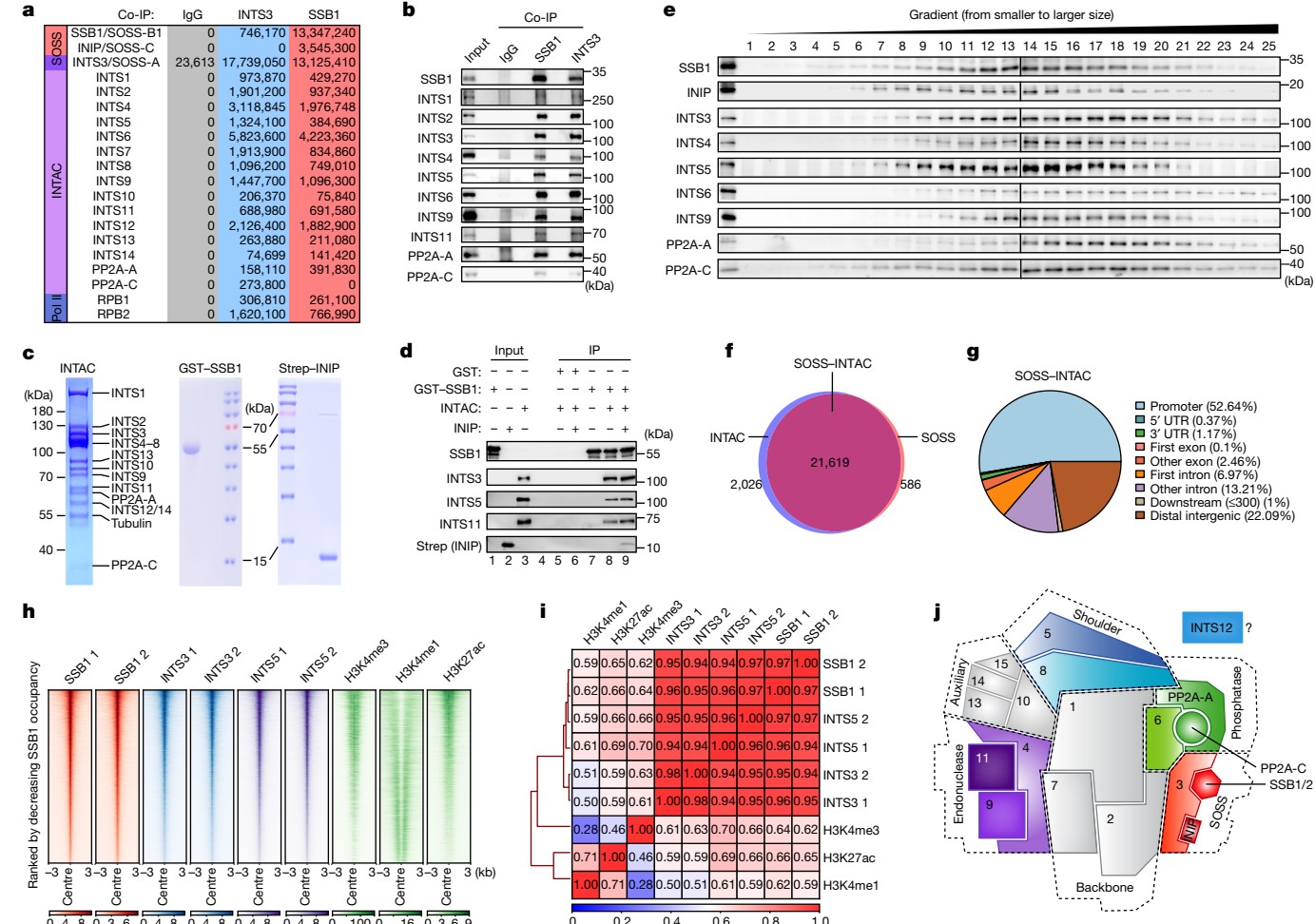

**Fig. 1 | Identification and genome-wide profiling of the SOSS–INTAC complex. a**, Mass spectrometry analyses of endogenous INTS3 and SSB1 IP using nuclear extracts. The values are intensity-based absolute quantification intensities for SOSS, INTAC and Pol II subunits. IgG was used as the binding control. **b**, Co-IP analysis of endogenous SSB1 and INTS3 followed by western blotting. Data represent two independent experiments. **c**, Coomassie staining of reconstituted human INTAC complex purified from HEK Expi293 cells, and GST-tagged human SSB1 and Strep-tagged human INIP proteins purified from *E. coli*. **d**, Immobilized GST or GST–SSB1 were incubated with purified INTAC in the presence or absence of INIP. The input and bound proteins were analysed by western blotting. Data represent two independent experiments. **e**, Gradient centrifugation using endogenous HEK Expi293 nuclear extracts. The fractionated samples were analysed using SDS–PAGE followed by western blotting. Data shown represent two independent experiments. **f**, The overlapping binding regions of INTAC (blue) and SOSS (red) in DLD-1 cells. **g**, The genomic distribution of SOSS–INTAC. **h**, ChIP–seq signals of SSB1,

INTS3, INTS5, H3K4me3, H3K4me1 and H3K27ac in DLD-1 cells. The peaks are centred on the SSB1 peak summits. **i**, Correlation analysis for the genomic occupancy of SSB1, INTS3, INTS5, H3K27ac, H3K4me3 and H3K4me1. The numbers are Pearson correlation coefficients. The ChIP–seq results shown represent two biologically independent samples. **j**, Schematic of the SOSS–INTAC complex. On the basis of structural and biochemical information[4,30,33,51,52], the complex can be divided into six modules, including the backbone (INTS1, INTS2 and INTS7), shoulder (INTS5 and INTS8), endonuclease (INTS4, INTS9 and INTS11), phosphatase (INTS6, PP2A-A and PP2A-C), auxiliary (INTS10/13/14/15) and SOSS (INTS3, SSB1/2, INIP) modules. The structural organization of the backbone, shoulder, endonuclease and phosphatase modules is illustrated on the basis of the structure of INTAC[4]. The organization of the SOSS module was placed according to the structures of SOSS[30] and INTS3/6[33]. The organization of the auxiliary module was estimated on the basis of structural and biochemical information of INTS10/13/14[52]. The structural placement of INTS12 is currently unclear.

To investigate associations of all SOSS subunits with INTAC, we over-expressed and purified protein-A-tagged SSB1, SSB2 and INIP in human embryonic kidney (HEK) Expi293 cells individually, followed by prot-eomics analysis. IP of each SOSS subunit successfully recovered most INTAC subunits (Extended Data Fig. 1c). The interaction between INIP and INTAC was further confirmed by Flag-tagged INIP overexpression followed by IP (Extended Data Fig. 1d). Our results suggest that the entire SOSS complex can be incorporated into INTAC.

To confirm the association between SSB1 and INTAC, we conducted in vitro pull-down assays using reconstituted INTAC complex from HEK Expi293 cells and purified SSB1 and INIP from *Escherichia coli* (Fig. 1c). INTAC subunits associated with GST-tagged SSB1 but not GST alone (Fig. 1d and Extended Data Fig. 1e). This interaction was not affected

by the presence of INIP (Fig. 1d; compare lanes 8 and 9), consistent with previous data showing the lack of a direct association between SSB1 and INIP[3,30]. Gradient centrifugation of nuclear extracts demonstrated co-migration of endogenous SSB1, INIP and INTAC (Fig. 1e), suggest-ing the existence of a stable SOSS–INTAC complex in cells. The major-ity of SSB1 and INTAC subunits co-localize at higher-molecular-mass fractions, further confirming the existence of SOSS–INTAC (Extended Data Fig. 1f).

## SOSS–INTAC targets active chromatin

To identify the genome locations of SOSS and INTAC, we performed chromatin IP followed by sequencing (ChIP–seq) analysis of SSB1, INTS3

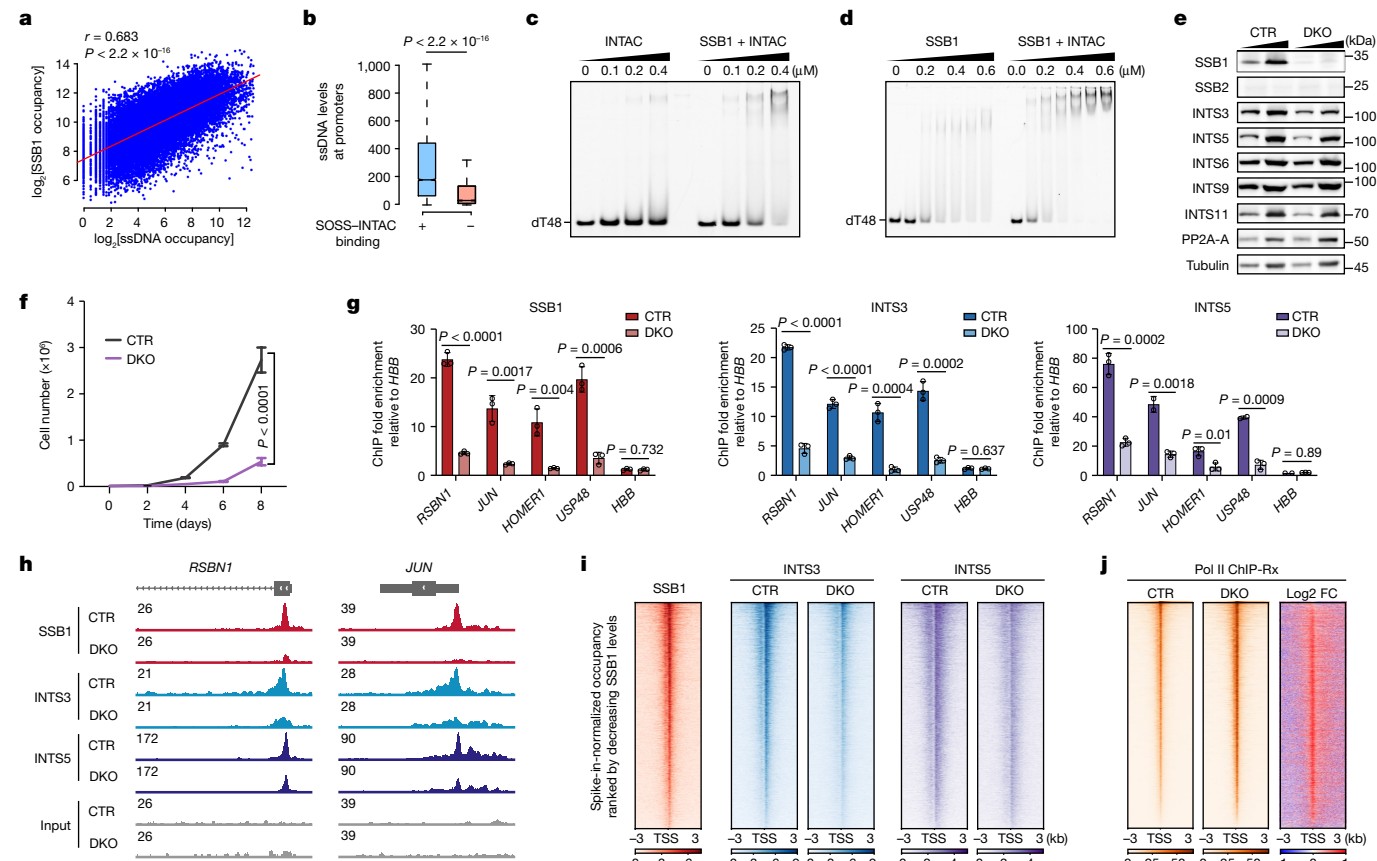

**Fig. 2 | SSB1 facilitates SOSS–INTAC recruitment to chromatin. a**, The correlation between ssDNA and SSB1 levels at SOSS–INTAC-bound regions in DLD-1 cells. $P$ values were computed using two-sided $t$-tests with 95% confidence intervals based on the Pearson's product moment correlation coefficient. $P < 2.2 \times 10^{-16}$. $n = 29{,}128$ peaks. **b**, ssDNA levels at promoters with or without SOSS–INTAC binding. For the box plots, the centre line indicates the median, the top and bottom hinges indicate the first and third quartiles, respectively, and the whiskers extend to the quartiles $\pm 1.5 \times$ interquartile range. $P$ values were calculated using two-sided Wilcoxon rank-sum tests. $P < 2.2 \times 10^{-16}$. **c,d**, EMSA using Cy3-labelled oligo (dT)48 incubated with INTAC alone (left) or with SSB1–INTAC proteins (right) (**c**), or with SSB1 alone (left) or with SSB1–INTAC proteins (right) (**d**). Data represent two independent experiments. **e**, Western blot analysis of whole-cell extracts from CTR (control, *NABP1* knockout) and DKO (*NABP2/NABP1* double-knockout) DLD-1 cells. Tubulin was

used as the loading control. Data represent two independent experiments. **f**, Growth curves of CTR and DKO DLD-1 cells. Data are mean $\pm$ s.d. $n = 4$ biological replicates. $P$ values were generated using two-way analysis of variance (ANOVA) performed for day 8. **g**, ChIP–qPCR experiments using SSB1 (red), INTS3 (blue) and INTS5 (purple) antibodies in CTR and DKO cells. Data are mean $\pm$ s.d. $n = 3$ biological replicates. Statistical analysis was performed using two-tailed $t$-tests. $P$ values are shown at the top of the graphs. **h**, Representative browser tracks showing ChIP–Rx signals of SSB1 (red), INTS3 (blue) and INTS5 (purple) in CTR and DKO cells. **i**, ChIP–Rx signals of SSB1, INTS3, INTS5 in CTR or DKO cells. Peaks are centred on transcription start site (TSS) of SOSS–INTAC-bound genes. **j**, Pol II ChIP–Rx signals on SOSS–INTAC target genes in CTR and DKO cells. Peaks are centred on the TSS and ranked by decreasing occupancy in CTR cells. FC, fold change.

and INTS5 in human colon adenocarcinoma DLD-1 cells (Extended Data Fig. 1g). To eliminate potential biases due to antibody efficiencies, only regions co-occupied by INTS3 and SSB1 were defined as reliable SOSS targets, whereas regions co-bound by INTS3 and INTS5 were considered to be faithful INTAC targets. A total of 21,619 loci co-bound by SOSS and INTAC comprise 97% of SOSS targets (Fig. 1f), mainly corresponding to promoter and intergenic regions (Fig. 1g and Extended Data Fig. 1h). Heat maps of SOSS–INTAC targets show a comparable occupancy of SSB1, INTS3 and INTS5 (Fig. 1h). Pearson correlation coefficient analysis shows that genomic distributions of SOSS–INTAC subunits are highly correlated with each other, in addition to their positive correlation with active chromatin marks of promoters and enhancers (Fig. 1i). Consistently, widespread binding of SOSS–INTAC at both active promoters and enhancers was observed (Extended Data Fig. 1i). The binding of SOSS–INTAC subunits on chromatin was further verified by ChIP followed by quantitative PCR (ChIP–qPCR) at promoters of example genes (Extended Data Fig. 1j). Together, these results reveal the formation of a stable SOSS–INTAC complex (Fig. 1j) that primarily localizes to promoter and enhancer regions.

## Recognition of ssDNA by SOSS–INTAC

We hypothesized that SSB1 contributes to SOSS–INTAC recruitment to promoters due to its potent ssDNA-binding ability and ssDNA being a prominent feature of actively transcribed regions[31]. To test this idea, we first confirmed that SSB1 preferentially binds to ssDNA but not to double-stranded DNA (dsDNA) or ssRNA on the basis of an electrophoretic mobility shift assay (EMSA) (Extended Data Fig. 2a). Using a kethoxal-assisted single-stranded DNA sequencing (KAS-seq) protocol[31], we found that SOSS–INTAC occupancy is positively correlated with ssDNA levels genome-wide, including at promoters and enhancers (Fig. 2a and Extended Data Fig. 2b–d). We next compared the ssDNA levels at promoters with and without SOSS–INTAC binding, which revealed greater enrichment of ssDNA at SOSS–INTAC-bound promoters (Fig. 2b).

To confirm direct ssDNA-binding ability, we performed EMSA using synthesized oligo (dT)48 incubated with INTAC alone or SSB1–INTAC protein[30]. INTAC alone has a weak ssDNA-binding affinity, probably mediated by its INTS3 subunit[32,33] (Fig. 2c (left)). Notably, adding SSB1

substantially boosts the interaction with the oligo (Fig. 2c (right)), indicating a key role of SSB1 in recognizing ssDNA. Compared with the migration of bands seen with the SSB1–ssDNA complex, super-shifted bands were observed after incubation of SSB1 with INTAC, suggesting the co-migration of SSB1–INTAC with ssDNA (Fig. 2d). These results support the conclusion that SSB1 facilitates the recruitment of SOSS–INTAC by recognizing ssDNA.

## SSB1 regulates SOSS–INTAC localization

In contrast to the ubiquitous expression of SSB1, its paralogue SSB2 is expressed tissue specifically and could have redundant roles with SSB1 in certain contexts[34,35] (Extended Data Fig. 2e,f). To avoid this potential redundancy, we generated *NABP1*-null cells to be used as a control cell line (hereafter, CTR cells) for later experiments. CTR cells exhibit no defect in cell growth (Extended Data Fig. 2g). As measured by western blotting and ChIP–qPCR, SOSS–INTAC protein stability and occupancy at the tested genes were not affected by the deletion of *NABP1* (Extended Data Fig. 2h,i). *NABP1/NABP2* double knockout cells (hereafter, DKO cells) were generated by additionally deleting *NABP2* in pooled cells to eliminate clonal variations and to minimize long-term culture-induced secondary effects (Fig. 2e). Compared with the CTR cells, DKO cells exhibit growth defects (Fig. 2f) and diminished INTAC occupancy at target genes (Fig. 2g). Induced expression of either SSB1 or SSB2 rescues the growth defects, corroborating the redundancy of these paralogues (Extended Data Fig. 2j).

To determine the genome-wide regulation of INTAC recruitment by SSB1, we performed calibrated INTS3 and INTS5 ChIP–seq analysis with reference exogenous genome as the spike-in (ChIP–Rx) in CTR and DKO cells. Track examples and genome-wide analyses show decreased INTS3 and INTS5 occupancies at promoters after SSB1 loss (Fig. 2h,i and Extended Data Fig. 3a–e). To determine how ssDNA recruits INTAC to chromatin, we generated SSB1 mutants that specifically compromise DNA binding (W55A/F78A) or disrupt the SSB1–INTAC interaction (E97A/F98A)[30] (Extended Data Fig. 3f,g). As shown by ChIP–qPCR analysis of example genes, both mutants exhibit reduced recruitment of INTAC, indicating that both the DNA-binding ability of SSB1 and its ability to interact with INTAC are required for the optimal association of INTAC with promoters (Extended Data Fig. 3h).

## SOSS–INTAC modulates Pol II occupancy

The INTAC complex is a major regulator of promoter-proximal termination of paused Pol II[4–6,23–26,36]. To evaluate whether the SOSS module of SOSS–INTAC regulates Pol II pausing, we conducted Pol II ChIP–Rx and observed a widespread increase in Pol II occupancy at promoters of SOSS–INTAC targets in DKO cells compared with in CTR cells (Fig. 2j, Extended Data Fig. 3i and Supplementary Fig. 2a–c). Using Pol II levels to normalize SOSS–INTAC subunit occupancy, SSB1, INTS3 and INTS5 were each markedly reduced in DKO cells, corroborating the notion that SSB1 recruits SOSS–INTAC to chromatin (Supplementary Fig. 2d–f). As previous reports described differential regulation of Pol II progression by Integrator depending on exon number, overall length and the coding or non-coding status of genes[24–26,37], we grouped genes by these properties; this demonstrated a general accumulation of Pol II at promoters for all gene classes. Pol II occupancy changes in gene bodies varied between classes, with monoexonic, non-coding and shorter genes exhibiting a substantially greater increase in polymerase levels in DKO cells compared with at longer or multiexonic genes (Supplementary Fig. 2g–i), consistent with a loss of Integrator function in DKO cells. Moreover, the accumulation of Pol II at promoters was recapitulated by the depletion of *INTS2*, supporting the functional connection between SOSS and INTAC (Extended Data Fig. 3j and Supplementary Fig. 2j–l). The pausing index—the ratio of Pol II occupancy at promoters over gene bodies, indicating the extent of pausing—is evidently higher after the loss of SSB1 or INTS2 (Supplementary Fig. 2m,n).

To measure paused Pol II changes after transcription initiation, we next used precision run-on sequencing (PRO–seq) to quantify nascent transcripts at the single-base resolution. Loss of SSB1 induces the accumulation of paused Pol II at promoters (Extended Data Fig. 3k–m and Supplementary Fig. 3a,b), in agreement with the disruption of INTAC or Integrator leading to defects in promoter-proximal termination[4–6,23,37]. Corroborating these findings, the levels of Pol II phosphorylated at serine 5 of its C-terminal domain, representing paused Pol II, were substantially increased in DKO cells (Extended Data Fig. 3n). Assay for transposase-accessible chromatin with sequencing (ATAC–seq) analyses demonstrated increased chromatin accessibility at SOSS–INTAC targets in DKO cells, probably resulting from Pol II accumulation (Extended Data Fig. 3o,p and Supplementary Fig. 3c). Indeed, as shown at example genes, changes in Pol II occupancy and chromatin accessibility were comparable (Extended Data Fig. 3i,q and Supplementary Fig. 3d,e). Thus, SOSS–INTAC prevents the accumulation of paused Pol II and limits chromatin accessibility.

## R-loops affect SOSS–INTAC localization

Owing in part to the higher thermodynamic stability of an RNA–DNA duplex compared with dsDNA, R-loops can accumulate at actively transcribed genomic regions, especially at promoters containing the highest levels of Pol II and associated short nascent transcripts[8–10,38]. To investigate whether promoter-associated R-loops modulate SOSS–INTAC recruitment to chromatin, we established a cell line with inducible expression of RNase H1, which degrades the RNA strand of RNA–DNA duplexes and can therefore resolve R-loops (Extended Data Fig. 4a). As shown at example genes, SSB1 levels decrease at promoters after doxycycline (DOX) treatment (Extended Data Fig. 4b,c). R-loop CUT&Tag followed by qPCR confirmed the decrease in R-loops at the corresponding promoter regions after the induction of RNase H1 expression (Extended Data Fig. 4d,e). Furthermore, RNase H1 over-expression induces a genome-wide attenuation of SSB1 occupancy at promoters (Fig. 3a,b and Extended Data Fig. 4f). INTS3 occupancy at SSB1-bound regions is similarly reduced after RNase H1 overexpression (Extended Data Fig. 4g–k). These results indicate that R-loops can be recognized by SSB1, leading to increased SOSS–INTAC at these promoters.

## SOSS–INTAC attenuates R-loop levels

On the basis of our findings that SSB1-mediated recruitment of SOSS–INTAC controls promoter-proximal termination and chromatin accessibility at promoters, we speculated that SOSS–INTAC could reciprocally influence R-loop levels. To examine this hypothesis, we measured cellular R-loop levels on the basis of immunofluorescence analysis using the S9.6 antibody, which recognizes RNA–DNA hybrids. Notably, a strong elevation in nuclear S9.6 signals was observed in SSB1- or INTS2-depleted cells (Extended Data Fig. 5a–d; DMSO conditions). Importantly, the accumulation of these nuclear signals could be suppressed by DOX-induced overexpression of wild-type RNase H1 (Extended Data Fig. 5a–d; DOX conditions), indicating that the S9.6 antibody is detecting nuclear R-loop increases after loss of SSB1 and INTS2.

As the S9.6 antibody detects dsRNAs in addition to RNA–DNA hybrids, we used a purified GFP-tagged catalytic-dead RNase H1 protein (GFP–dRNASEH1) as the R-loop sensor[39,40]. Although pretreatment with ssRNA endonuclease RNase T1 and dsRNA endonuclease RNase III greatly eliminates the signals detected by S9.6, it has no notable effect on the GFP–dRNASEH1 signal, suggesting that R-loop measurements made with GFP–dRNASEH1 are unlikely to be confounded by ssRNA

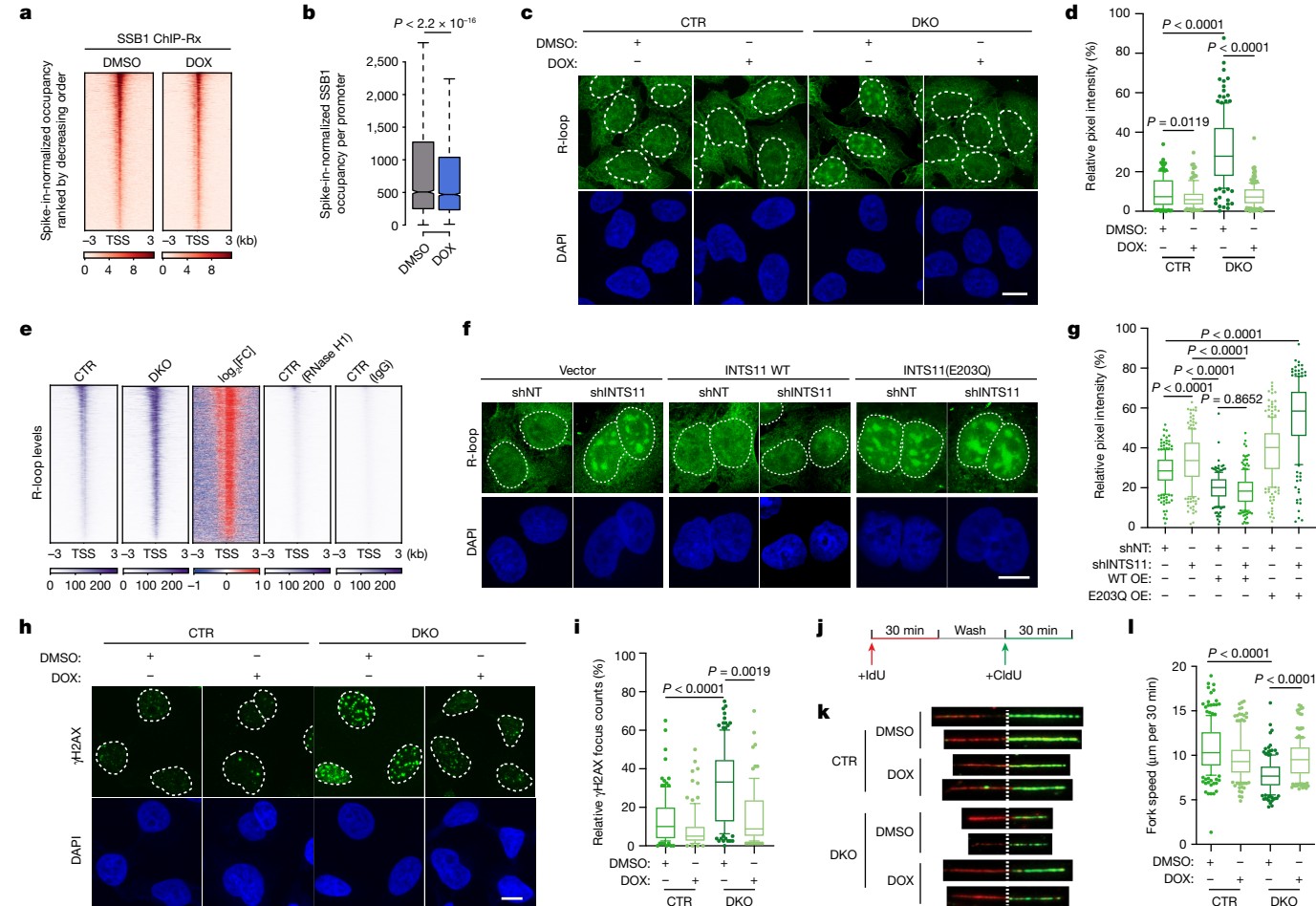

**Fig. 3 | SOSS–INTAC regulates R-loop levels. a**, SSB1 occupancy over 6 kb regions centred on the TSS of SOSS–INTAC target genes in DLD-1 cells with DOX-inducible RNase H1 expression. **b**, Comparison of SSB1 occupancy at SOSS–INTAC target promoters for DMSO- and DOX-treated cells. For the box plots, the centre line indicates the median, the top and bottom hinges indicate the first and third quartiles, respectively, and the whiskers extend to the quartiles $\pm 1.5 \times$ interquartile range. $P$ values were calculated using two-sided Wilcoxon rank-sum tests. $P < 2.2 \times 10^{-16}$. $n = 10,650$ promoters. **c**, R-loop detection in CTR and DKO cells with DOX-inducible GFP–RNASEH1 expression. Scale bar, 10 μm. **d**, Quantification of nuclear R-loop signals for **c**. $P$ values were calculated using two-tailed unpaired $t$-tests. $n = 110$ foci from one representative experiment, which was performed twice with similar results. The centre lines indicate the median values. **e**, R-loop CUT&Tag signals over 6 kb regions centred on the TSS of SOSS–INTAC target genes in CTR and DKO cells. CTR cells were treated with RNase H1 protein during CUT&Tag (lane 4) or incubated with IgG (lane 5) to confirm the specificity of detected R-loop signals. **f**, Immunofluorescence analysis of R-loop signals in DLD-1 cells with INTS11 or non-targeting (NT) shRNA and overexpression of wild-type (WT) or

catalytically dead (E203Q) INTS11 and empty vector control. Scale bar, 10 μm. **g**, Quantification of the nuclear R-loop signals for **f**. Statistical analysis was performed using two-tailed unpaired $t$-tests; $P$ values are shown above the graphs. $n = 180$ foci from one representative experiment, which was performed twice with similar results. The centre lines indicate the median values. **h**, Immunostaining of γH2AX signals in CTR and DKO cells with DOX-inducible RNase H1 expression. Scale bar, 10 μm. **i**, Quantification of the γH2AX focus number in **h**. Statistical analysis was performed using two-tailed unpaired $t$-tests; $P$ values are shown above the graphs. $n = 90$ foci from one representative experiment, which was performed twice with similar results. The centre lines indicate the median values. **j**, Schematic of the DNA fibre assay. Cells were sequentially pulsed with two different thymidine analogues–IdU and CIdU. **k**, Representative images of stretched DNA fibres. CTR and DKO cells with DOX-inducible RNase H1 expression were treated with DMSO or DOX as indicated. Red tracks, IdU; green tracks, CIdU. **l**, Replication fork speed was measured by IdU (red) and CIdU (green) incorporation. $P$ values were determined using two-tailed unpaired $t$-tests. $n = 160$ fibres were measured for each group.

and dsRNA binding[39,40] (Supplementary Fig. 4). We therefore used GFP–dRNASEH1 to quantify cellular R-loop levels in further studies.

The loss of SSB1 induces the formation of R-loop foci and higher R-loop levels, which are eliminated by DOX-induced expression of wild-type RNase H1 (Fig. 3c,d). Quantitative analysis shows that R-loop levels in DKO cells are substantially higher than in CTR cells (Fig. 3d). R-loop CUT&Tag was used to evaluate genome-wide changes in R-loops, revealing a large-scale induction of R-loops in DKO cells that was eliminated by treatment with RNase H1, further indicating that the measured signals are bona fide R-loops (Fig. 3e and Extended Data Fig. 5e–g). Loss of INTS2 elicits a similar accumulation of cellular R-loops (Extended Data Fig. 5h,i). To determine whether R-loop regulation by SSB1 is

mediated through the endonuclease activity of SOSS–INTAC, we depleted INTS11, the catalytic subunit of the endonuclease module, and rescued INTS11 loss with ectopic expression of wild-type or catalytically dead (E203Q) INTS11 (Extended Data Fig. 5j,k). INTS11 depletion alone induces substantial R-loop accumulation (Fig. 3f,g and Extended Data Fig. 5l). This accumulation was rescued by wild-type but not catalytically dead INTS11, and simultaneous *INTS11* knockdown and expression of catalytically dead INTS11 gave rise to the greatest R-loop enrichment (Fig. 3f,g and Extended Data Fig. 5l). To corroborate the functional connection between SOSS and INTAC in R-loop regulation, we over-expressed wild-type SSB1 or SSB1(E97A/F98A), the mutant defective in INTAC interaction, in DKO cells. Notably, wild-type SSB1 but not the

SSB1(E97A/F98A) mutant prevents R-loop accumulation in DKO cells (Extended Data Fig. 5m). These data reveal a function for SOSS–INTAC in preventing aberrant R-loop accumulation.

We next examined whether RNA exonucleases facilitate R-loop removal after RNA cleavage by SOSS–INTAC. The major 5′ and 3′ exonucleases responsible for RNA degradation in the nucleus are XRN2 and the exosome complex, respectively. We therefore depleted XRN2, two catalytic subunits of the exosome (DIS3 and EXOSC10) and the nuclear exosome-targeting (NEXT) complex MTR4 subunit that unwinds structured RNA substrates for exosomal degradation (Extended Data Fig. 6a,b). As shown by R-loop CUT&Tag–qPCR analysis, individual depletion of XRN2, DIS3 and MTR4 induces a small but significant upregulation of R-loops at promoters in CTR cells. Simultaneous loss of XRN2 and DIS3 leads to greater R-loop accumulation, indicating that both XRN2 and the exosome contribute to R-loop attenuation (Extended Data Fig. 6c (left)). Although the loss of SSB1 in DKO cells leads to upregulation of R-loops, additional disruption of XRN2 and the exosome does not augment this change (Extended Data Fig. 6c (right)). SOSS–INTAC-loss-induced R-loop accumulation could be epistatic to that caused by disrupting XRN2 and the exosome, whereby endonucleolytic cleavage of RNA by SOSS–INTAC could expose the 5′ and 3′ ends for exonucleolytic digestion by XRN2 and the exosome.

We next examined whether the recruitment of XRN2 and the exosome are regulated by SOSS–INTAC. Notably, the promoter occupancy of XRN2, but not exosome or NEXT subunits, is compromised in DKO cells (Extended Data Fig. 6d). Plotting XRN2 occupancy for genes of different classes indicated a highly similar pattern of XRN2 and Pol II (compare Supplementary Figs. 2g–i and 5a–c), as previously reported[41]. The importance of the endonuclease activity of SOSS–INTAC in these processes is demonstrated by the ability of wild-type but not catalytically dead INTS11 to reverse the changes in Pol II occupancy (Supplementary Fig. 5d,e).

## SOSS–INTAC regulates genome stability

Unresolved R-loops can expose ssDNA to damaging agents and induce DNA damage by forming obstacles to replication fork progression, causing transcription–replication conflicts and DNA breaks[11,12]. We therefore measured γH2AX levels using immunofluorescence and found that the loss of SSB1 in DKO cells stimulates the accumulation of γH2AX, whereas DOX-induced RNase H1 overexpression suppresses γH2AX induction after SSB1 depletion (Fig. 3h,i). γH2AX CUT&Tag analysis further demonstrates elevated γH2AX levels at promoters in DKO cells (Extended Data Fig. 6e), consistent with the accumulation of R-loops at corresponding loci. Knockout of *INTS2* induces a comparable change in γH2AX levels at the cellular or genome-wide scale (Extended Data Fig. 6f–h).

Flow cytometry analysis after propidium iodide staining and γH2AX labelling revealed an induction of γH2AX in both G1 and S phases (Extended Data Fig. 6i). We posited that SSB1 loss induces genome instability in part through impeding replication-fork progression. We therefore quantified replication-fork velocity by consecutive pulse labelling with thymidine analogues 5-iodo-2′-deoxyuridine (IdU) and 5-chloro-2′-deoxyuridine (CldU) (Fig. 3j). Disruption of SOSS–INTAC in DKO cells resulted in retarded replication fork progression, which was partially rescued by RNase H1 induction in DOX-treated cells (Fig. 3k,l). These results support that SSB1-mediated SOSS–INTAC recruitment is crucial for restraining R-loop levels and maintaining genome stability.

## SOSS–INTAC forms nuclear puncta

The ability of SSB1, a relatively small (22 kDa) protein, to govern the recruitment of SOSS–INTAC, a complex that is around 70 times larger, motivated us to further investigate the biochemical features of SSB1.

Comparing the distributions of reconstituted SOSS–INTAC (Extended Data Fig. 1f) with INTAC alone (Extended Data Fig. 7a) after fractionation by gradient centrifugation, we noticed that the association between SOSS and INTAC causes a substantial shift to higher-molecular-mass fractions that cannot be explained by the size of SOSS alone (Fig. 4a), suggesting SOSS-dependent multivalent interactions or oligomerization. *E. coli* SSB contains an IDR at its C terminus that drives LLPS[42]. Human SSB1 has an even more disordered C-terminal IDR compared with its *E. coli* counterpart (Fig. 4b), and the percentage of IDR regions and the disorder intensity of SSB1 are considerably greater compared with other SOSS–INTAC subunits (Extended Data Fig. 7b).

To examine the condensation ability of SSB1, we conducted immunofluorescence using an anti-SSB1 antibody and detected nuclear puncta (Fig. 4c). Lacking suitable antibodies for INTAC immunofluorescence, we knocked-in an N-terminal Flag tag at the endogenous loci of two INTAC phosphatase module subunits (INTS5 and INTS8) and two INTAC endonuclease module subunits (INTS4 and INTS11). The immunofluorescence results indicate the presence of INTAC puncta co-localizing with SSB1 nuclear foci (Fig. 4c).

To investigate the interdependency of SSB1 and INTAC for punctum formation, we first depleted SSB1 and SSB2 simultaneously in the cells expressing the Flag–INTAC subunit and performed Flag immunofluorescence analysis. Notably, the loss of SSB1 and SSB2 abolishes punctum formation of INTAC subunits (Extended Data Fig. 7c). However, depletion of INTS11 exerts no noticeable impact on the formation of SSB1 puncta (Extended Data Fig. 7d), indicating that SSB1/2 is the major driver of punctum formation.

## SSB1 forms liquid-like condensates

We next examined whether human SSB1 has the ability to form condensates in vitro using protein purified from *E. coli*. Fluorescence microscopy analysis showed that GFP-tagged SSB1 readily self-associates as micrometre-sized spherical droplets in the absence of crowding reagents (Fig. 4d,e). This droplet formation is sensitive to increased ionic strength, indicating the requirement of electrostatic interactions for SSB1 condensation. Sequentially lowering and increasing salt concentration induces a rapid appearance and disappearance of SSB1 droplets, proving its liquid-like property (Fig. 4f). Moreover, 1,6-hexanediol (1,6-Hex), a compound that perturbs weak multivalent interactions and disassembles structures exhibiting liquid-like properties, hinders droplet formation (Fig. 4g,h). Without the assistance of crowding reagents, the number and size of SSB1 droplets increase gradually when increasing the SSB1 protein concentration (Extended Data Fig. 8a,b). In agreement with the liquid-like property, SSB1 droplets are highly dynamic and readily coalesce into larger ones that are immediately relaxed into a spherical structure (Fig. 4i). Fluorescence signals recover within 2 min after photobleaching in the centre of the droplet (Fig. 4j), consistent with liquid-like condensates.

To examine whether SSB1 can form liquid-like condensates in cells, we used the optoDroplet system fusing SSB1 with mCherry-labelled *Arabidopsis* photoreceptor cryptochrome 2 (CRY2)[43]. We found that droplet formation of SSB1, but not the control, was substantially increased after light induction (Extended Data Fig. 8c). Moreover, SSB1 puncta undergo frequent fusion and fission events (Extended Data Fig. 8d,e). The fluorescence signals of foci recover readily after photobleaching (Extended Data Fig. 8f), which is indicative of liquid-like behaviour. On the basis of these findings, we conclude that SSB1 forms liquid-like condensates in vitro and in cells.

## SSB1 drives SOSS–INTAC condensation

In contrast to the clearly formed SSB1 droplets (green), no condensates were observed for labelled INTAC (red) alone in the same

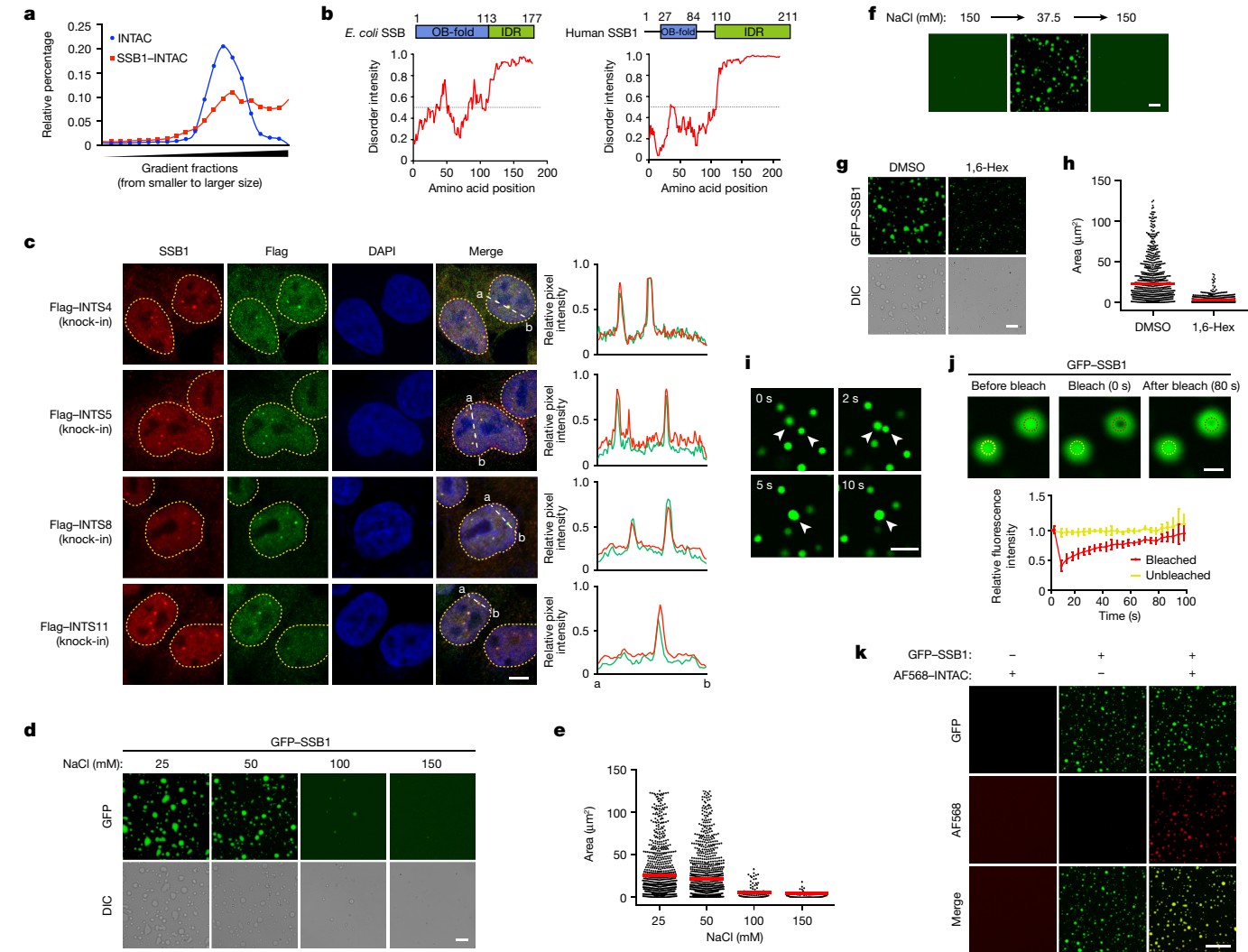

**Fig. 4 | SSB1 drives the formation of SOSS–INTAC condensates.**
**a**, Quantification of purified INTAC (all subunits) and SSB1–INTAC distribution after sucrose density-gradient centrifugation and western blotting. Five subunits were used for quantification (INTS5, INTS6, INTS11, PP2A-A and PP2A-C). **b**, The domain structure and the intrinsically disordered tendency of *E. coli* SSB (left) and human SSB1 (right). IUPred assigned scores of disordered tendencies between 0 and 1 to the sequences, and a score of higher than 0.5 indicates disorder. **c**, Representative images showing the relative locations of endogenous SSB1 and INTAC subunits along with the DAPI signal in DLD-1 cells. Representative curves (right) describe the distribution of relative fluorescence intensities for SSB1 (red) and INTAC subunits (green). Data represent two independent experiments. **d,e**, GFP–SSB1 (50 μM) was analysed using droplet formation assays with the indicated concentrations of NaCl (**d**), and the size of the droplets was quantified (**e**). Each dot represents a droplet. *n* = 100 foci from one representative experiment, which was performed twice with similar results. The red lines indicate the mean value in each population. **f**, NaCl concentrations in the GFP–SSB1 solution were changed sequentially as indicated and then examined under a fluorescence microscope. **g,h**, 1,6-Hex (5%) treatment disrupts droplet formation. GFP–SSB1 (50 μM) was analysed with 37.5 mM NaCl with or without 5% 1,6-Hex (**g**), and the size of droplets was quantified (**h**). Each dot represents a droplet. The red lines indicate the mean value in each population. **i**, Time-lapse imaging of GFP–SSB1 droplets undergoing spontaneous fusions as indicated by the arrows. **j**, Representative micrographs of GFP–SSB1 droplets before and after photobleaching (top). FRAP quantification of GFP–SSB1 droplets over a period of 100 s (bottom). *n* = 3 droplets analysed from 1 representative experiment, which was performed 3 times with similar results. **k**, GFP–SSB1 and Alexa Fluor 568 (AF568)-labelled INTAC (all subunits), either individually or mixed together as indicated, were analysed using a droplet formation assay and then examined under a fluorescence microscope. Scale bars, 5 μm (**c**, **i** and **j**), 20 μm (**d**, **f** and **g**) and 50 μm (**k**).

condition (Fig. 4k and Extended Data Fig. 8g,h), in agreement with predicted disorder intensities (Extended Data Fig. 7b). However, after mixing together, SSB1 and INTAC co-form droplets, suggesting that SSB1 drives the formation of SOSS–INTAC condensates (Fig. 4k and Extended Data Fig. 8g,h). To determine whether INTAC modulates SSB1 condensation formation, we incubated different concentrations of SSB1 with INTAC. Although increasing SSB1 concentrations stimulate INTAC droplet formation, the condensation capacity of SSB1 is at most marginally affected by the presence of INTAC (Extended Data Fig. 8i,j), further indicating that SSB1 drives the formation of SOSS–INTAC condensates.

## SSB1 mutations impair condensation

To confirm whether the SSB1 IDR is required for droplet formation, we generated SSB1 lacking the IDR (Fig. 5a and Extended Data Fig. 8k), which did not form droplets alone (Extended Data Fig. 8l,m) or in the context of SOSS–INTAC (Fig. 5b,c) in vitro. To determine the essential amino acids within the IDR that mediate SSB1 droplet formation, we mutated all IDR-enriched residues, except for alanine and proline, to IDR-depleted residues bearing comparably sized side chains, and successfully purified three soluble mutants—SSB1(HY) (all histidine to tyrosine), SSB1(SI) (all serine to isoleucine) and SSB1(RY) (all arginine to tyrosine) (Fig. 5a and

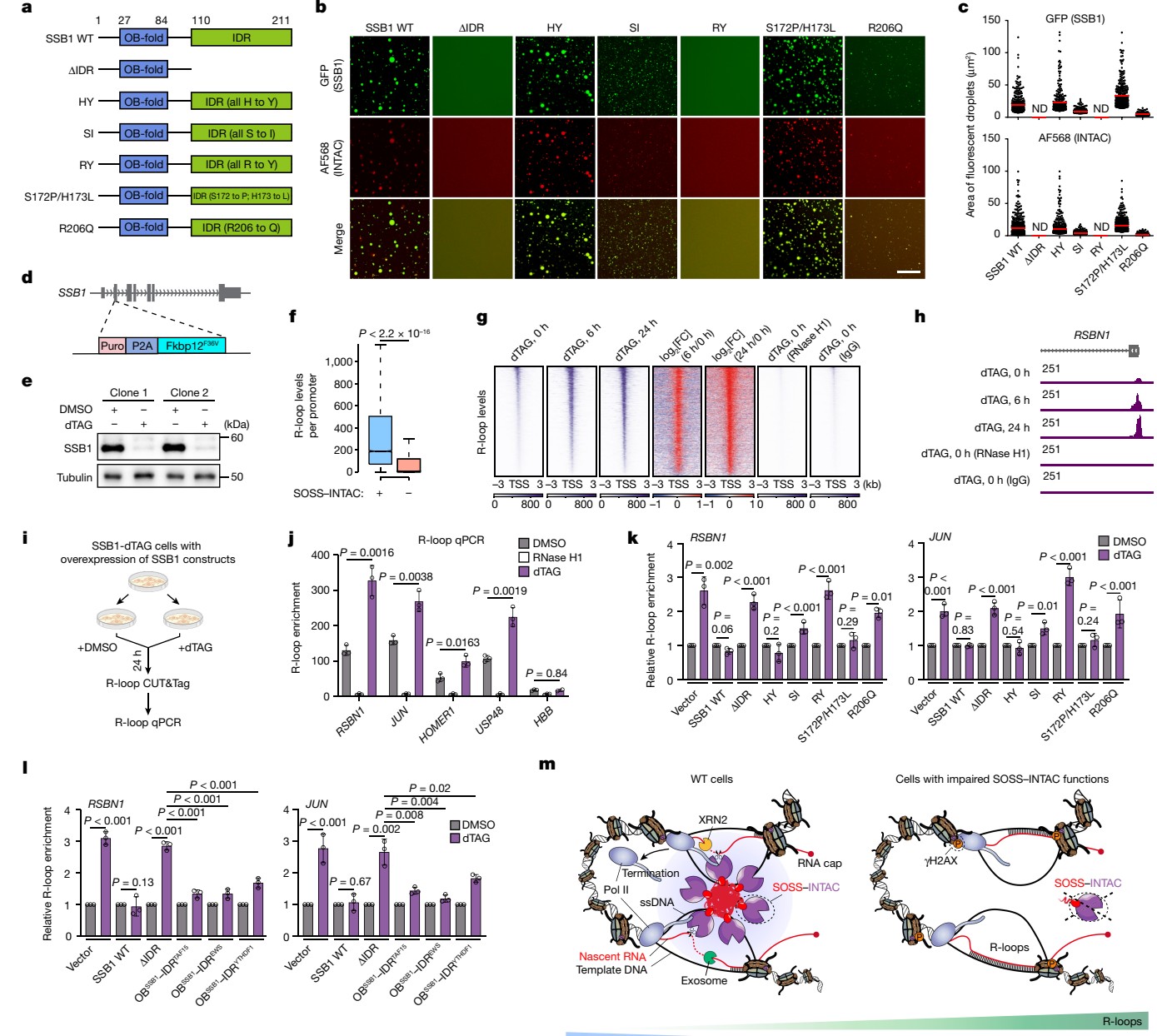

**Fig. 5 | SOSS–INTAC condensation regulates R-loop levels. a**, Schematic of the SSB1 domains and SSB1 mutants. OB-fold, oligonucleotide/oligosaccharide-binding fold. **b,c**, Fluorescence microscopy analysis of purified GFP–SSB1 mutants mixed with Alexa-Fluor-568-labelled INTAC (all subunits) (**b**), and quantification of the GFP and Alexa Fluor 568 signal (**c**). $n = 1,500$ foci were analysed across two independent experiments. The red lines indicate the mean values. Scale bars, 50 μm (**b**). ND, not detected. **d,e**, Schematic of the generation of SSB1-dTAG DLD-1 cells (**d**) and verification of SSB1 degradation by treatment for 6 h with dTAG (100 nM) (**e**). **f**, The R-loop levels at promoters with or without SOSS–INTAC binding measured by R-loop CUT&Tag under the DMSO-treated condition in SSB1-dTAG cells. For the box plots, the centre line indicates the median, the top and bottom hinges indicate the first and third quartiles, respectively, and the whiskers extend to the quartiles ± 1.5 × interquartile range. $P$ values were calculated using two-sided Wilcoxon rank-sum tests. **g**, R-loop CUT&Tag signals over 6 kb regions centred on the TSS of SOSS–INTAC target genes in SSB1-dTAG cells with dTAG time-course treatment. One sample was treated with RNase H1 protein during CUT&Tag to verify the specificity of R-loop signals. **h**, Representative browser tracks showing the R-loop signals in SSB1-dTAG cells with time-course dTAG treatment. **i**, Schematic of the R-loop CUT&Tag–qPCR workflow. **j**, R-loop CUT&Tag–qPCR analysis of example genes in SSB1-dTAG DLD-1 cells after 24 h treatment of DMSO or dTAG. The RNase H1

control was as shown in **h**. Data are mean ± s.d. $n = 3$ biological replicates. Statistical analysis was performed using two-tailed unpaired $t$-tests; $P$ values are shown above the graphs. **k**, R-loop CUT&Tag–qPCR analysis of DMSO- or dTAG-treated SSB1-dTAG cells with overexpression of wild-type, mutant SSB1 or empty vector. Data are mean ± s.d. $n = 3$ biological replicates. Statistical analysis was performed using two-tailed unpaired $t$-tests; $P$ values are shown above the graphs. **l**, R-loop CUT&Tag–qPCR analysis of DMSO- or dTAG-treated SSB1-dTAG cells with overexpression of wild-type SSB1 or fusion proteins comprising the N terminus of SSB1 and IDR from TAF15, EWS or YTHDF1. Data are mean ± s.d. $n = 3$ biological replicates. Statistical analysis was performed using two-tailed unpaired $t$-tests; $P$ values are shown above the graphs. **m**, Working model demonstrating the proposed mechanism by which SOSS–INTAC attenuates R-loop accumulation and maintains genome stability. In wild-type cells, the SSB1 subunit of SOSS interacts with ssDNA to recruit SOSS–INTAC to promoters and drives condensate formation. RNA cleavage by SOSS–INTAC condensates permits RNA degradation by a combination of XRN2 and exosome activities, leading to premature promoter-proximal termination by RNA Pol II and R-loop attenuation. Cancer-associated mutations of SSB1 that impair condensation and disrupt SOSS–INTAC recruitment lead to the loss of premature promoter-proximal Pol II termination and aberrant accumulation of R-loops, with potential adverse consequences, such as DNA damage.

Extended Data Fig. 8k,n,o). As shown by fluorescence microscopy, the SSB1(HY) mutation does not affect in vitro droplet formation, whereas SSB1(SI) significantly compromises in vitro droplet formation (Fig. 5b,c and Extended Data Fig. 8l,m). Notably, the SSB1(RY) mutation completely abolishes condensate formation (Fig. 5b,c and Extended Data Fig. 8l,m), highlighting the essentiality of arginine within the C-terminal IDR in mediating the condensation ability of SSB1.

The SSB1 IDR contains three potential cancer mutation hotspots at Ser172, His173 and Arg206 (Extended Data Fig. 8p). To elucidate whether these affect SSB1 condensation, we generated two constructs SSB1(S172P/H173L) (Ser172 to proline and His173 to leucine) and SSB1(R206Q) (Arg206 to glutamine) based on cancer-derived mutations (Fig. 5a and Extended Data Fig. 8k). As confirmed by EMSA and co-IP, both mutant proteins retain ssDNA binding (Extended Data Fig. 8q) and INTAC association (Extended Data Fig. 8r). SSB1(S172P/H173L) forms droplets as readily as wild-type SSB1, whereas SSB1(R206Q) exhibits severely impaired condensate formation (Fig. 5b,c and Extended Data Fig. 8l,m). For all of the SSB1 mutant proteins tested, the condensation ability was not affected by the presence of INTAC (Fig. 5b,c and Extended Data Fig. 8l,m), corroborating that SSB1 drives the formation of SOSS–INTAC condensates.

## Dynamic regulation of R-loops by SSB1

To investigate the dynamic change of R-loop levels after SSB1 depletion, we introduced the FKBP12[F36V] degradation tag N-terminally at the endogenous *NABP2* locus in CTR cells[44] (SSB1-dTAG cells; Fig. 5d). Addition of dTAG-13 (hereafter, dTAG) induces rapid depletion of endogenous SSB1 and induction of R-loop levels in SSB1-dTAG cells (Fig. 5e and Extended Data Fig. 9a–c). γH2AX signals are enhanced substantially after SSB1 depletion, recapitulating the dynamics in R-loop levels (Extended Data Fig. 9d,e). To determine the genomic features of R-loops, we performed CUT&Tag and quantified R-loop levels in SSB1-dTAG cells with dTAG treatment for 6 h and 24 h. Consistent with R-loops facilitating SOSS–INTAC recruitment, SOSS–INTAC-occupied promoters show higher R-loop levels (Fig. 5f). SSB1 degradation induces a pervasive accumulation of R-loops at SOSS–INTAC-bound promoters (Fig. 5g and Extended Data Fig. 9f), as also seen at example genes (Fig. 5h and Extended Data Fig. 9g). RNase H1 treatment eliminates the R-loop CUT&Tag signal (Fig. 5g,h and Extended Data Fig. 9f,g), confirming its specificity. Accumulation of R-loops was verified by R-loop CUT&Tag–qPCR at example genes (Fig. 5i,j), showing consistency with R-loop CUT&Tag–seq.

## SSB1 condensation suppresses R-loops

To examine whether SSB1 condensate formation contributes to R-loop regulation, we conducted rescue experiments with wild-type or mutant SSB1 in SSB1-dTAG cells (Extended Data Fig. 9h). Consistent with in vitro results (Fig. 5b,c), punctum formation was abolished with the SSB1(ΔIDR) and SSB1(RY) mutants, and severely impaired with the SSB1(SI) and SSB1(R206Q) mutants in dTAG-treated cells (Extended Data Fig. 9i,j). Testing all of the mutant constructs described above, we found that SSB1(RY) and SSB1(SI) did not fully rescue R-loop levels compared with wild-type SSB1 (Fig. 5k). The cancer-derived mutant SSB1(S172P/H173L) with LLPS ability, but not droplet-impaired SSB1(R206Q) (Extended Data Fig. 9i–k), restricted R-loops to basal levels, as shown at example SOSS–INTAC targets (Fig. 5k). Immuno-fluorescence analysis of R-loop and γH2AX signals confirmed that SSB1(S172P/H173L), but not SSB1(R206Q), can attenuate cellular R-loop levels and maintain genome stability (Extended Data Fig. 9l–o).

The relationship between R-loop levels and SSB1 mutant status and pausing was revealed by SSB1-depletion-induced Pol II changes being fully rescued by the expression of wild-type SSB1, SSB1(HY) and SSB1(S172P/H173L), but not by the SSB1 ΔIDR, SI, RY or R206Q mutants

that have an impaired condensation ability (Extended Data Fig. 10a). Increased pausing index, the ratio of Pol II occupancy at promoters to gene bodies, was observed at longer genes in dTAG-treated cells, and this was reversed by ectopic expression of wild-type SSB1, SSB1(HY) and SSB1(S172P/H173L), but not by ectopic expression of the SSB1 ΔIDR, SI, RY or R206Q mutants (Extended Data Fig. 10b).

To confirm the condensation ability of SSB1 for suppressing R-loop levels, we replaced its IDR with unrelated IDRs capable of forming liquid-like condensates. The chimeric proteins comprise the SSB1 N terminus and the C-terminal IDRs from TAF15, EWS and YTHDF1[45–47]. We induced their expression in SSB1-dTAG cells and assayed the R-loop levels (Extended Data Fig. 10c). Notably, all chimeras suppressed R-loop levels, with the IDRs of TAF15 and EWS showing the greatest R-loop-restraining activity (Fig. 5l). These results establish a causal relationship between SSB1 condensation and the attenuation of R-loop levels at SOSS–INTAC targets.

## Discussion

Here we identified a stable complex comprising the genome stability regulator SOSS and the transcription regulator INTAC. SOSS–INTAC targets active promoter and enhancer regions, relying in part on SSB1 recognition of ssDNA in the context of R-loops. SOSS–INTAC restrains aberrant accumulation of paused Pol II and prevents excessive chromatin accessibility to limit transcription-associated R-loops and maintain genome stability. SOSS–INTAC condensate formation in cells requires the SSB1 IDR, with residues mediating SOSS–INTAC condensate formation contributing to the suppression of R-loop accumulation to promote transcriptional regulation and genome stability (Fig. 5m).

Given the importance of transcription–replication conflicts for genome stability, efforts devoted to identifying transcriptional regulators involved in this process have identified known transcription initiation and elongation factors, but not transcriptional pausing regulators, despite paused polymerases being a major barrier to replication progression and contributing to genome instability[2]. Recent studies using rapid disruption of the endonuclease activity of INTAC have revealed pervasive roles of this activity in terminating paused Pol II[26,27,48]. Thus, the identification in this study of SOSS–INTAC connecting a general regulator of Pol II pausing with genome stability maintenance provides a basis for future investigations of pausing regulation in other contexts beyond transcription, such as replication and DNA damage and repair.

The N terminus of SSB1 recognizes ssDNA, whereas the conserved C-terminal IDR drives liquid-like condensate formation of SOSS–INTAC. We propose that condensation elevates the local concentration of SOSS–INTAC catalytic activity to promote promoter-proximal termination of transcription. Dysregulation of SSB1 is linked to cancer and developmental defects[34,35,49,50]. Cancer-derived mutations in SSB1 disrupting SOSS–INTAC condensation compromise its role in regulating R-loops and genome stability, which could potentially contribute to oncogenic programs. However, it is important to note that the IDR of SSB1 could possess condensation-independent functions, such that mutations disrupting condensation may also introduce additional impacts yet to be identified. Thus, future studies are warranted to systematically investigate the biophysical properties of SOSS–INTAC and their contributions to transcription, R-loop regulation and genome stability, and the degree to which the condensation ability of SOSS–INTAC contributes to these processes.

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

## Methods

### Reagents, materials and cell culture

Detailed information for reagents and materials, including antibodies and cell lines, used in this study is provided in Supplementary Table 1. Human DLD-1 cells were grown in McCoy's 5A medium (BasalMedia) supplemented with 10% fetal bovine serum (FBS, Yeasen), 1× penicillin–streptomycin (Gibco). HEK293T cells and mouse embryonic fibroblasts (MEFs) were cultured with Dulbecco's modified Eagle medium (DMEM, BasalMedia) supplemented with 10% FBS and 1× penicillin–streptomycin. HEK Expi293 cells were grown in suspension in serum-free medium. All cells were cultured at 37 °C and 5% $CO_2$ and were negative for mycoplasma contamination.

### Genome editing for CRISPR–Cas9 knockout and dTAG endogenous knock-in

*NABP1*-null single knockout cells (CTR) were generated using the CRISPR–Cas9 system from DLD-1 parental cells. In brief, the sgRNA targeting genomic regions of *NABP1* were designed using CHOPCHOP (http://chopchop.cbu.uib.no), cloned into PX458 vector and then mixed with $1 \times 10^6$ DLD-1 cells followed by electroporation (Neon). The pool of transfected cells was allowed to recover for 2 days before fluorescence-activated cell sorting of GFP-positive cells. Cells were seeded into 96-well plates by limited dilution at a density of one cell per well. After culturing for 10–14 days, cell clones were picked followed by clonal expansion. Western blotting of SSB2 was used to screen knockout clones. All oligonucleotide information for cloning and qPCR is included in Supplementary Table 2.

*NABP2*/*NABP1* DKO cells were generated by additionally deleting *NABP2* in pooled *NABP1*-null (CTR) cells. sgRNAs targeting *NABP2* exon 1 were cloned into lentiCRISPR v2 vector for lentivirus packaging. CTR cells were infected with lentivirus containing *NABP2* sgRNAs supplemented with 10 μg ml$^{-1}$ polybrene (Yeason) for 24 h. The infected cells were selected with 2 μg ml$^{-1}$ puromycin (Meilunbio) for an extra 48 h. The cells were then switched into growth medium without antibiotics and grown for an additional 24–36 h before being collected for further analysis.

The clones for the dTAG assays were performed according to previously described criteria[44]. CTR cells were used as parental cells to generate SSB1-dTAG cells. For endogenous knock-in of dTAG cassettes, CTR cells were seeded to $1 \times 10^6$ cells per well of the six-well plates the day before transfection to ensure exponential growth. The next day, cells were transfected with PITCh plasmids containing the sgRNAs targeting and cutting the genomic region of *NABP2* (PX459-sgSSB1), the dTAG repair template plasmids (pCRISPR-PITChv2-SSB1) as microhomology, and general sgRNAs (sg-PITCh) targeting the upstream of the 5′ and downstream of the 3′ ends of the microhomology region by electroporation. The cell suspension was immediately carefully transferred to 2 ml of pre-equilibrated, warm antibiotic-free DMEM in six-well plates. The cells were allowed to recover for 5 days before starting antibiotic selection of the pools in 10 ml DMEM in 10 cm dishes. Recovered cells were expanded to several 10 cm dishes by limited dilution and cultured with DMEM supplemented with 1 μg ml$^{-1}$ puromycin. After 10–14 days of selection, the surviving clones were picked and cultured in 96-well plates without antibiotics for 5–7 days. Positive clones were screened by PCR analysis of the integration site followed by verifying the protein degradation efficiency using western blotting. One working clone and up to two backup clones were selected and retained for further experiments.

### RNA interference, the generation of stable cell lines and gene-rescue experiments

To generate lentivirus for gene knockdown assays, HEK293T cells were co-transfected with shRNAs targeting genes of interest (or non-targeting shRNA as the control), psPAX2 and pMD2.G with a ratio of 3:2:1 in Opti-MEM medium using the polycation polyethylenimine (PEI) (Sigma-Aldrich) transfection reagent. The culture supernatant containing virus particles was collected at 48 h after transfection and filtered using a 0.45 μm filter. The cells were infected with lentivirus in the presence of 8 mg ml$^{-1}$ Polybrene (Sigma-Aldrich) for 24 h. The infected cells were treated with 2 mg ml$^{-1}$ puromycin for an extra 48 h before collection. The knockdown efficiency was examined qPCR with reverse transcription and western blotting.

To generate stable cell lines with the inducible overexpression of RNase H1, DLD-1 cells were initially infected with lentivirus expressing pLVX-Tet3G-rtTA and selected with G418 (Meilunbio, 500 μg ml$^{-1}$) for 2 weeks. These cells were then infected with virus expressing Flag–RNASEH1 cloned into pLVX-Tet-On vector and cultured in the presence of blasticidin (10 μg ml$^{-1}$) for an additional 2 weeks. The induction of Flag–RNASEH1 was determined by western blotting using cellular extracts from cells treated with DMSO or DOX for 24 h.

For the RNAi rescue experiments, the cells were simultaneously transduced with shRNAs targeting genes of interest (or non-target shRNAs as the control) and vectors expressing the cDNAs of corresponding genes (or empty vector as the control). At 24 h after infection, antibiotics were administered to select the cells stably expressing the resistance genes from the shRNA and overexpressing vectors for additional 2 days before further analysis. For rescue experiments in SSB1-dTAG cells, the cells were first transduced with vectors expressing wild-type or mutant *NABP2* (or empty vector as the control). At 24 h after infection, the cells were cultured under the appropriate antibiotics for an additional 2 days. The cells were then treated dTAG-13 for 12 h before further analysis. Detailed information of shRNAs, qPCR primers and cDNAs used in this study is provided in Supplementary Table 2.

### Nuclear extracts and density-gradient sedimentation

HEK Expi293 cells were collected by centrifugation and washed twice with 5 ml of ice-cold phosphate-buffered saline (PBS) and once with 2 ml of ice-cold buffer A (10 mM HEPES pH 7.4, 5 mM MgCl$_2$, 250 mM sucrose, 0.5 mM dithiothreitol (DTT), 1× protease inhibitor). The cell pellets were resuspended with 2 ml of ice-cold buffer A supplemented with 0.1% NP40 and incubated on ice for 15 min followed by centrifugation for 5 min at 4 °C and 1,000g. The nucleus fraction was collected by resuspending the pellet with buffer A (twice the volume of the original cell pellet) and centrifugation. The nuclei were next suspended with 0.75 ml of buffer B (20 mM HEPES pH 7.4, 1.5 mM MgCl$_2$, 20% glycerol, 0.5 mM EDTA, 0.5 mM DTT, 0.42 M NaCl, 1× protease inhibitor) and incubated for 30 min rotation at 4 °C. Finally, the mixture was centrifuged in the Beckman SW40 Ti rotor at 40,000 rpm for 90 min at 4 °C, and the supernatants were saved as the nuclear extract for further density-gradient sedimentation.

The HEK Expis293 nuclear extracts or purified INTAC proteins were layered on top of 4 ml of an 8–40% (v/v) glycerol gradient in buffer containing 20 mM HEPES pH 7.4, 200 mM NaCl, 0.05% CHAPS, 2 mM DTT and centrifuged at 34,000 rpm for 16 h. The samples were collected manually from the top of the gradient with each 200 μl as a fraction and analysed by western blotting.

### Co-IP assays

For co-IP assays, DLD-1 cells were collected by scraping followed by washing twice with ice-cold PBS. The cell pellet was suspended with 900 μl of ice-cold lysis buffer (20 mM Tis-HCl pH 8.0, 150 mM NaCl, 1 mM EDTA, 0.5% NP40, 10% glycerol, 1× protease inhibitor) and rotated at 4 °C for 1 h. The lysate was cleared by centrifugation for 20 min at 4 °C and 20,000g. The supernatant was incubated with 2–5 μg of antibody for each IP reaction (including IgG as negative control) followed by 9.5 h of rotation at 4 °C. Protein A/G magnetic beads (Smart Lifesciences, blocked with 1 mg ml$^{-1}$ BSA for 1 h) were added to the samples and the mixture was rotated for 3 h at 4 °C. After incubation, the samples containing the beads were collected using a magnetic rack and the beads

were washed four times with lysis buffer. Finally, the samples were collected by adding 100 µl of 1× SDS loading buffer followed by western blotting or mass spectrometry analysis.

## Protein expression and purification

Expression and purification of the INTAC protein complex was performed as described previously[4]. In brief, the full-length *INTS1* to *INTS14* open reading frames were separately cloned into a modified pCAG vector and *INTS2*, *INTS3*, *INTS4* and *INTS10* were tagged with N-terminal Flag–4×protein A. Plasmids were cotransfected into HEK Expi293 cells using PEI (Polysciences) to a final concentration of 3 mg l$^{-1}$. After being cultured at 37 °C for 72 h, cells were collected for lysis and purification. Cell pellets from 16 l of HEK Expi293 cells were resuspended and lysed in lysis buffer containing 50 mM HEPES pH 7.4, 200 mM NaCl, 0.2% CHAPS, 5 mM MgCl$_2$, 5 mM adenosine triphosphate (ATP), 10% glycerol, 2 mM DTT, 1 mM phenylmethylsulfonyl fluoride (PMSF), 1 mg ml$^{-1}$ aprotinin, 1 mg ml$^{-1}$ pepstatin and 1 mg ml$^{-1}$ leupeptin for 30 min and cleared by centrifugation for 30 min at 16,000 rpm to collect the supernatant. After incubating with immunoglobulin G (IgG) resins for overnight, the mixtures were washed with buffer containing 50 mM HEPES pH 7.4, 200 mM NaCl, 0.1% CHAPS, 10% glycerol and 2 mM DTT followed by on-column cleavage for 4 h. The immobilized proteins were then eluted out and concentrated for further purification by density-gradient sedimentation. The concentrated proteins were layered on top of a 4 ml 8–40% (v/v) glycerol gradient in buffer containing 20 mM HEPES pH 7.4, 200 mM NaCl, 0.05% CHAPS, 2 mM DTT and centrifuged at 34,000 rpm for 16 h. The fractions were collected manually from the top of the gradient for each 200 µl and analysed using a 4–12% Bis-Tris gel followed by Coomassie blue staining. Peak fractions corresponding to the INTAC complex were pooled and concentrated to 1 to 2 mg ml$^{-1}$ accompanied with the removal of glycerol.

For proteins used for the in vitro droplet assay, plasmids encoding proteins tagged with GFP–Strep were transformed and expressed in *E. coli* BL21 (DE3) cells after induction overnight with 0.25 mM IPTG at 16 °C. The cells were collected by centrifugation at 6,200$g$ for 25 min and then resuspended in 20 ml lysis buffer containing 50 mM Tris-HCl pH 7.5, 500 mM NaCl, 1 mM EDTA, 20 mM BME and 1 mM PMSF and stored at −80 °C for further protein purification.

All of the purification steps were performed at 4 °C to prevent protein degradation. After two rounds of freeze and thaw, the suspensions were lysed by sonication and centrifuged at 11,500 rpm for 1 h. The soluble fractions containing the GFP–Strep fusion proteins were loaded onto the Streptactin Beads 4FF (Smart Lifesciences) for purification. The eluted proteins were then dialysed overnight at 4 °C in 1 l dialysis buffer containing 10 mM Tris-HCl pH 7.5, 150 mM NaCl, 1 mM PMSF and 1 mM BME, and concentrated using Amicon Ultra Centrifugal Filters (Millipore). The protein concentration was measured using the Bradford Protein Quantification Kit (Vazyme) and then flash-frozen in liquid nitrogen and stored at −80 °C.

## GST pull-down assay

GST or GST–SSB1 immobilized on the glutathione-Sepharose beads were preblocked with 1% BSA and then incubated with recombinant INIP or INTAC proteins overnight at 4 °C. The next day, the beads were washed extensively with wash buffer containing 50 mM Tris-HCl pH 7.5, 100 mM NaCl, 1 mM EDTA and 0.05% NP-40 and then directly boiled in 40 µl SDS–PAGE sample-loading buffer. The samples were analysed by Coomassie Blue staining and western blotting.

## EMSA

The purified SSB1 and INTAC alone or mixed as indicated were incubated with 100 nM Cy3-labelled ssDNA, dsDNA or ssRNA on ice for 30 min in binding buffer containing 20 mM Tris-HCl pH 7.5, 50 mM NaCl, 5 mM MgCl$_2$, 0.2 mM EDTA and 1 mM DTT. The DNA–protein complexes were loaded onto a 6% native polyacrylamide gel in 0.5× TBE buffer and run

for 30 min at 150 V in a cold room. After electrophoresis, the gels were scanned using the RGB channel of an Azure C400 instrument.

## ChIP–Rx and ChIP–qPCR

The ChIP–Rx experiments were performed as described previously[53]. In brief, for each IP, $1 × 10^7$ cells were cross-linked with 1% formaldehyde at room temperature for 10 min and consequently quenched with 125 mM glycine for 5 min at room temperature. Cells were scraped and centrifuged with 1,000$g$ for 10 min. The cell pellets were washed twice with ice-cold PBS and resuspended in lysis buffer containing 50 mM HEPES pH 7.4, 150 mM NaCl, 2 mM EDTA, 0.1% Na-deoxycholate, 0.1% SDS, 1× protease inhibitor, 1× phosphatase inhibitor, followed by sonicating (Qsonica) to appropriate fragment (200–700 bp). After sonication, the lysate was centrifuged at maximal speed for 15 min to collect the supernatant and mixed with 20% of lysate from MEFs processed identically as spike-in for normalization.

The chromatin samples were incubated with specific antibodies overnight at 4 °C. After incubation, the protein–DNA complex was immobilized on pre-blocked (BSA, 2 mg ml$^{-1}$ for 2 h) magnetic protein A/G beads for 3 h at 4 °C. Immobilized, the bound fractions were washed three times with high-salt wash buffer (20 mM HEPES pH 7.4, 500 mM NaCl, 1 mM EDTA, 1.0% NP40, 0.25% Na-deoxycholate, 1× protease inhibitor, 1× phosphatase inhibitor), twice with low-salt wash buffer (20 mM HEPES pH 7.4, 150 mM NaCl, 1 mM EDTA, 0.5% NP40, 0.1% Na-deoxycholate, 1× protease inhibitor, 1× phosphatase inhibitor) and once with Tris-EDTA (TE) buffer supplemented with 50 mM NaCl. Elution and re-cross-linking were performed in elution buffer (50 mM Tris-HCl pH 8.0, 10 mM EDTA, 1% SDS) supplemented with protease K at 65 °C for overnight. The DNA samples were purified using the phenol–chloroform DNA extraction method. The precipitated DNA sample was either analysed by qPCR or subjected to library preparation using the VAHTS Universal Plus DNA Library Prep Kit for Illumina (Vazyme). The library was then sequenced using the NovaSeq 6000 platform (Mingma Technologies).

## PRO–seq

PRO–seq library preparation was performed as previously described[54,55], and all of the procedures below were carried out on ice. In brief, the cells cultured in 15 cm dishes were collected by washing twice with 5 ml ice-cold PBS and scraping with 5 ml permeabilization buffer (10 mM Tris-HCl pH 8.0, 5% glycerol, 250 mM sucrose, 10 mM KCl, 5 mM MgCl$_2$, 1 mM EGTA, 0.5 mM DTT, 0.1% NP40, 0.05% Tween-20, 1× protease inhibitors (Roche), 4 U ml$^{-1}$ RNase inhibitor (SUPERaseIN)), followed by incubating on ice for up to 5 min. Permeabilized cells were collected by centrifugation (800$g$, 4 min, 4 °C) and washed twice with ice-cold cell wash buffer (10 mM Tris-HCl pH 8.0, 5% glycerol, 10 mM KCl, 5 mM MgCl$_2$, 0.5 mM DTT, 4 U ml$^{-1}$ RNase inhibitor). Washed nuclei were resuspended in freezing buffer (50 mM Tris-HCl pH 8.0, 40% glycerol, 5 mM MgCl$_2$, 1 mM EDTA, 0.5 mM DTT, 4 U ml$^{-1}$ RNase inhibitor) at a density of $3 × 10^6$ cells per 50 µl and immediately frozen in liquid nitrogen. Cells were stored in −80 °C until use.

A total of 3 million permeabilized cells (mixed with $3 × 10^5$ MEFs as a spike-in) were added to the same volume of 2× nuclear run-on mixture (10 mM Tris-HCl pH 8.0, 300 mM KCl, 1% Sarkosyl (Sigma-Aldrich), 5 mM MgCl$_2$, 1 mM DTT, 40 mM Biotin-11-C/GTP (Perkin Elmer), 0.8 U ml$^{-1}$ RNase inhibitor) and incubated at 30 °C for 5 min. Nascent RNA was extracted using TRIzol LS (Ambion) followed by ethanol precipitation. Extracted RNA was fragmented by base hydrolysis in 0.25 N NaOH for 10 min on ice and immediately neutralized with 1× volume of 1 M Tris-HCl pH 6.8, followed by passing through a calibrated RNase-free P30 column (Bio-Rad, 732-6251). Fragmented RNA was dissolved in H$_2$O and incubated with 10 pmol of reverse 3′ RNA adapter and treated with T4 RNA ligase (NEB) for 1 h at 25 °C. After 3′ RNA ligation, fragmented nascent RNA was bound to 25 µl of prewashed Streptavidin Magnetic Beads (NEB) in binding buffer (10 mM Tris-HCl pH 7.4, 300 mM NaCl,

0.1% Triton X-100, 1 mM EDTA) for 20 min at 25 °C. The bound beads were washed once with high-salt wash buffer (50 mM Tris-HCl pH 7.4, 2 M NaCl, 0.5% Triton X-100, 1 mM EDTA) and once with low-salt wash buffer (5 mM Tris-HCl pH 7.4, 0.1% Triton X-100, 1 mM EDTA). The on-bead reaction of RNA 5′ hydroxyl repair was performed in PNK mix (1× PNK buffer, 1 mM ATP, 10 U PNK (NEB)) at 37 °C for 30 min. For nascent RNA 5′ de-capping, the RNA products were incubated with RppH mix (1× ThermoPol buffer, 5 U RppH (NEB)) for 1 h at 37 °C. The RNA 5′ adapter ligation was performed using the ligation mix (1× T4 RNA ligase buffer, 1 mM ATP, 15% PEG8000, 10 U T4 RNA ligase) at 25 °C for 1 h. Adapter-ligated nascent RNA was enriched with biotin labelled products by another round of Streptavidin bead binding, once with high-salt wash buffer and once with low-salt wash buffer, followed by TRIzol extraction of the RNA product. The air-dried RNA pellet was resuspended in RT resuspension mix (3 μM RP1, 0.74 mM dNTP mix) and denatured at 65 °C for 5 min and snap-cooled on ice, followed by the addition of 6.5 μl of RT master mix (3× RT buffer, 15.4 mM DTT, 10 U RNase inhibitor) to each sample. Reverse transcription was performed using the 200 U superscript III enzyme (Invitrogen). The reverse-transcription products immediately underwent PreCR treatment, test amplification and full-scale library amplification using the Q5 DNA polymerase (NEB). The libraries were then sequenced using the NovaSeq 6000 platform (Mingma Technologies).

## R-loop CUT&Tag

R-loop CUT&Tag was optimized according to a previously published protocol[8,56]. DLD-1 cells were collected by Accutase (Thermo Fisher Scientific) to avoid overdigestion. For a single R-loop CUT&Tag, half a million cells were typically used to obtain sufficient DNA extraction for library construction. The cells were centrifuged (600g, 3 min) at room temperature, washed twice with 800 μl of wash buffer (20 mM HEPES pH 7.5, 150 mM NaCl, 0.5 mM spermidine, 1× protease inhibitor) and finally resuspended with 100 μl of wash buffer in low-retention PCR tubes. The concanavalin-A-coated magnetic beads (Smart-Lifesciences) were activated in advance and resuspended with the same volume of the binding buffer (20 mM HEPES pH 7.5, 10 mM KCl, 1 mM CaCl$_2$, 1 mM MnCl$_2$). A total of 10 μl of activated concanavalin A beads was added to $5 \times 10^5$ cells with incubation for 10 min under gentle rotation. The bead-bound cells were magnetized to remove the liquid with a pipettor and resuspended in 50 μl of antibody buffer (20 mM HEPES pH 7.5, 150 mM NaCl, 0.5 mM spermidine, 1× protease inhibitor, 0.05% digitonin, 0.01% NP-40, 2 mM EDTA). Next, 1 μg of S9.6 (Active Motif) was added to combine the DNA−RNA hybrid by rotating at 4 °C overnight. A total of 10 μg of RNase H1 (Thermo Fisher Scientific) was added with S9.6 to cleave the DNA−RNA hybrid as a negative control. For the IgG control, mouse IgG was used instead. After successive incubation with rabbit anti-mouse IgG (Solarbio, 1:100 dilution) and mouse anti-rabbit IgG (Solarbio, 1:100 dilution) in 100 μl of antibody buffer for 1 h at room temperature, the bead-bound cells were washed three times with dig-wash buffer (antibody buffer without 2 mM EDTA) to remove the unbound antibody.

The pAG-Tn5 adapter complex was mixed in dig-300 buffer (20 mM HEPES pH 7.5, 300 mM NaCl, 0.5 mM spermidine, 1× protease inhibitors, 0.01% digitonin, 0.01% NP-40) to a final concentration of 0.2 μM. The bead-bound cells were resuspended in 100 μl of pAG-Tn5 mix and incubated at room temperature for 1 h followed by removing the supernatant. After adequate washing, the tagmentation reaction was performed in 40 μl of tagmentation buffer (10 mM TAPS-KOH pH 8.3, 10 mM MgCl$_2$, 1% DMF) at 37 °C for 1 h. Next, 1.5 μl of 0.5 M EDTA, 0.5 μl of 10% SDS and 1 μl of 20 mg ml$^{-1}$ protease K were added to stop the reaction. After incubation for 1 h at 55 °C, DNA purification was performed using VAHTS DNA Clean Beads (Vazyme), and eluted in 10 μl of 0.1% Tween-20. The eluent was mixed with 10 U of Bst 2.0 WarmStart DNA polymerase (NEB) and 1 × Q5 polymerase reaction buffer (NEB) in a 20 μl reaction system. The reaction was completed at 65 °C for 30 min

and then at 80 °C for 20 min to inactivate the Bst 2.0 WarmStart DNA polymerase. The purified DNA was amplified by Q5 high-fidelity DNA polymerase (NEB) with a universal i5 primer and a uniquely barcoded i7 primer. The exact PCR cycles were estimated by qPCR before amplification. PCR amplification with 13−14 cycles yielded enough quantity of library for sequencing. After library size-selection with 0.56−0.85 VAHTS DNA Clean Beads, with library sizes ranging from 200 to 700 bp, the products were next either analysed using qPCR or sequenced on the NovaSeq 6000 platform (Mingma Technologies).

## KAS−seq

KAS−seq was performed as described previously with minor modifications[57]. A total of 1 million DLD-1 cells was labelled with 2.5 mM N$_3$-kethoxal for 10 min at 37 °C. The gDNA was isolated using the PureLink genomic DNA mini kit (Thermo Fisher Scientific). The extracted gDNA was biotinylated with 1 mM DBCO-PEG$_4$-biotin (Sigma-Aldrich) through a click cycloaddition reaction. After sonication, the biotinylated gDNA was fragmented into sizes of ~300 bp before mixing the fragments with 10 μl of Dynabeads Myone Streptavidin C1 beads (Thermo Fisher Scientific). After incubation and brief washes, the beads were resuspended in nuclease-free water at 95 °C for 15 min to facilitate the dissolution of N$_3$-kethoxal-modified gDNA fragments. Next, the DNA fragments were repaired with the phi29 DNA polymerase (NEB) and purified using VAHTS DNA Clean Beads. Library preparation was performed using the VAHTS Universal Plus DNA Library Prep Kit for Illumina (Vazymes). The library was then sequenced on the NovaSeq 6000 platform.

## Immunofluorescence analysis

DLD-1 cells were seeded on coverslips at least 24 h before the experiment. After washing with PBS, cells were incubated with 4% paraformaldehyde (PFA) for 10 min. After washing three times with PBS, cells were permeabilized with 0.5% Triton X-100 in PBS for 10 min and blocked with 4% BSA in PBS for 30 min. Primary antibodies were dissolved in ice-cold 4% BSA with the dilution ratio recommended by producers, and the cells were then immersed in the primary antibody buffer for overnight incubation at 4 °C. After three washes in PBS, cells were incubated with the appropriate secondary antibodies for 1 h. Next, cells were mounted in ProLong Gold Antifade Mountant with DAPI (Invitrogen) before imaging. For rapid R-loop immunofluorescence, GFP−RNASEH1 was used as the primary sensor, and the protein was purified as previously described[39]. Cells were incubated with 2 μg of GFP−dRNASEH1 in 4% BSA overnight at 4 °C. After washing three times with PBS, cells were directly mounted before imaging. The presented images were obtained using the Leica TCS SP8 laser-scanning confocal microscopy. Unless otherwise indicated, all procedures were performed at room temperature.

## γH2AX FACS assay

Single-cell suspensions of CTR and DKO cells were incubated with 70% ethanol at −20 °C for 2 h. After two washes with PBS, cells were fixed with 4% PFA for 15 min. Next, cells were permeabilized with 0.25% Triton X-100 in PBS for 15 min and blocked with 2% BSA in PBS for 30 min. For intracellular γH2AX staining, $1 \times 10^6$ cells were incubated with 1 μg γH2AX antibodies (Thermo Fisher Scientific) overnight at 4 °C, followed by incubation with Alexa-Fluor-488-conjugated secondary antibodies for 30 min at room temperature. After washing three times with PBS, cells were treated with propidium iodide staining buffer (Sangon Biotech) according to the manufacturer's protocol. Data were acquired using FACSDiva Flow Cytometry Software (BD Biosciences) and analysed using FlowJo (TreeStar).

## OptoDroplet assay

Hela cells expressing SSB1−mCherry−CRY2 or empty mCherry−CRY2 vector were imaged using two laser wavelengths (488 nm for mCry2

activation and 560 nm for mCherry imaging). To examine droplet formation, mCherry-positive cells were subjected to repetitive on/off cycles, whereby they were first exposed under a 488 nm laser for 1 s, and then an image was captured for the mCherry signal.

### DNA fibre assay

DLD-1 cells were sequentially labelled with 10 mM IdU (Sigma-Aldrich) and 100 mM CldU (Sigma-Aldrich) for 30 min each. After labelling, cells were placed on ice immediately to stop DNA replication and subsequently centrifuged (300g, 5 min at 4 °C). After washing three times in PBS, $1 \times 10^6$ cells were placed onto a microscope slide and incubated with the spreading buffer (200 mM Tris-HCl pH 7.5, 0.5% SDS and 50 mM EDTA) for 1 min. The slides were tilted 15° to extend the DNA fibres. After fixation using methanol/acetic acid (3:1), the DNA was denatured using 2.5 M HCl and blocked with 1% BSA for 2 h before staining with primary (rat anti-BrdU for CldU and mouse anti-IdU) and secondary antibodies conjugated with Alexa Fluor 488 or 546. Images were acquired using a confocal microscope (Lecia TCS SP8) and analysed using the ZEN 2.3 SP1 (ZEISS) software. Statistical analysis was performed using Prism 8 (GraphPad software).

### Analyses for protein disorder and amino acid sequence features

Disordered regions were identified using IUPred and IUPred3 (http://iupred.elte.hu/). Amino acid composition was analysed using Composition Profiler (http://www.cprofiler.org/cgi-bin/profiler.cgi). The net charge per residue was analysed using CIDER 40 (http://pappulab.wustl.edu/CIDER/analysis/).

### In vitro droplet assay

Recombinant proteins were diluted to the indicated salt concentrations with buffer containing 10 mM Tris-HCl pH 7.5 to induce phase separation. A total of 8 µl of phase-separation solution was loaded onto a glass slide, covered with a coverslip and images were acquired using the Zeiss LSM880 microscope. For identifying droplet fusion events, glass slides loaded with protein solutions were inverted on the microscope lens, and images were acquired at 1 s intervals and further analysed using ImageJ. For FRAP assays, droplets containing fluorescent proteins were bleached with the desired laser intensity and 100 post-bleach frames were recorded with a time interval of 1 s. The fluorescence intensity at bleached region was corrected with an unbleached region and normalized to the pre-bleaching fluorescence intensity. For the co-phase separation assay of wild-type or mutant GFP–SSB1 with INTAC, INTAC was labelled using the Alexa Fluor 568 protein labelling kit (Thermo Fisher scientific) according to manufacturer's protocols. The labelled INTAC proteins were diluted with unlabelled ones to a desired concentration and then mixed with GFP fusion proteins to induce phase separation.

### Quantification and statistical analysis

**ChIP–Rx analysis.** Raw ChIP–Rx reads were trimmed using Trim Galore v.0.6.6 (Babraham Institute) in paired-end mode. Trimmed reads were aligned to human hg19 and mouse mm10 genome assemblies using Bowtie (v.2.4.4)[58] with the default parameters. All unmapped reads, low mapping quality reads (MAPQ < 30) and PCR duplicates were removed using SAMtools (v.1.12)[59] and the MarkDuplicates function of Picard Tools v.2.25.5 (Broad Institute). Peaks were called using MACS2 (v.2.2.7.1)[60] with the option 'nomodel' and peak annotation was performed with R package ChIPseeker (v.1.28.3)[61].

For quantitative comparison, read counts were normalized to the corresponding total reads aligned to spike-in genome in previous ChIP–Rx studies[53,62]. However, the number of reads mapped to spike-in genome could be influenced by the actual mixing ratio of chromatin samples before IP, which should also be scaled. To better compare the ChIP–Rx datasets, we derived a new scale factor $\alpha$ for each IP experiment as described in Supplementary Note 1.

Normalized bigwig files were generated with the bamCoverage function from deepTools (v.3.5.1)[63] using scale factors calculated according to Supplementary Note 1. Reads mapping to the ENCODE blacklist regions[64] were removed using bedTools (v.2.30.0)[65]. Heat maps (10 bp per bin) and metagene plots were generated using the computeMatrix function followed by the plotHeatmap and plotProfile functions of deepTools (v.3.5.1)[63]. Spike-in normalized occupancy at per promoter (1 kb upstream and 1 kb downstream of the TSS) was calculated using getCountsByRegions function from R package BRGenomics[66], which can get the sum of the signal in normalized bigwig that overlaps defined regions. Pearson correlations of ChIP–Rx samples were calculated using deepTools (v.3.5.1)[63] (multiBamSummary followed with plotCorrelation) with the read counts split into 10 kb bins across the genome. The pausing index was defined as the ratio of Pol II occupancy at promoters (from 100 bp upstream to 300 bp downstream of the TSS) to Pol II occupancy over gene bodies (from 300 bp to 2 kb downstream of the TSS). Pol II occupancy was also calculated using getCountsByRegions function from R package BRGenomics.

**KAS–seq analysis.** Raw reads of KAS–seq were trimmed as described for ChIP–Rx above. Trimmed reads were aligned to the human hg19 and mouse mm10 genomes using Bowtie (v.2.4.4)[58] with the option '-X 1000'. Removal of low mapping quality reads and duplicated reads, peak calling and annotation were performed in the same manner as described for ChIP–Rx. The scale factor for normalizing ssDNA signals was calculated as 1 over the number of reads mapping to spike-in genome (mm10) per million as previously described. Normalized bigwig files were generated using the bamCoverage function from deepTools (v.3.5.1)[63] and reads mapping to the ENCODE blacklist regions[64] were removed using bedTools (v.2.30.0)[65].

**PRO–seq analysis.** Raw PRO–seq reads were processed as described for ChIP–Rx above, with reads longer than 15 bp retained. Ribosomal RNA reads were removed using Bowtie (v.2.4.4)[58] with '--un-conc-gz'. The remaining reads were aligned to human hg19 and mouse mm10 genome assemblies using Bowtie (v.2.4.4)[58] with the parameters '--local --very-sensitive-local --no-unal --no-mixed --no-discordant'. Removal of low mapping quality reads and duplicated reads and calculation of scale factor were performed in the same manner as described for KAS–seq. Single-base-pair resolution, normalized, stranded read coverage tracks were generated using the bamCoverage function of deepTools (v.3.5.1)[63] with the parameters '--Offset 1 --samFlagInclude 82' and '--Offset 1 --samFlagInclude 98' for the forward and reverse strand, respectively. TSSs of sense and antisense transcription were determined using published PRO–Cap data of DLD-1 cells and according to a previously published protocol[67].

**ATAC–seq analysis.** After trimming the adapters and low-quality reads as described for ChIP–Rx above, the remaining reads were aligned to human hg19 using Bowtie (v.2.4.4)[58] with the parameters '-N 1 -L 25 -X 2000 --no-mixed --no-discordant'. For spike-in normalization, the reads were also aligned to the *E. coli* genome by Bowtie (v.2.4.4)[58] with the options '--end-to-end --very-sensitive --no-overlap --no-dovetail --no-mixed --no-discordant -I 10 -X 700'. Mitochondrial reads and PCR duplicates were then filtered using SAMtools (v.1.12)[59] and Picard Tools (v.2.25.5; Broad Institute). Finally, the reads were shifted to compensate for the offset in tagmentation site relative to the Tn5 binding site using the alignmentSieve function of deepTools (v.3.5.1)[63] with the '--ATACshift' option. Read counts were adjusted to total reads aligned to *E. coli* genome using deepTools (v.3.5.1)[63].

**CUT&Tag analysis.** Adapters and low-quality reads were trimmed as described for ChIP–Rx above and the resulting reads were aligned to human hg19 genome using Bowtie (v.2.4.4)[58] with the default parameters. For quantitative comparison, the reads were also aligned to the

*E. coli* genome using Bowtie (v.2.4.4)[58] with the options '--end-to-end --very-sensitive --no-overlap --no-dovetail --no-mixed --no-discordant -I10 -X 700'. Duplicated reads were removed with Picard Tools (v.2.25.5; Broad Institute) and the reads were shifted as described for ATAC–seq. Read counts adjusted to total reads were aligned to *E. coli* genome using deepTools (v.3.5.1)[63].

**Statistics and reproducibility.** Wilcoxon rank-sum tests were used throughout this study unless otherwise specified. Unless otherwise indicated, each experiment was performed with three independent replicates.

## Reporting summary

Further information on research design is available in the Nature Portfolio Reporting Summary linked to this article.

## Data availability

The high-throughput sequencing data, including ChIP–Rx, KAS–seq, PRO–seq and CUT&Tag, have been deposited at the Gene Expression Omnibus under accession number GSE223997. Expression of *NABP2* and *NABP1* across tissues was analysed by GTEx (https://gtexportal.org/home/). *NABP2* mutations in human cancer were analysed by COSMIC (https://cancer.sanger.ac.uk/cosmic).

## Code availability

The scripts used to analyse the data from this study are freely available at GitHub (https://github.com/chenjiwei124128/SSB1_NGS_analysis).

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

**Acknowledgements** We thank C. Liu for help with imaging; S. Hu for CRISPR knock-ins; T. Wang for DNA fibre assays; P. Zhang for flow cytometry; and Y. Yu for sharing IDR plasmids. This work was supported by grants from the National Key R&D Program of China (2021YFA1301700, 2021YFA1300100), the National Natural Science Foundation of China (32070636,82003086, 92053114 and 32070632) and the Shanghai Natural Science Foundation (20ZR1412100 and 22ZR1412400).

**Author contributions** C.X. and Y.X. performed most of the cell-based and biochemistry experiments with help from Z.Q., P.F., S.M., J.L. and B.T. Chengyu Li performed the in vitro droplet formation, EMSA and OptoDroplet assays. J. Chen and A.S. analysed the sequencing data. Conghui Li conducted the KAS-seq. Y.-J.L., K.L., J.W., Z.Z., X.Y., H.Z., J. Cheng, R.X., Q.W., P.Z., H.G., D.Y., P.W., J.X., Y.C, W.X. and T.X. contributed intellectual input. F.X.C., H.L. and C.X. conceptualized the project, designed the experiments and wrote the manuscript.

**Competing interests** The authors declare no competing interests.

**Additional information**
**Correspondence and requests for materials** should be addressed to Huasong Lu or Fei Xavier Chen.

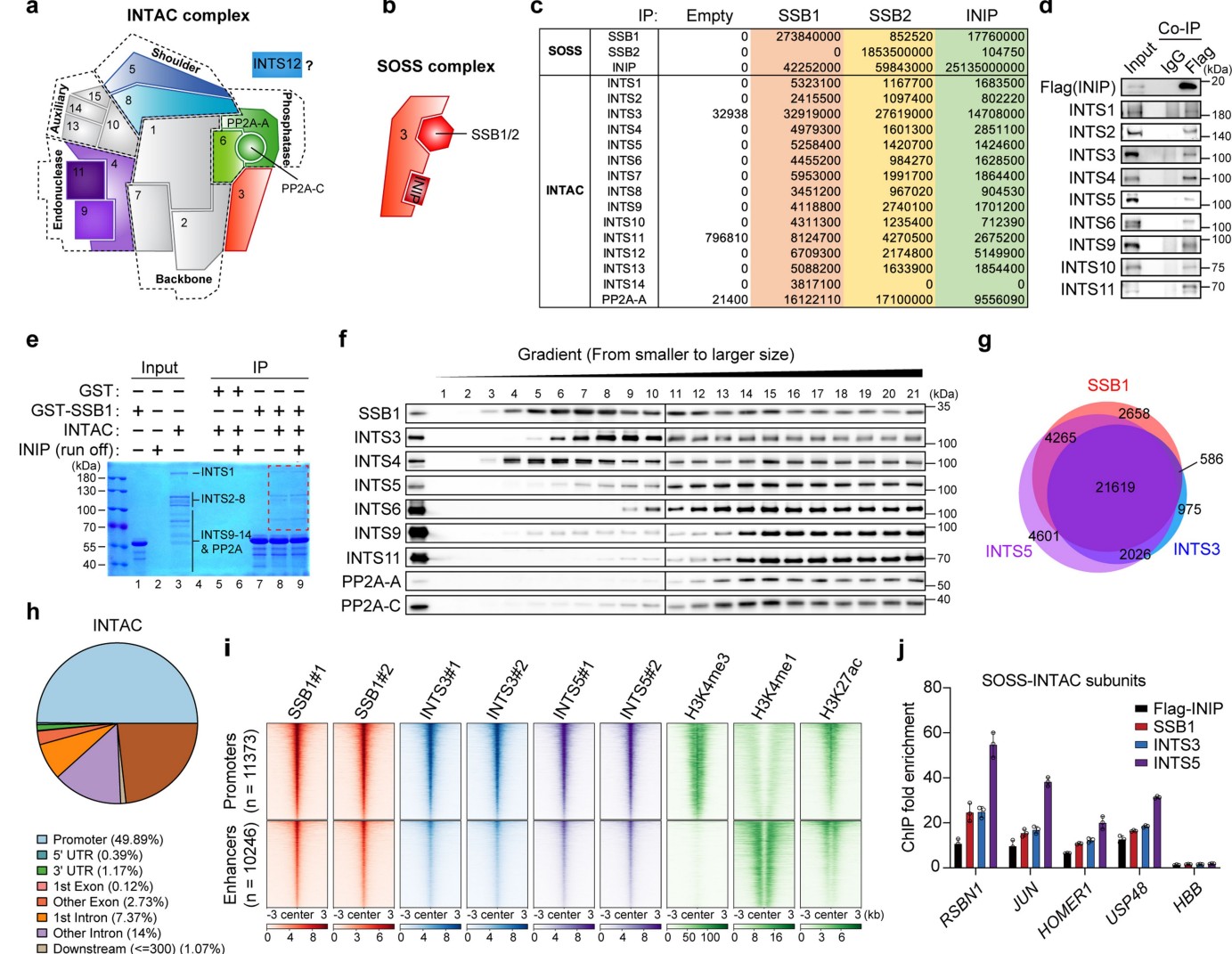

**Extended Data Fig. 1 | Biochemical and genomic analyses of the SOSS–INTAC complex.** (a-b) Schematic of the INTAC (a) and SOSS (b) complexes. (c) Mass spectrometry analyses of Protein A-tagged SSB1, SSB2 and INIP immunoprecipitation (IP) in DLD-1 cells. The values are intensity-based absolute quantification intensity for SOSS and INTAC subunits. (d) Flag IP in cells with overexpression of Flag-tagged INIP followed by western blotting in DLD-1 cells. IgG was used as the binding control. Data represent two independent experiments. (e) Immobilized GST or GST–SSB1 were incubated with purified INTAC in the presence or absence of INIP. The input and bound proteins were analysed by Coomassie blue staining. (f) Gradient centrifugation using nuclear extracts of HEK Expi293 cells with overexpression of SSB1 and all

INTAC subunits. The fractionated samples were examined by SDS–PAGE followed by western blotting. Data represent two independent experiments. (g) Venn diagram showing the overlapping binding regions of INTS3 (blue), INTS5 (purple) and SSB1 (red) peaks in DLD-1 cells. (h) Genomic distribution of INTAC alone. (i) Heatmaps of occupancy of SSB1, INTS3, INTS5, H3K4me3, H3K4me1 and H3K27ac over 6 kb regions centred on the SOSS–INTAC peak summits divided into promoter and enhancer regions. (j) ChIP–qPCR experiments using SSB1 (red), INTS3 (blue) and INTS5 (purple) antibodies in DLD-1 cells. Due to the lack of a suitable INIP antibody for IP, Flag ChIP–qPCR was conducted in DLD-1 cells with overexpression of Flag-tagged INIP. n = 3 biological replicates.

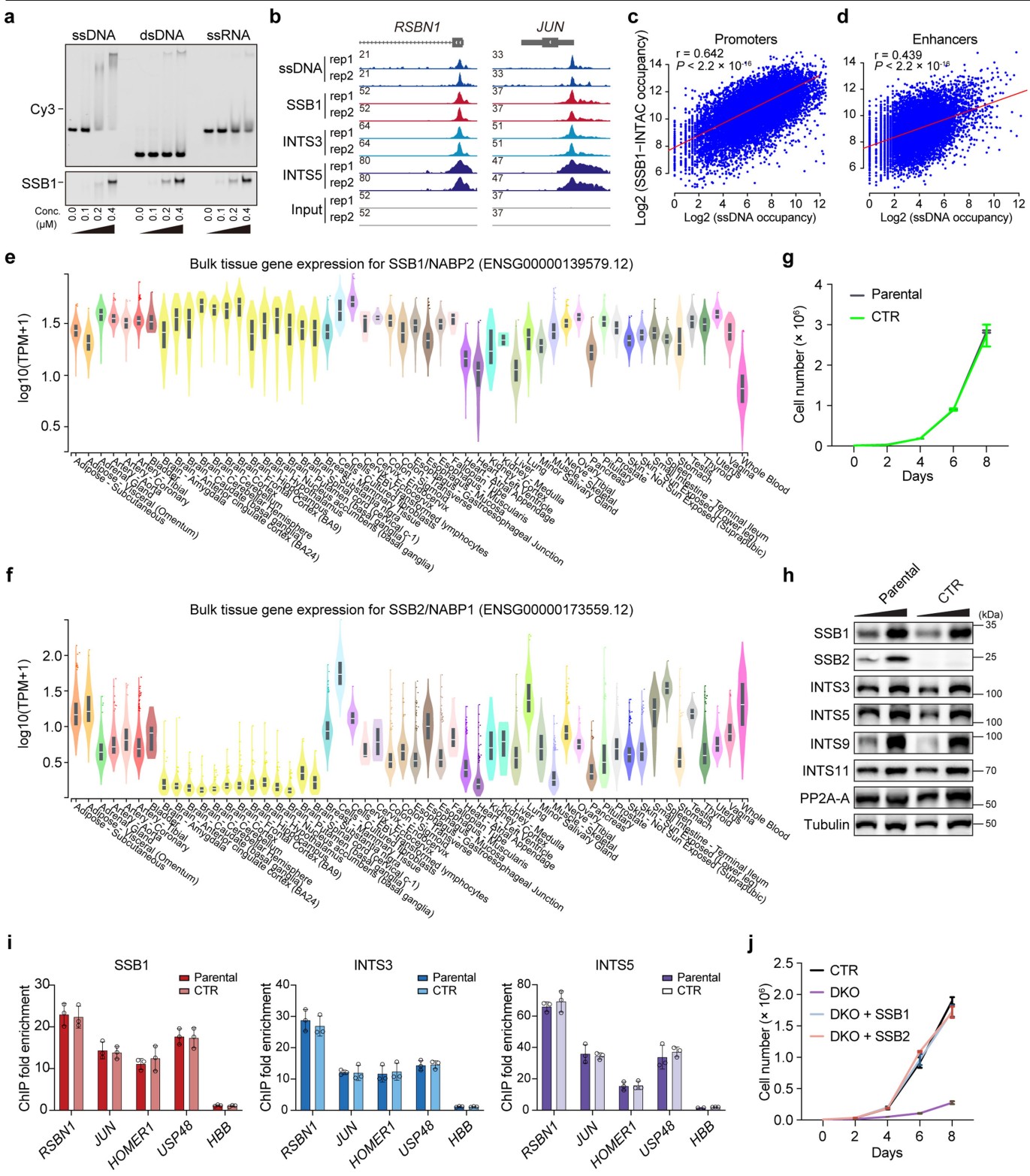

**Extended Data Fig. 2 | SsDNA binding, expression pattern and functional redundancy of SSB1 and SSB2.** (a) EMSA assays using Cy3-labelled ssDNA, dsDNA and ssRNA incubated with SSB1. Data represent two independent experiments. (b) Representative browser tracks showing KAS−seq signals compared with the genomic occupancy of SOSS−INTAC subunits in DLD-1 cells. (c-d) Correlation between ssDNA levels and SSB1 occupancy over SOSS−INTAC-bound promoters (c, P < 2.2e-16, n = 11,373 peaks) and enhancers (d, P < 2.2e-16, n = 10,246 peaks). P values were computed using two-sided t-test with 95% confidence interval based on Pearson's product moment correlation coefficient. Data represent two independent experiments. (e-f) The expression of SSB1 (e)

and SSB2 (f) across tissues using GTEx database. (g) Growth curves of parental and CTR DLD-1 cells. Data are mean ± SD from 4 independent experiments. (h) Western blotting of whole-cell extracts from parental and CTR DLD-1 cells. Tubulin is a loading control. Data represent two independent experiments. (i) ChIP−qPCR experiments using SSB1 (red), INTS3 (blue) and INTS5 (purple) antibodies in parental and CTR DLD-1 cells. Values are mean ± SD (n = 3 biological replicates). (j) Growth curves of CTR and DKO cells with or without overexpression of SSB1 or SSB2. Data are mean ± SD from 4 independent experiments.

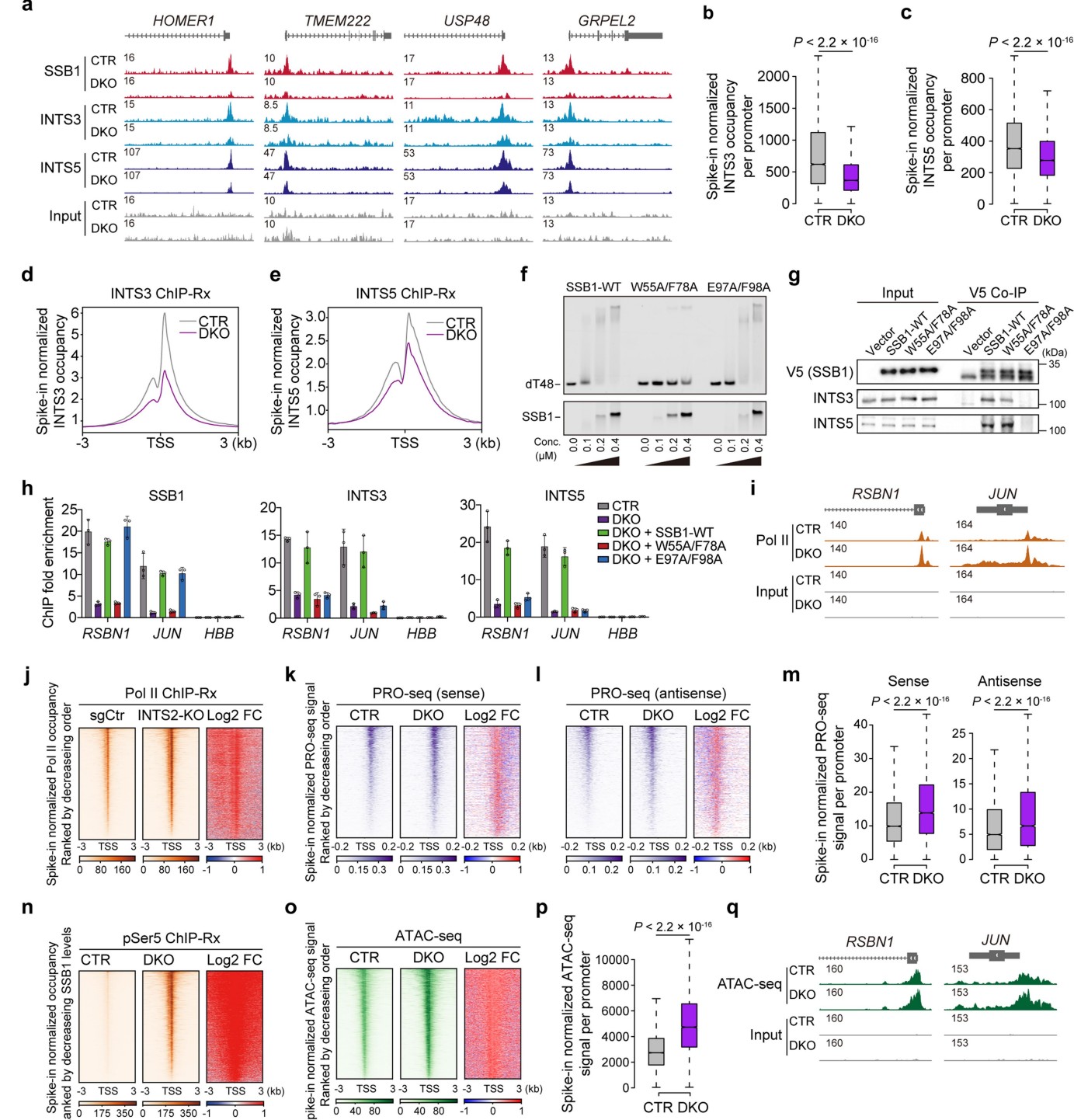

**Extended Data Fig. 3** | See next page for caption.

**Extended Data Fig. 3 | SSB1 facilitates SOSS–INTAC recruitment to induce promoter-proximal termination.** (a) Representative browser tracks showing the ChIP–Rx signals of SSB1 (red), INTS3 (blue) and INTS5 (purple) in CTR and DKO cells. (b-c) Boxplots showing the comparison of INTS3 (b) and INTS5 (c) signals at SOSS–INTAC target promoters between CTR and DKO cells. In boxplots, the centre line is the median, the top and bottom hinges correspond to the first and third quartiles, respectively, whiskers extend to quartiles ± 1.5 × interquartile range. *P* values were calculated using two-sided Wilcoxon rank-sum tests. *P* < 2.2e-16, n = 10,650 promoters. (d-e) Metaplots of INTS3 (d) and INTS5 (e) signals over 6 kb regions centred on TSS of SOSS-INTAC target genes in CTR and DKO cells. (f) EMSA assays using Cy3-labelled oligo (dT)48 incubated with purified wild-type SSB1, W55A/F78A (the mutant defective in binding ssDNA), or E97A/F98A (the mutant defective in interacting with INTS3). Data represent two independent experiments. (g) V5 Co-IP in cells overexpressed with V5-tagged wild-type SSB1, W55A/F78A, or E97A/F98A. Data represent two independent experiments. (h) ChIP–qPCR of SSB1, INTS3 and INTS5 in CTR and DKO cells with overexpression of wild-type SSB1, W55A/F78A, or E97A/F98A. Values are mean ± SD. n = 3 biological replicates. (i) Representative browser tracks showing the ChIP–Rx signals of Pol II in CTR and DKO cells. (j) Heatmaps of Pol II ChIP–Rx signals on SOSS–INTAC target genes in DLD-1 cells with control sgRNA (sgCtr) and sgRNA targeting INTS2 (INTS2-KO). The peaks are centred on TSS and ranked by decreasing occupancy in sgCtr cells. (k-l) Heatmaps of PRO–seq signals for sense (k) and antisense (l) transcripts over 400 bp regions centred on TSS of SOSS–INTAC target genes ranked by decreasing occupancy in CTR cells. (m) Boxplots showing the comparison of sense and antisense transcription levels at SOSS–INTAC-bound promoters between CTR and DKO cells. In boxplots, the centre line is the median, the top and bottom hinges correspond to the first and third quartiles, respectively, whiskers extend to quartiles ± 1.5 × interquartile range. *P* values were calculated using two-sided Wilcoxon rank-sum tests. *P* < 2.2e-16, n = 6,860 promoters for sense transcription and *P* < 2.2e-16, n = 5,767 promoters for antisense transcription. (n) Heatmaps showing the occupancy of Pol II phosphorylated at CTD Serine 5 (pSer5) on SOSS–INTAC target genes in CTR and DKO cells. The peaks are centred on TSS and ranked by decreasing occupancy in CTR cells. (o) Heatmaps of ATAC–seq signals on SOSS–INTAC target genes in CTR and DKO cells. The peaks are centred on TSS and ranked by decreasing occupancy in CTR cells. (p) Boxplots showing the comparison of ATAC–seq signals at SOSS–INTAC target promoters between CTR and DKO cells. In boxplots, the centre line is the median, the top and bottom hinges correspond to the first and third quartiles, respectively, whiskers extend to quartiles ± 1.5 × interquartile range. *P* values were calculated using two-sided Wilcoxon rank-sum tests. *P* < 2.2e-16, n = 10,650 promoters. (q) Representative browser tracks showing the ATAC–seq signals in CTR and DKO cells.

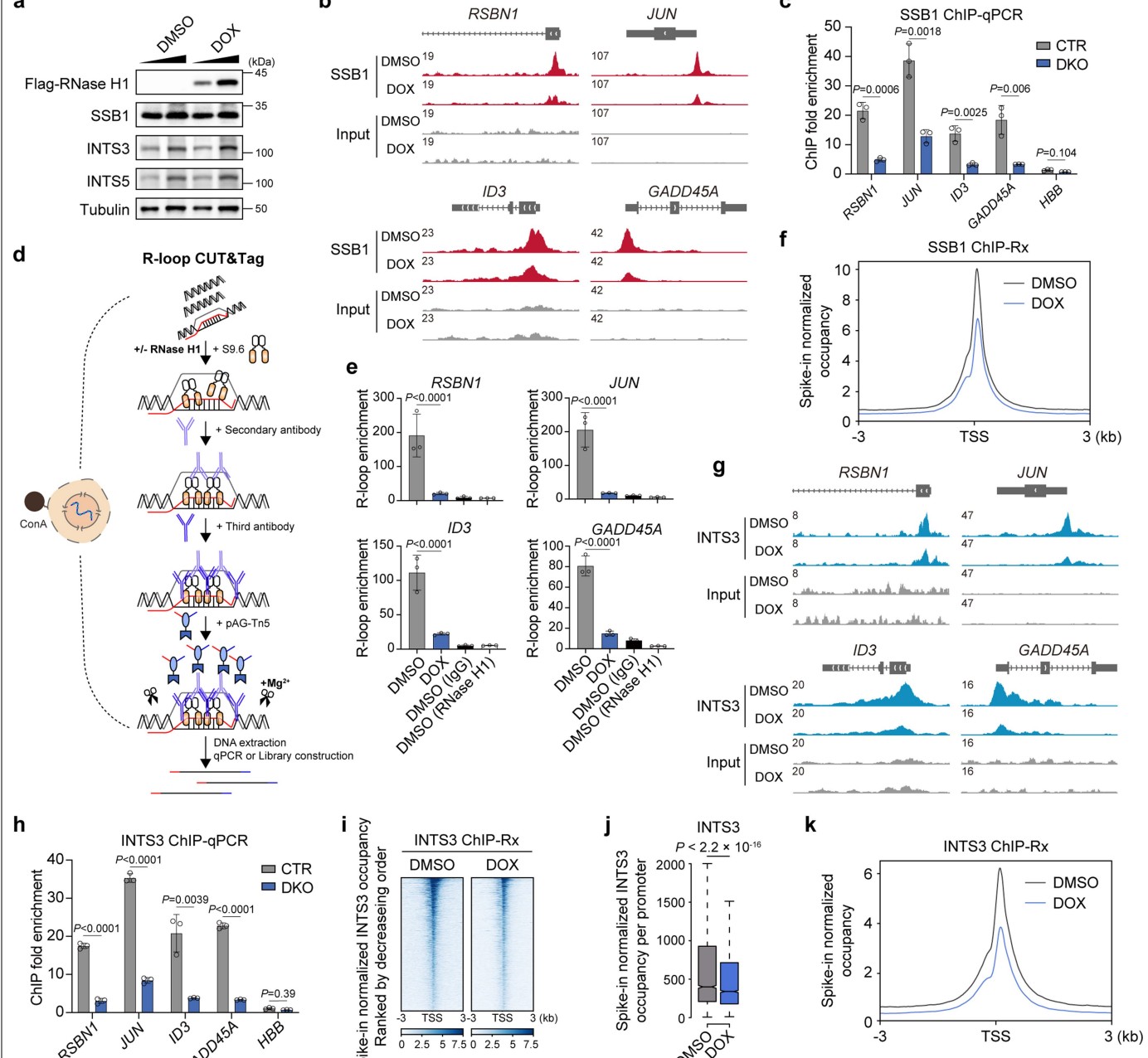

**Extended Data Fig. 4 | SOSS–INTAC recognizes R-loops.** (a) Western blotting of whole-cell extracts from DOX-inducible Flag-RNase H1 DLD-1 cells treated with DMSO or DOX. Data represent two independent experiments. (b) Representative browser tracks showing the SSB1 ChIP–Rx signals in DMSO- and DOX-treated cells with DOX-inducible RNase H1 expression. (c) SSB1 ChIP–qPCR on promoters of example genes in cells with DOX-inducible RNase H1 expression. Values are mean ± SD. n = 3 biological replicates. Statistical analysis was performed using two-tailed t-tests. *P* values are shown at the top of the graphs. (d) Schematic presentation of the workflow of R-loop CUT&Tag experiments. (e) R-loop CUT&Tag–qPCR in cells with DOX-inducible RNase H1 expression. DMSO-treated cells were incubated with IgG but not S9.6 (3rd lane) or treated with RNase H1 during CUT&Tag (4th lane) to confirm the specificity of detected R-loop signals. Values are mean ± SD. n = 3 biological replicates. Statistical analysis was performed using two-tailed t-tests. *P* values are shown at the top of the graphs. (f) Metaplots of SSB1 signals over 6 kb regions centred on TSS of SOSS–INTAC target genes in DMSO- and DOX-treated DLD-1 cells with inducible RNase H1 expression. (g) Representative browser tracks showing

the INTS3 ChIP–Rx signals in DMSO- and DOX-treated DLD-1 cells with DOX-inducible RNase H1 expression. (h) INTS3 ChIP–qPCR on promoters of example genes in cells with DOX-inducible RNase H1 expression. Values are mean ± SD. n = 3 biological replicates. Statistical analysis was performed using two-tailed t-tests. *P* values are shown at the top of the graphs. (i) Heatmaps showing INTS3 signals over 6 kb regions centred on TSS of SOSS–INTAC target genes in DMSO- and DOX-treated cells with DOX-inducible RNase H1 expression. (j) Boxplots of INTS3 signals at promoters of SOSS–INTAC target genes in DMSO- and DOX-treated cells with DOX-inducible RNase H1 expression. In boxplots, the centre line is the median, the top and bottom hinges correspond to the first and third quartiles, respectively, whiskers extend to quartiles ± 1.5 × interquartile range. *P* values were calculated using two-sided Wilcoxon rank-sum tests. *P* < 2.2e-16, n = 10,650 promoters. (k) Metaplot of INTS3 signals over 6 kb regions centred on TSS of SOSS–INTAC target genes in DMSO- and DOX-treated cells with DOX-inducible RNase H1 expression.

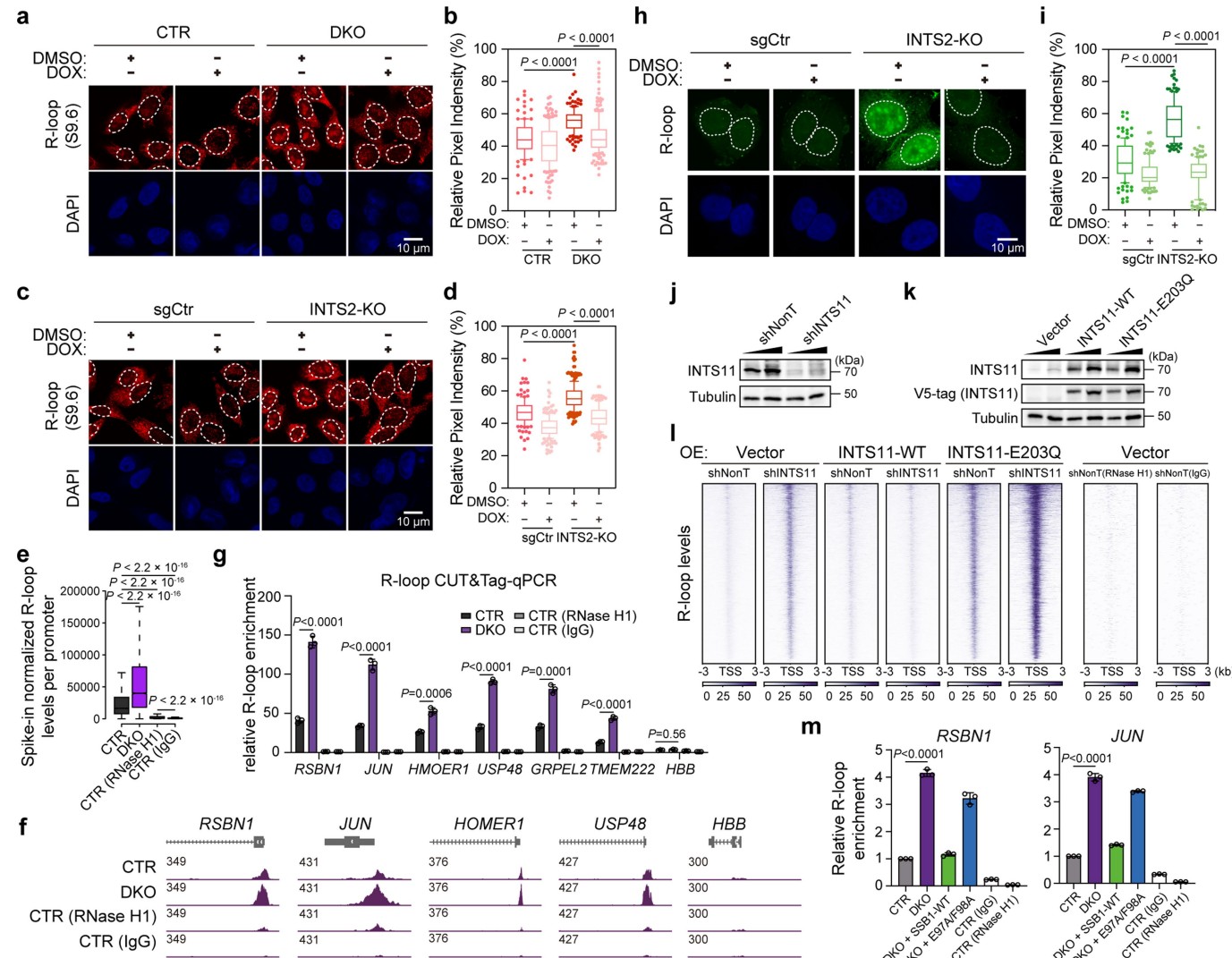

**Extended Data Fig. 5 | SOSS–INTAC regulates cellular R-loop levels.** (a-b) IF of S9.6-based R-loop detection in CTR and DKO cells with DOX-inducible RNase H1 expression (a) and the quantification of the nuclear R-loop signals (b). Statistical analyses were performed using two-tailed unpaired t-test (n = 120 foci from one representative experiment, which has been performed twice with similar results). *P* values are shown at the top of the graphs. (c–d) IF of S9.6-based R-loop detection in sgCtr and INTS2-KO DLD-1 cells with DOX-inducible RNase H1 expression (c) and the quantification of the nuclear R-loop signals. Statistical analyses were performed using two-tailed unpaired t-test (n = 120 foci from one representative experiment, which has been performed twice with similar results). *P* values are shown at the top of the graphs. (e) Boxplots of R-loop signals at promoters of SOSS–INTAC target genes in CTR and DKO cells. CTR cells were treated with RNase H1 protein during CUT&Tag (4th lane) or incubated with IgG but not S9.6 (5th lane) to confirm the specificity of detected R-loop signals. In boxplots, the centre line is the median, the top and bottom hinges correspond to the first and third quartiles, respectively, whiskers extend to quartiles ±1.5 × interquartile range. *P* values were calculated using two-sided Wilcoxon rank-sum tests. *P* < 2.2e-16, n = 10,650 promoters for all comparisons. (f) Representative browser tracks

showing the R-loop signals in CTR and DKO cells. (g) R-loop CUT&Tag–qPCR on example genes in CTR or DKO cells. Values are mean ± SD. n = 3 biological replicates. Statistical analysis was performed using two-tailed t-tests. *P* values are shown at the top of the graphs. (h-i) GFP–dRNASEH1-based IF of R-loops in sgCtr and INTS2-KO DLD-1 cells with DOX-inducible RNase H1 expression (h) and the quantification of the nuclear R-loop signals. Statistical analyses were performed using two-tailed unpaired t-test (n = 110 foci from one representative experiment, which has been performed twice with similar results). *P* values are shown at the top of the graphs. (j) Western blotting showing INTS11 knockdown efficiency in DLD-1 cells. (k) Western blotting showing the overexpression of wild-type or catalytic-dead (E203Q) INTS11 in DLD-1 cells. (l) Heatmaps of R-loop CUT&Tag signals over 6 kb regions centred on TSS of SOSS–INTAC target genes in DLD-1 cells with INTS11 knockdown and overexpression of wild-type or E203Q INTS11. (m) R-loop CUT&Tag–qPCR on example genes in CTR or DKO cells overexpressed with empty vector, wild-type SSB1 or E97A/F98A, the mutant defective in interacting with INTS3. Values are mean ± SD. n = 3 biological replicates. Statistical analysis was performed using two-tailed t-tests. *P* values are shown at the top of the graphs.

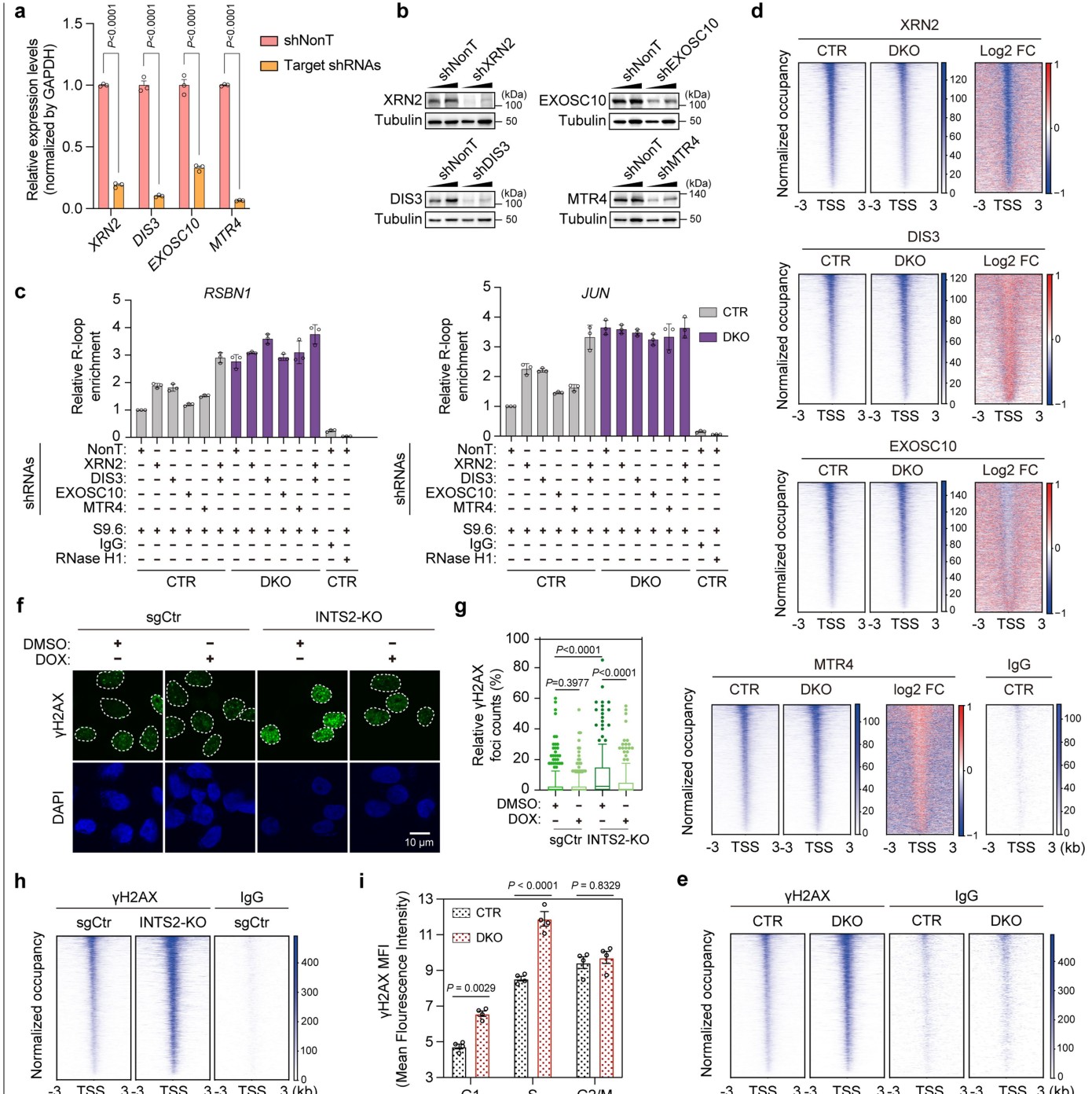

**Extended Data Fig. 6 | Cooperation of SOSS–INTAC and RNA exonucleases.**
(a-b) Quantitative reverse transcription PCR (RT–qPCR) (a) and western
blotting (b) to determine the knockdown efficiency of XRN2, DIS3, EXOSC10,
and MTR4 in DLD-1 cells. n = 3 biological replicates. Statistical analysis was
performed using two-tailed t-tests. P values are shown at the top of the graphs.
(c) R-loop CUT&Tag in CTR and DKO cells with knockdown of XRN2, DIS3,
EXSOC10, and MTR4. Values are mean ± SD (n = 3 biological replicates).
(d) Heatmaps showing the occupancy of XRN2, DIS3, EXOSC10, and MTR4
in CTR and DKO cells. (e) Heatmaps showing γH2AX occupancy in CTR and
DKO cells. The peaks were centred on TSS of SOSS–INTAC target genes.
(f-g) Immunostaining of γH2AX signal in sgCtr and INTS2-KO DLD-1 cells with

DOX-inducible RNase H1 expression (f) and the quantification of the nuclear
γH2AX foci number (g). Statistical analyses were performed using two-tailed
unpaired t-test (n = 180 foci from one representative experiment, which has
been performed twice with similar results). P values are shown at the top of the
graphs. (h) Heatmaps showing γH2AX occupancy in sgCtr and INTS2-KO cells.
The peaks were centred on TSS of SOSS–INTAC target genes. (i) Flow cytometry
analysis following propidium iodide labelling and γH2AX staining in CTR and
DKO cells. Propidium iodide signal was used to separate cells into G1, S, and
G2/M phases. Values are mean ± SD (n = 3 biological replicates). Statistical
analysis was performed using two-tailed t-tests. P values are shown at the top of
the graphs.

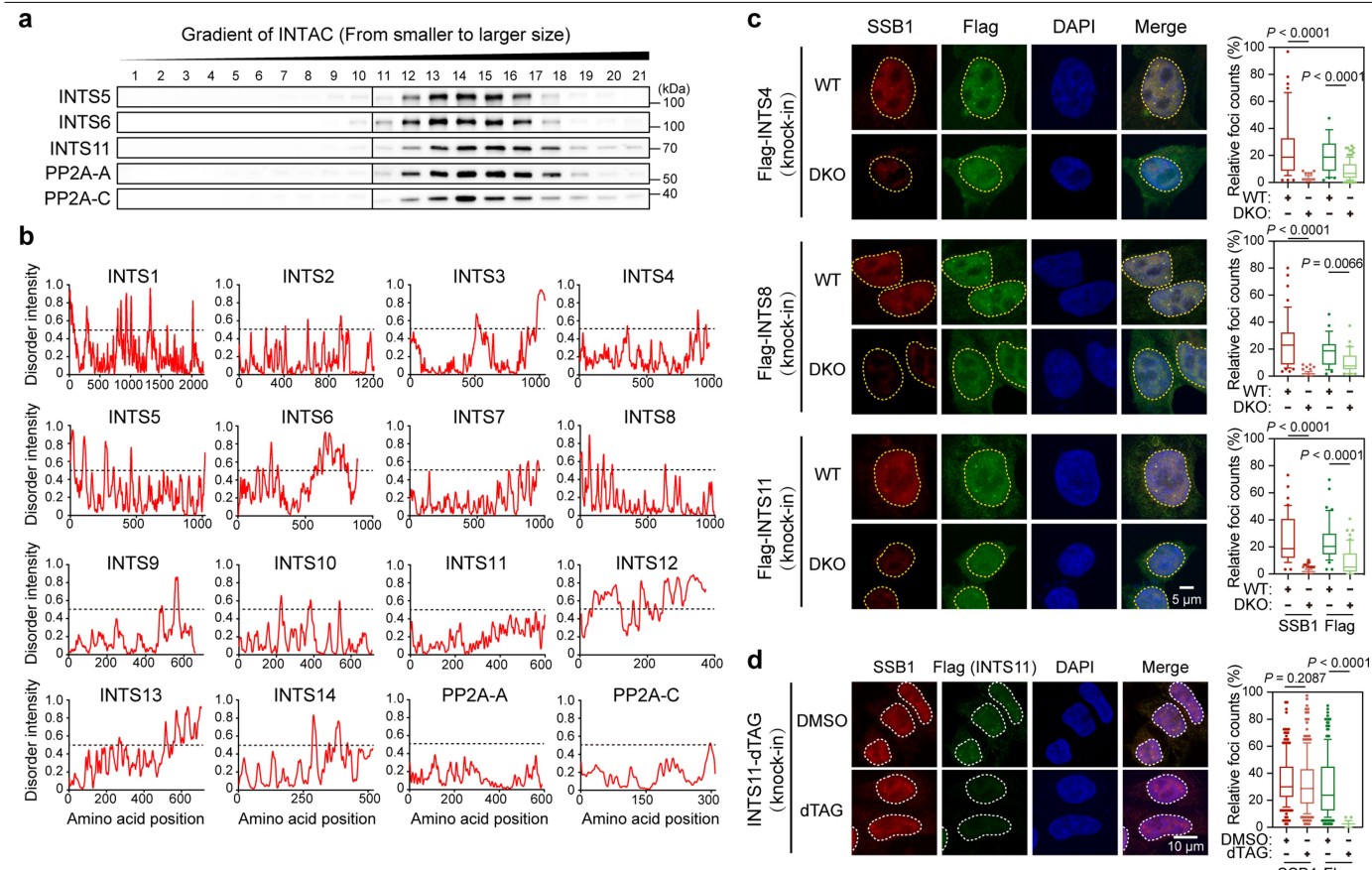

**Extended Data Fig. 7 | Disordered tendency prediction of SOSS–INTAC and its punctum formation in cells.** (a) Gradient centrifugation using purified INTAC from HEK Expi293 cells with overexpression of all INTAC subunits. The fractionated samples were examined by SDS–PAGE followed by western blotting. Data shown represent two independent experiments. (b) Intrinsically disordered tendency of all INTAC subunits. IUPred assigned scores of disordered tendencies between 0 and 1 to the sequences, and a score of more than 0.5 indicates disorder. (c) The immunofluorescent images of SSB1 (red) and INTAC subunits (green) in wild-type and DKO cells (left) and the quantification of the relative foci counts (right, n = 150 foci from one representative experiment, which has been performed twice with similar results). Statistical analysis was performed using two-tailed t-tests. P values are shown at the top of the graphs. (d) The immunofluorescent images of SSB1 (red) and INTS11 (green) in DMSO- or dTAG-treated INTS11-dTAG DLD-1 cells (left) and the quantification of the relative foci counts (right, n = 150 foci from one representative experiment, which has been performed twice with similar results). Statistical analysis was performed using two-tailed t-tests. P values are shown at the top of the graphs.

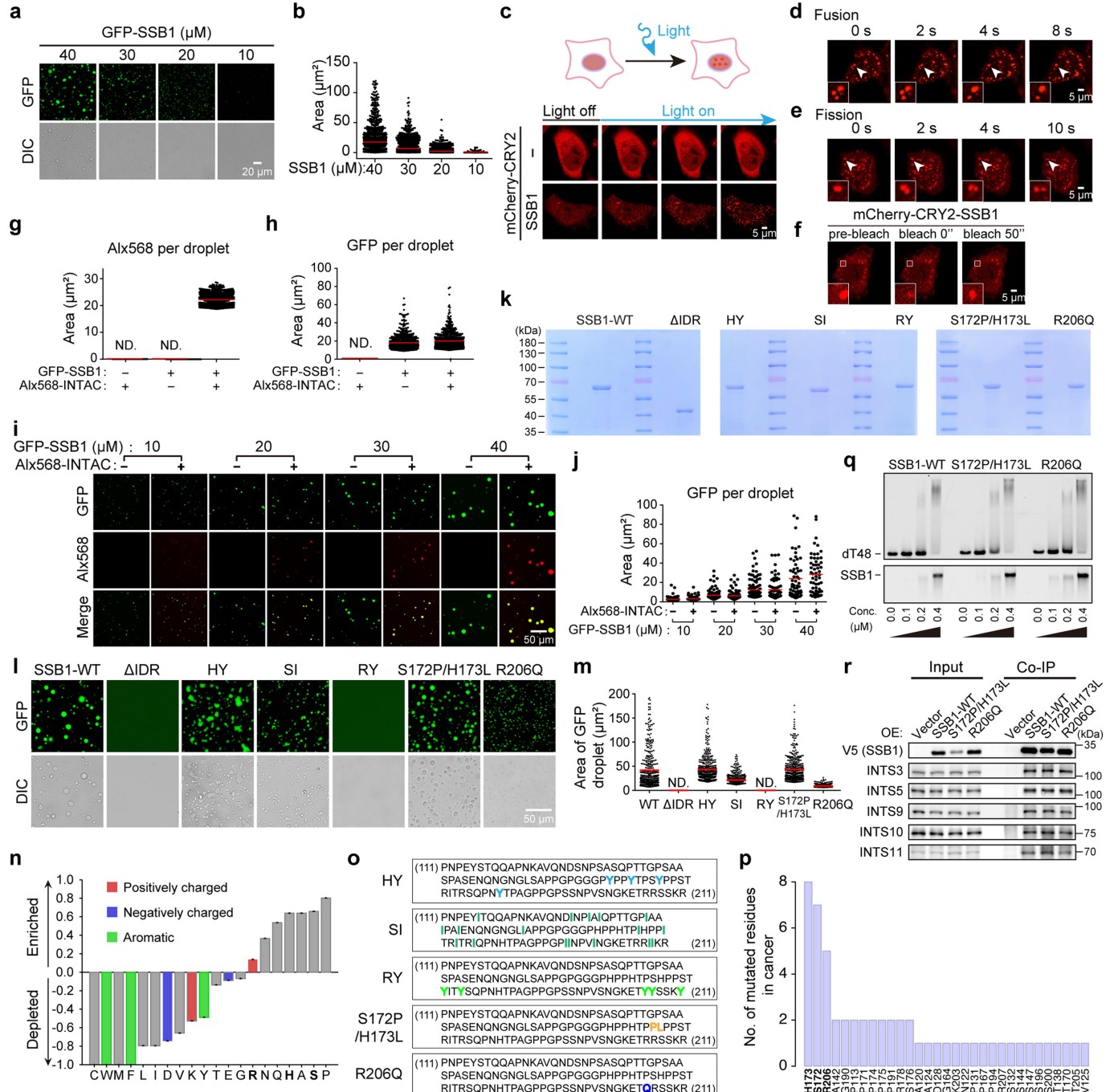

**Extended Data Fig. 8 | Analysis of condensate formation capacity of SSB1 and SOSS–INTAC.** (a-b) GFP–SSB1 was analysed using droplet formation assays with indicated concentration at 37.5 mM NaCl (a) and the quantification of the size of droplets (b). Red lines indicate the mean in each population (n = 500 foci analysed across two independent experiments). (c) The establishment of the "optoDroplet" system by fusing SSB1 with mCherry-labelled Arabidopsis photoreceptor cryptochrome 2 (CRY2) in Hela cells. Representative images of SSB1–mCherry–CRY2 and empty mCherry–CRY2 vector were shown before and after light induction. (d-e) Time-lapse imaging demonstrating spontaneous fusions (d) and fissions (e), as indicated by the arrows, of SSB1 condensates in cells. (f) Representative micrographs of SSB1 puncta before and after photobleaching. (g-h) Quantification of the relative intensity of Alx568 (g) and GFP (h) per droplet for Alx568-labelled INTAC, GFP–SSB1, and the mixture of Alx568-labelled INTAC and GFP–SSB1. Red lines indicate the mean in each group (n = 500 foci analysed across two independent experiments). ND, not detected. (i-j) Different concentrations of GFP–SSB1 were mixed with Alx568-labelled INTAC and analysed using the droplet formation assay (i),

followed by the quantification of the relative GFP intensity per droplet (j). Red lines indicate the mean in each group (n = 300 foci analysed across two independent experiments). (k) Recombinant wild-type or mutant GFP–SSB1 were purified from *E. coli*. Each protein was examined by SDS–PAGE followed by Coomassie blue staining. (l-m) Fluorescence microscopy images of purified GFP–SSB1 mutants (l). Quantification of the scale per GFP droplets is shown in (m). Red lines indicate the mean in each group. ND, not detected. (n) Analysis of amino acid enrichment for SSB1 IDR by Composition Profiler. The full-length SSB1 is used as background. (o) Diagram summarizing the mutated residues of the indicated SSB1 mutants. (p) Mutation information of SSB1/NABP2 in the COSMIC reference database. (q) EMSA assays using Cy3-labelled oligo (dT)48 incubated with wild-type SSB1, SSB1 (S172P/H173L), or SSB1 (R206Q). Data represent two independent experiments. (r) V5 Co-IP in cells overexpressed with V5-tagged wild-type SSB1, SSB1(S172P/H173L), or SSB1(R206Q) followed by western blotting of SOSS–INTAC subunits. Data represent two independent experiments.

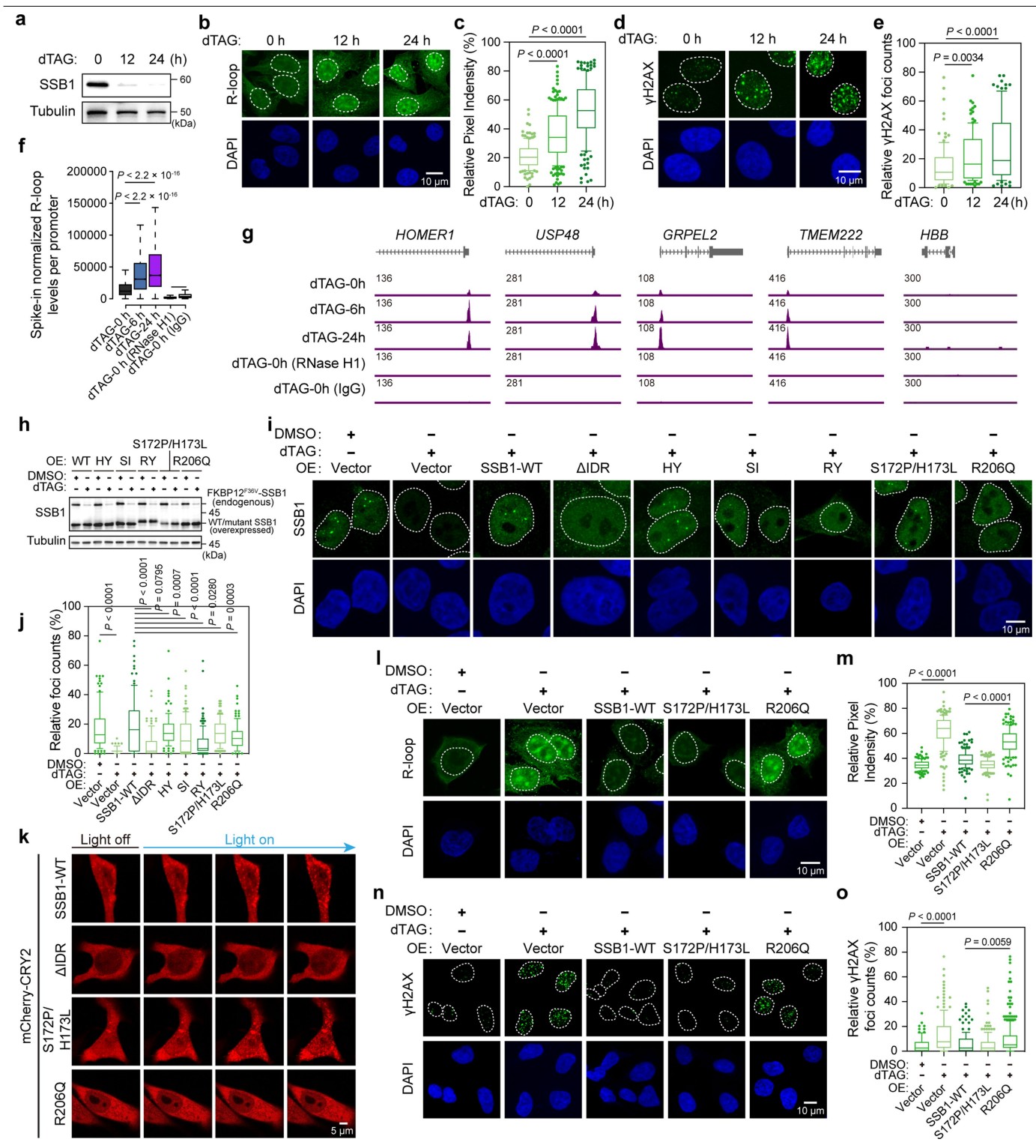

**Extended Data Fig. 9** | See next page for caption.

**Extended Data Fig. 9 | Dynamic regulation of R-loops by SOSS–INTAC and its puncta formation in cells.** (a) Western blotting of SSB1-dTAG DLD-1 cells with time-course treatment of dTAG. Data represent two independent experiments. (b-c) Immunostaining of R-loop signals in SSB1-dTAG DLD-1 cells with time-course dTAG treatment (b). Quantification of the nuclear R-loop signals is shown in (c). Statistical analyses were performed using two-tailed unpaired t-test (n = 150 foci from one representative experiment, which has been performed twice with similar results). Statistical analysis was performed using two-tailed t-tests. P values are shown at the top of the graphs. (d-e) Immunostaining of γH2AX signal in SSB1-dTAG DLD-1 cells with time-course dTAG treatment (d). Quantification of the nuclear γH2AX foci number is shown in (e). Statistical analyses were performed using two-tailed unpaired t-test (n = 150 foci from one representative experiment, which has been performed twice with similar results). Statistical analysis was performed using two-tailed t-tests. P values are shown at the top of the graphs. (f) Boxplots of R-loop CUT&Tag signals at promoters of SOSS–INTAC target genes in SSB1-dTAG cells with time-course dTAG treatment. One sample was treated with RNase H1 protein (4th lane) or incubated with IgG but not S9.6 (5th lane) during CUT&Tag to verify the specificity of R-loop signals. In boxplots, the centre line is the median, the top and bottom hinges correspond to the first and third quartiles, respectively, whiskers extend to quartiles ±1.5 × interquartile range. P values were calculated using two-sided Wilcoxon rank-sum tests. P values are shown at the top of the graphs, n = 10,650 promoters for all comparisons.

(g) Representative browser tracks showing the R-loop signals in DMSO- or dTAG-treated SSB1-dTAG cells. (h) DMSO- or dTAG-treated SSB1-dTAG cells were overexpressed with wild-type or mutant SSB1 and analysed by western blotting. Data represent two independent experiments. (i-j) Representative images of SSB1 immunofluorescent signals in dTAG-treated SSB1-dTAG cells with overexpression of wild-type or mutant SSB1 (i). Quantification of the nuclear SSB1 foci number is shown in (j) (n = 150 foci from one representative experiment, which has been performed twice with similar results). Statistical analysis was performed using two-tailed t-tests. P values are shown at the top of the graphs. (k) The "optoDroplet" assay measuring the punctum formation ability of wild-type SSB1, ΔIDR, and cancer-derived mutants (S172P/H173L and R206Q) in Hela cells. Representative images were shown before and after light induction. (l-m) R-loop IF in dTAG-treated SSB1-dTAG cells with overexpression of wild-type SSB1 or cancer-derived mutants (l). Quantification of the nuclear R-loop signals is shown in (m) (n = 150 foci from one representative experiment, which has been performed twice with similar results). Statistical analysis was performed using two-tailed t-tests. P values are shown at the top of the graphs. (n-o) Immunostaining of γH2AX signal in dTAG-treated SSB1-dTAG cells with overexpression of wild-type SSB1 or cancer-derived mutants (n). Quantification of the nuclear γH2AX foci number is shown in (o). (n = 150 foci from one representative experiment, which has been performed twice with similar results). Statistical analysis was performed using two-tailed t-tests. P values are shown at the top of the graph.

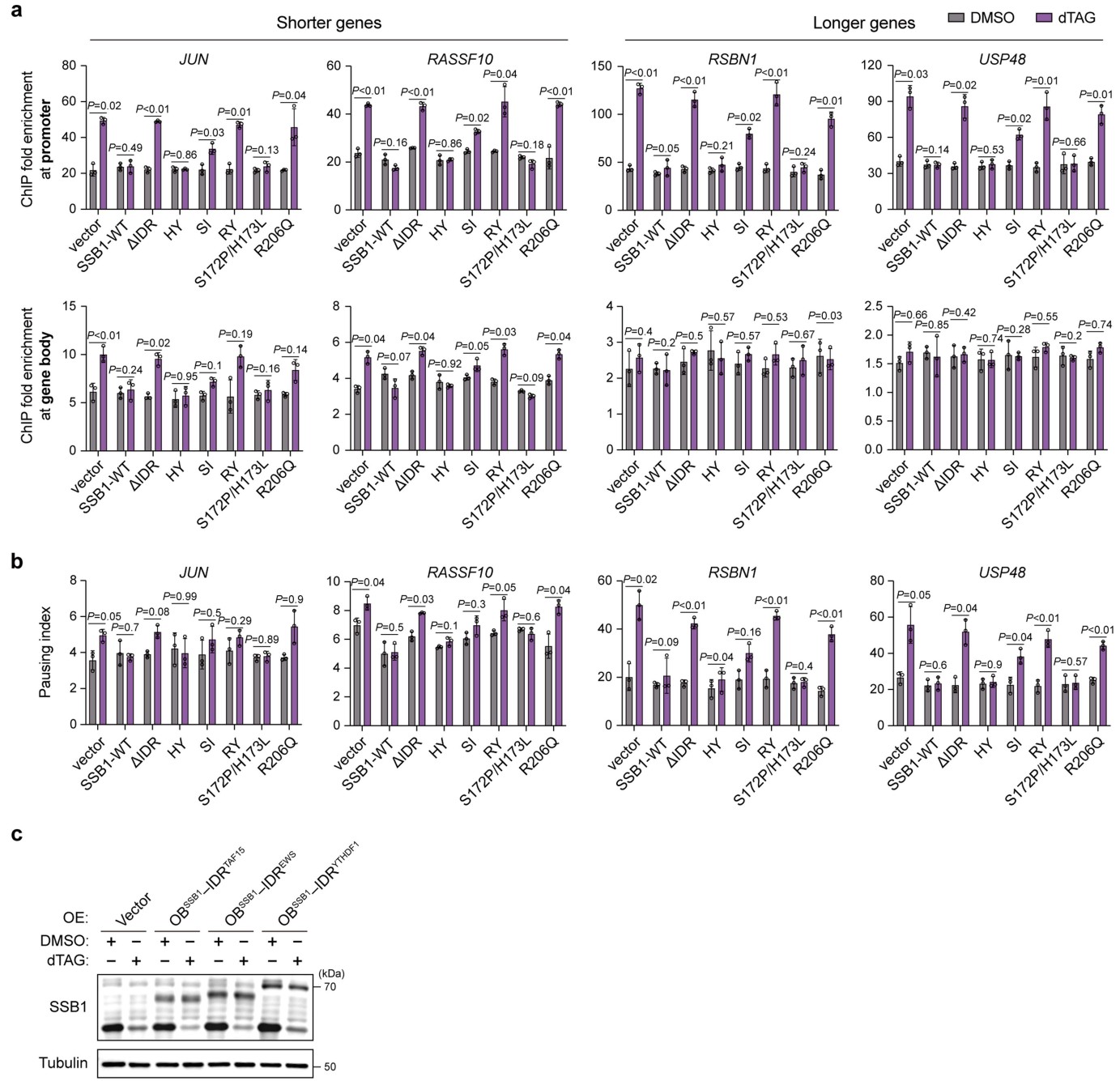

**Extended Data Fig. 10 | Analysis of Pol II pausing regulated by SSB1 mutants.** (a) Pol II ChIP–qPCR at promoters (top) and gene bodies (bottom) of example genes (*JUN* and *RASSF10* as shorter genes; *RSBN1* and *USP48* as longer genes) in DMSO- or dTAG-treated SSB1-dTAG cells with overexpression of wild-type or mutant SSB1. Values are mean ± SD (n = 3 biological replicates). Statistical analysis was performed using two-tailed t-tests. *P* values are shown at the top

of the graphs. (b) Pausing index of example genes (*JUN* and *RASSF10* as shorter genes; *RSBN1* and *USP48* as longer genes) in DMSO- or dTAG-treated SSB1-dTAG cells with overexpression of wild-type or mutant SSB1. Values are mean ± SD (n = 3 biological replicates). Statistical analysis was performed using two-tailed t-tests. *P* values are shown at the top of the graphs. (c) DMSO- or dTAG-treated SSB1-dTAG cells were overexpressed with fused proteins comprising N terminus of SSB1 and IDRs of TAF15, EWS, or YTHDF1 and followed by

# Reporting Summary

## Statistics

For all statistical analyses, confirm that the following items are present in the figure legend, table legend, main text, or Methods section.

| n/a | Confirmed | |
|---|---|---|
| ☐ | ☒ | The exact sample size (*n*) for each experimental group/condition, given as a discrete number and unit of measurement |
| ☐ | ☒ | A statement on whether measurements were taken from distinct samples or whether the same sample was measured repeatedly |
| ☐ | ☒ | The statistical test(s) used AND whether they are one- or two-sided *Only common tests should be described solely by name; describe more complex techniques in the Methods section.* |
| ☒ | ☐ | A description of all covariates tested |
| ☐ | ☒ | A description of any assumptions or corrections, such as tests of normality and adjustment for multiple comparisons |
| ☐ | ☒ | A full description of the statistical parameters including central tendency (e.g. means) or other basic estimates (e.g. regression coefficient) AND variation (e.g. standard deviation) or associated estimates of uncertainty (e.g. confidence intervals) |
| ☐ | ☒ | For null hypothesis testing, the test statistic (e.g. *F*, *t*, *r*) with confidence intervals, effect sizes, degrees of freedom and *P* value noted *Give P values as exact values whenever suitable.* |
| ☒ | ☐ | For Bayesian analysis, information on the choice of priors and Markov chain Monte Carlo settings |
| ☐ | ☒ | For hierarchical and complex designs, identification of the appropriate level for tests and full reporting of outcomes |
| ☐ | ☒ | Estimates of effect sizes (e.g. Cohen's *d*, Pearson's *r*), indicating how they were calculated |

*Our web collection on statistics for biologists contains articles on many of the points above.*

## Software and code

Policy information about availability of computer code

| Data collection | Expression of SSB1 and SSB2 across tissues were analyzed by GTEx (https://gtexportal.org/home/). SSB1 mutations in human cancer were analyzed by COSMIC (https://cancer.sanger.ac.uk/cosmic). |
|---|---|
| Data analysis | Disordered regions were identified using IUPred v2A (http:// iupred.elte.hu/). Amino acid composition was analyzed using Composition Profiler (http://www.cprofiler.org/cgi-bin/profiler.cgi). The net charge per residue was analyzed by CIDER 40 (http://pappulab.wustl. edu/CIDER/analysis/). Images were acquired by confocal microscope (Lecia TCS SP8) and analyzed with software ZEN v2.3 SP1 (ZEISS). Statistical analysis was performed using Prism v8. |

For manuscripts utilizing custom algorithms or software that are central to the research but not yet described in published literature, software must be made available to editors and reviewers. We strongly encourage code deposition in a community repository (e.g. GitHub). See the Nature Portfolio guidelines for submitting code & software for further information.

## Data

Policy information about availability of data

All manuscripts must include a data availability statement. This statement should provide the following information, where applicable:
- Accession codes, unique identifiers, or web links for publicly available datasets
- A description of any restrictions on data availability
- For clinical datasets or third party data, please ensure that the statement adheres to our policy

The high-throughput sequencing data, including ChIP-Rx, KAS-seq, PRO-seq, ATAC-seq and CUT&Tag, have been deposited in Gene Expression Omnibus database with the accession number GSE223997. The scripts used to analyze the data from this study are freely available at: https://github.com/chenjiwei124128/SSB1_NGS_analysis

## Human research participants

Policy information about studies involving human research participants and Sex and Gender in Research.

| Reporting on sex and gender | N/A |
|---|---|
| Population characteristics | N/A |
| Recruitment | N/A |
| Ethics oversight | N/A |

Note that full information on the approval of the study protocol must also be provided in the manuscript.

# Field-specific reporting

Please select the one below that is the best fit for your research. If you are not sure, read the appropriate sections before making your selection.

☒ Life sciences   ☐ Behavioural & social sciences   ☐ Ecological, evolutionary & environmental sciences

For a reference copy of the document with all sections, see nature.com/documents/nr-reporting-summary-flat.pdf

# Life sciences study design

All studies must disclose on these points even when the disclosure is negative.

| Sample size | No sample size was predetermined. Sample size was estimated by robustness of phenotype based on our preliminary experiments and previous work (Zheng et al. Science 2020 Nov 27;370(6520)). proper negative and whenever possible positive controls were used for each experiment. |
|---|---|
| Data exclusions | No data exclusions occurred in this study. |
| Replication | All key in vitro experiments were performed at least 3 times and all replication attempts were successful. |
| Randomization | cells were always randomly allocated into control and experimental groups. All samples used in each set of experiments were equal, except the experimental condition being tested. |
| Blinding | blinding was not done since this study relies on the investigator studying differences in cellines |

# Reporting for specific materials, systems and methods

We require information from authors about some types of materials, experimental systems and methods used in many studies. Here, indicate whether each material, system or method listed is relevant to your study. If you are not sure if a list item applies to your research, read the appropriate section before selecting a response.

## Materials & experimental systems

| n/a | Involved in the study |
|---|---|
| ☐ | ☒ Antibodies |
| ☐ | ☒ Eukaryotic cell lines |
| ☒ | ☐ Palaeontology and archaeology |
| ☒ | ☐ Animals and other organisms |
| ☒ | ☐ Clinical data |
| ☒ | ☐ Dual use research of concern |

## Methods

| n/a | Involved in the study |
|---|---|
| ☐ | ☒ ChIP-seq |
| ☐ | ☒ Flow cytometry |
| ☒ | ☐ MRI-based neuroimaging |

# Antibodies

| | |
|---|---|
| Antibodies used | Anti-SSB1 (WB, 1:1000; IP, 2ug; ChIP, 2ug) Proteintech Cat. # 14809-1-AP<br>Anti-SSB2(WB, 1:1000) Proteintech Cat. # 16719-1-AP<br>Anti-INIP(WB, 1:300) sino biological Cat: # 204969-T10<br>Anti-INTS1(WB, 1:1000) Bethyl Laboratories Cat. # A300361A<br>Anti-INTS2(WB, 1:500) Santa Cruz Cat. # sc-514945<br>Anti-INTS3(WB, 1:2000; IP, 2ug; ChIP, 2ug) Proteintech Cat. # 16620-1-AP<br>Anti-INTS4(WB, 1:1000) Proteintech Cat. # 16130-1-AP<br>Anti-INTS5(WB, 1:2000; IP, 2ug; ChIP, 2ug) Proteintech Cat. # 14069-1-AP<br>Anti-INTS6(WB, 1:1000) Santa Cruz Cat. # sc-376524<br>Anti-INTS9(WB, 1:1000) Cell Signaling technology Cat. # 13945S<br>Anti-INTS10(WB, 1:1000) Proteintech Cat. # 15271-1-AP<br>Anti-INTS11(WB, 1:1000) Bethyl Laboratories Cat. # A301-274A<br>Anti-PP2A-A(WB, 1:1000) Proteintech Cat. # 15882-1-AP<br>Anti-PP2A-C(WB, 1:1000) Proteintech Cat. # 13482-1-AP<br>Anti-XRN2(WB, 1:1000; ChIP, 2ug) Proteintech Cat. # 11267-1-AP<br>Anti-DIS3(WB, 1:1000; ChIP, 2ug) Bethyl Laboratories Cat. # A303-765A<br>Anti-EXOSC10(WB, 1:1000; ChIP, 2ug) Proteintech Cat. # 16731-1-AP<br>Anti-MTR4(WB, 1:1000; ChIP, 2ug) Proteintech Cat. # 12719-2-AP<br>Anti-Tubulin(WB, 1:5000) Abclonal Cat. # AC008<br>V5 Tag Monoclonal Antibody (E10/V4RR)(WB, 1:5000; IP, 2ug) ThermoFisher Scientific Cat. # MA5-15253<br>Anti-H3K27ac(ChIP, 2ug) Abclonal Cat. # A7253<br>Anti-H3K4me1(ChIP, 2ug) Abclonal Cat. # A2355<br>Anti-H3K4me3(ChIP, 2ug) Abclonal Cat. # A2357<br>Anti-Rabbit IgG(IP, 2ug) Proteintech Cat. # B900610<br>Anti-Mouse IgG(IP, 2ug) Proteintech Cat. # B900620<br>Anti-Phospho-Rpb1CTD (Ser5) (D9N5I)(WB, 1:1000; ChIP, 2ug) Cell Signaling technology Cat. # 13523S<br>Anti-γH2AX(WB, 1:2000; CUT&Tag, 2ug; IF: 1:200) Bethyl Laboratories Cat. # A700-053<br>Anti-DYKDDDDK (Flag)(WB, 1:5000; IP, 2ug; IF, 1:100; ChIP, 2ug) Abclonal Cat. # AE005<br>Anti-S9.6(CUT&Tag, 2ug) Active Motif Cat. # 65683<br>Anti-Strep(WB, 1:1000) Abclonal Cat. # AE066<br>Anti-GST(WB, 1:1000) Huabio Cat. # EM80701<br>Alexa Fluor 488 conjugated goat anti-mouse IgG(IF, 1:1000) Yeasen Cat. # 33206ES60<br>Alexa Fluor 488 conjugated goat anti-rabbit IgG(IF, 1:1000) Yeasen Cat. # 33106ES60<br>Rhodamine (TRITC) Goat Anti-Mouse IgG (IF, 1:1000) Yeasen Cat. # 33209ES60<br>Rhodamine (TRITC) Goat Anti-Rabbit IgG(IF, 1:1000) Yeasen Cat. # 33109ES60<br>Mouse anti-rabbit IgG(CUT&Tag, 1:100) Solarbio Cat. # SPA231<br>Rabbit anti-mouse IgG(CUT&Tag, 1:100) Solarbio Cat. # K0034M<br>Anti-IdU(DNA fiber assay, 1:200) ThermoFisher Scientific Cat. # MA5-24879<br>Anti-BrdU(DNA fiber assay, 1:200) Abcam Cat. # AB6326 |
| Validation | All these antibodies were commercially obtained and validated by vendors and multiple published studies, see manufacture's website for references. in addition, for key antibodies such as SSB1,SSB2 were validated by knock out or pooled knock out as negative control. other INTAC subunits antibodies were validated in our previous work (Zheng et al. Science 2020 Nov 27;370(6520)). |

# Eukaryotic cell lines

Policy information about cell lines and Sex and Gender in Research

| | |
|---|---|
| Cell line source(s) | cell lines (SSB2 single KO, SSB2/1 double KO, SSB1-dTAG, INTS11-dTAG) were generated from DLD-1 cell line . DLD-1 cell was a gift from ALi shilatifard lab (Chikago, nortwestern university) who purchased it from ATCC. HEK Expi293 cell line were gifted from Yanhui Xu lab who purchased it from ATCC. |
| Authentication | Cells were cultured in media recommended by the vendors. We froze down stocks upon receiving the cell lines, and all experiments will be conducted on cells that have been passaged no more than 10 times. SSB2 single KO, SSB2/1 double KO were authenticated by western blot. SSB1-dTAG ,INTS11-dTAG cell line were authenticated by PCR and western blot. all rescue cell lines in SSB1-dTAG cell line were authenticated by western blot. |
| Mycoplasma contamination | All cell lines tested for Mycoplasma contamination every month. |

| Commonly misidentified lines (See ICLAC register) | No commonly misidentified lines were used. |

# ChIP-seq

## Data deposition

☒ Confirm that both raw and final processed data have been deposited in a public database such as GEO.

☒ Confirm that you have deposited or provided access to graph files (e.g. BED files) for the called peaks.

| Data access links *May remain private before publication.* | https://www.ncbi.nlm.nih.gov/geo/query/acc.cgi?acc=GSE223997 |

| Files in database submission | GSM7009573 ChIP-H3K27ac_WT<br>GSM7009574 ChIP-H3K4me1_WT<br>GSM7009575 ChIP-Input_CTR-rep1<br>GSM7009576 ChIP-Input_CTR-rep2<br>GSM7009577 ChIP-Input_DKO-rep1<br>GSM7009578 ChIP-Input_DKO-rep2<br>GSM7009579 ChIP-Input_INTS2KO-rep1<br>GSM7009580 ChIP-Input_INTS2KO-rep2<br>GSM7009581 ChIP-Input_RNaseH1-DMSO<br>GSM7009582 ChIP-Input_RNaseH1-DOX<br>GSM7009583 ChIP-Input_sgCtr-rep1<br>GSM7009584 ChIP-Input_sgCtr-rep2<br>GSM7009585 ChIP-INTS3_CTR-rep1<br>GSM7009586 ChIP-INTS3_CTR-rep2<br>GSM7009587 ChIP-INTS3_DKO-rep1<br>GSM7009588 ChIP-INTS3_DKO-rep2<br>GSM7009589 ChIP-INTS3_RNaseH1-DMSO<br>GSM7009590 ChIP-INTS3_RNaseH1-DOX<br>GSM7009591 ChIP-INTS3_WT-rep1<br>GSM7009592 ChIP-INTS3_WT-rep2<br>GSM7009593 ChIP-INTS5_CTR-rep1<br>GSM7009594 ChIP-INTS5_CTR-rep2<br>GSM7009595 ChIP-INTS5_DKO-rep1<br>GSM7009596 ChIP-INTS5_DKO-rep2<br>GSM7009597 ChIP-INTS5_WT-rep1<br>GSM7009598 ChIP-INTS5_WT-rep2<br>GSM7009599 ChIP-polII-NTD_CTR-rep1<br>GSM7009600 ChIP-polII-NTD_DKO-rep1<br>GSM7009601 ChIP-polII-NTD_DKO-rep2<br>GSM7009602 ChIP-polII-NTD_INTS2KO-rep1<br>GSM7009603 ChIP-polII-NTD_INTS2KO-rep2<br>GSM7009604 ChIP-polII-NTD_sgCtr-rep1<br>GSM7009605 ChIP-polII-NTD_sgCtr-rep2<br>GSM7009606 ChIP-polII-pSer5_CTR-rep1<br>GSM7009607 ChIP-polII-pSer5_CTR-rep2<br>GSM7009608 ChIP-polII-pSer5_DKO-rep1<br>GSM7009609 ChIP-polII-pSer5_DKO-rep2<br>GSM7009610 ChIP-polII-pSer5_INTS2KO-rep1<br>GSM7009611 ChIP-polII-pSer5_INTS2KO-rep2<br>GSM7009612 ChIP-polII-pSer5_sgCtr-rep1<br>GSM7009613 ChIP-polII-pSer5_sgCtr-rep2<br>GSM7009614 ChIP-SSB1_CTR-rep1<br>GSM7009615 ChIP-SSB1_CTR-rep2<br>GSM7009616 ChIP-SSB1_DKO-rep1<br>GSM7009617 ChIP-SSB1_DKO-rep2<br>GSM7009618 ChIP-SSB1_RNaseH1-DMSO<br>GSM7009619 ChIP-SSB1_RNaseH1-DOX<br>GSM7009620 ChIP-SSB1_WT-rep1<br>GSM7009621 ChIP-SSB1_WT-rep2 |

| Genome browser session (e.g. UCSC) | no longer applicable |

## Methodology

| Replicates | Two biological replicates for all next-generation sequencing assays. Most replication attempts were successful. |

| Sequencing depth | Experiment Total_reads Mapped_reads paired_or_single<br>ChIP-H3K27ac_WT 34386162 25400290 paired-end<br>ChIP-H3K4me1_WT 67740324 61197309 paired-end |

ChIP-INTS3_WT-rep1 61111200 42982933 paired-end
ChIP-INTS3_WT-rep2 54636934 40997251 paired-end
ChIP-INTS5_WT-rep1 64929820 47559737 paired-end
ChIP-INTS5_WT-rep2 50327662 37935209 paired-end
ChIP-SSB1_WT-rep1 56914618 42028409 paired-end
ChIP-SSB1_WT-rep2 64059192 49107486 paired-end
ChIP-SSB1_RNaseH1-DOX 57227964 49531601 paired-end
ChIP-SSB1_RNaseH1-DMSO 49184466 42439246 paired-end
ChIP-INTS3_RNaseH1-DOX 57668212 48020248 paired-end
ChIP-INTS3_RNaseH1-DMSO 57507732 48865122 paired-end
ChIP-INTS3_DKO-rep1 83293772 64896365 paired-end
ChIP-INTS3_CTR-rep1 45954694 38904877 paired-end
ChIP-INTS5_DKO-rep1 70733478 57259021 paired-end
ChIP-INTS5_CTR-rep1 49765950 41948072 paired-end
ChIP-SSB1_DKO-rep1 71510928 55547819 paired-end
ChIP-SSB1_CTR-rep1 38565858 33180987 paired-end
ChIP-INTS3_DKO-rep2 49993702 38254882 paired-end
ChIP-INTS3_CTR-rep2 56256174 46334771 paired-end
ChIP-INTS5_CTR-rep2 97567818 86752833 paired-end
ChIP-INTS5_DKO-rep2 77418878 67633730 paired-end
ChIP-SSB1_DKO-rep2 33969230 29784057 paired-end
ChIP-SSB1_CTR-rep2 53014482 47735092 paired-end
ChIP-polII-NTD_CTR-rep1 50741734 42489741 paired-end
ChIP-polII-NTD_DKO-rep2 52704998 44642137 paired-end
ChIP-polII-NTD_CTR-rep1 57975194 48550095 paired-end
ChIP-polII-NTD_INTS2KO-rep1 94371462 86506303 paired-end
ChIP-polII-NTD_sgCtr-rep1 68007300 61152353 paired-end
ChIP-polII-NTD_INTS2KO-rep2 61953164 51956438 paired-end
ChIP-polII-NTD_sgCtr-rep2 52508578 44312944 paired-end
ChIP-polII-pSer5_INTS2KO-rep1 54104288 46678714 paired-end
ChIP-polII-pSer5_sgCtr-rep1 46304356 37169083 paired-end
ChIP-polII-pSer5_DKO-rep1 71650252 62088506 paired-end
ChIP-polII-pSer5_CTR-rep1 70624292 57028670 paired-end
ChIP-polII-pSer5_INTS2KO-rep2 79133512 70577984 paired-end
ChIP-polII-pSer5_CTR-rep2 54497894 41170575 paired-end
ChIP-polII-pSer5_DKO-rep2 40652272 28648739 paired-end
ChIP-polII-pSer5_sgCtr-rep2 91952552 78205003 paired-end

| | |
|---|---|
| Antibodies | SSB1 (14809-1-AP, Proteintech), INTS3 (16620-1-AP, Proteintech), INTS5 (14069-1-AP, Proteintech), Pol II (NTD) (14958, Cell Signaling), Pol II (pSer5) (13523, Cell Signaling), H3K27ac (A7253, Abclonal). |
| Peak calling parameters | macs2 callpeak -f BAMPE -g hs --nomodel |
| Data quality | mapping rate > 70%, number of peaks > 20000 |
| Software | ChIP-Rx analyses used Trim Galore v0.6.6, Bowtie v2.4.4, SAMtools v1.12, Picard Tools v2.25.5, MACS2 v2.2.7.1, R package ChIPseeker v1.28.3, deepTools v3.5.1, bedTools v2.30.0, R package BRGenomics v1.4.0 |

# Flow Cytometry

## Plots

Confirm that:

☒ The axis labels state the marker and fluorochrome used (e.g. CD4-FITC).

☒ The axis scales are clearly visible. Include numbers along axes only for bottom left plot of group (a 'group' is an analysis of identical markers).

☒ All plots are contour plots with outliers or pseudocolor plots.

☒ A numerical value for number of cells or percentage (with statistics) is provided.

## Methodology

| | |
|---|---|
| Sample preparation | Single-cell suspension of CTR and DKO were incubated with 70% ethanol in −20°C for 2 h. Following twice of PBS wash, cells were fixed with 4% formaldehyde (PFA) for 15 min. Next, cells were permeabilized with 0.25% Triton X-100 in PBS for 15 min and blocked with 2% BSA in PBS for 30 min. For intracellular γH2AX staining, 1 × 106 cells were incubated with 1 μg γH2AX antibody (Thermo) overnight at 4°C, followed by incubation with Alexa Fluor 488-conjugated secondary antibody for 30 min at room temperature. After washing with PBS for 3 times, cells were treated with PI staining buffer (Sangon Biotech) according to the manufacturer's protocol. |
| Instrument | CytoFLEX Flow Cytometer (Beckman Coulter). |
| Software | FACSDiva Flow Cytometry Software (BD Biosciences), FlowJo (TreeStar). |

Cell population abundance | The initial cell population is greater than 30,000 and sorted cells with a purity >90% were subject to the following experiments. After sorting, sorted cells were re-run with the exact setting on the same instruments.

Gating strategy | Forward versus side scatter gating was used to identify cells and exclude debris and dead cells. A forward scatter height vs. forward scatter area density plot was used to exclude doublets. The PI (Propidium Iodide) signal was used to separate cells into G1, S, and G2/M phases.

☒ Tick this box to confirm that a figure exemplifying the gating strategy is provided in the Supplementary Information.

