## [Peer Review File · Nature]

Manuscript Title: R-loop dependent promoter-proximal termination ensures genome stability

Reviewer Comments & Author Rebuttals

Reviewer Reports on the Initial Version:

Referees' comments:

Referee #1 (Remarks to the Author):

Xu and colleagues begin their manuscript by uncovering a novel interaction between a recently discovered complex INTAC (INTS1-14 + PP2A core enzymes) and the SOSS complex (SSB1, INIP, INTS3, the latter protein being common in the 2 complexes) and further examine the mechanisms of recruitment of this complex and its function. They show that SOSS-INTAC complex is located at promoters and enhancers (ChIP-seq SSB1, INTS3, INTS5) and that SSB1 is required for INTS3 and INTS5 recruitment (using a Double KO cell line for SSB1 and SSB2, "DKO", to avoid potential SSB2 redundancy in the absence of SSB1). The manner SSB1 is recruited to chromatin is likely through its intrinsic ssDNA binding capacity (in vitro by EMSA and in vivo by correlation between SOSS-INTAC occupancy and ssDNA presence (KAS-seq)). In line with the known INTAC function, the authors propose a role for SOSS-INTAC complex in stimulating early termination at paused sites by preventing aberrant accumulation of paused Polymerase II (here demonstrated in the SSB1/2 DKO context). Since R-loops structure contains a displaced ssDNA strand, the authors investigated relationship between SOSS-INTAC and R-loops. On the one hand they showed that R-loop presence is required for SSB1 and INTS3 binding at promoters, and on the other hand that SSB1 and INTS2 and 11 are also needed to avoid the aberrant presence of R-loops, which causes increased DNA damage and delayed replication fork progression. Lastly, the authors investigate the capacity of SSB1 to mediate its function via liquid-like condensate formation. In vitro, purified human SSB1 form droplets that mediates SOSS-INTAC condensate formation. The authors identified the domain and residues involved in condensation formation and that correspond to residues mutated in cancer. The authors further show that the condensation capacity of SSB1 is necessary to avoid unwanted R-loops levels in vivo.

Altogether, the SOSS-INTAC complex is presented as a gatekeeper of genomic integrity by limiting aberrant R-loop formation at the transcriptional pause step through its capacity to form liquid droplets.

Overall, the paper reads very nicely, is presented in a coherent flow, and the data presented are of excellent quality with a wide range of complementary techniques. This work follows up on previously published data of the authors on the INTAC complex (2020) and greatly extends it by presenting novel insights into the manner in which this INTAC complex is recruited to chromatin. This study should be of interest for a wide readership since it establishes that the liquid condensation properties of SSB1, a protein 70x smaller than INTAC, enables INTAC recruitment to chromatin. Moreover, they show that that cancer-derived mutations abolish this condensation capacity and SSB1 intrinsic physical condensation properties are required for suppressing R-loop at promoters, a process important to avoid transcription-replication conflicts and undesired genomic instability. Overall, the authors connect RNA pol II pausing with genome stability maintenance. Some of the data presented in this study nevertheless need to be reinforced to better support the conclusions made (see comments below), mainly on the four main points of the paper, i.e., R-loop formation, Pol II pausing, genome instability, and LLPS

MAJOR COMMENTS

1) The authors show convincing evidence of IP-MS in nuclear extracts that SSB1 and INIP can independently interact with the INTAC complex and that GST-pull down of purified SSB1 also

precipitates reconstituted INTAC and purified INIP. The evidence suggesting that SOSS-INTAC form a super complex only comes from gel filtration experiments, but importantly lack the INIP panel. Thus, at this stage the evidence for the presence of this SOSS-INTAC super complex at promoter/enhancers is weak as one could easily argue that it is SSB1 alone, and not as part of the SOSS, that forms this super complex with INTAC. To be able to keep this statement "SOSS-INTAC" throughout the text, the authors need to reinforce their data (if not this should be modified to SSB1-INTAC complex). This could be done by showing INIP panel in gel filtration, or/and by performing endogenous INIP ChIP-qPCR to verify that its presence at few target SOSS-INTAC sites.

2) To show that the genomic instability observed with γ H2AX in DKO cells is directly connected with the role of SSB1 at promoters/enhancers and thus with Pol II pausing, the authors could perform γ H2AX by ChIP (or ChIP-seq) to address whether damage occurs at promoters/enhancer. In other words, is genomic instability caused at positions of SOSS-INTAC upon its depletion? Moreover, to differentiate whether it is mainly caused by replication conflicts, the authors could assess whether γ H2AX presence is only visible in S-phase cells. If not, the authors should slightly modify their statement.

3) Regarding the link with R-loops. Here are several suggestions

a. Figure 4: Analyses of R-loops by imaging is not appropriate without all the necessary controls (which are missing here) (see Smolka et al, J Cell Biol, 2020, PMID: 33830170 "S9.6 is subject to pervasive artifacts without pretreatments and controls that mitigate its promiscuous recognition of cellular RNAs" or Chedin et al, EMBO J, 2021, PMID 33411340 "S9.6 IF microscopy approaches should probably be avoided whenever possible. We also encourage investigators to make efforts to eliminate other confounding variables such as modifications to the nascent transcriptome. It is important to note that those recommendations apply equally if imaging were to be performed using dRNaseH1-based reagents instead of the S9.6 antibody"). R-loop IF data should be interpreted with extreme caution. S9.6 Cut&Tag could come to complement this analysis in DKO (as performed with SSB1 dTag Figure 6).

b. For the Cut&Tag the IgG control should be included. It is important to have this control since Cut&Tag has a preference for open regions such as those detected here (promoter/enhancer). It is thus critical to determine what is the real R-loop Cut&Tag profile from unspecific tagmentation in open regions.

4) The authors interpret increased PolII occupancy and increased PRO-seq at promoter and 5'end, as a sign of PolII pausing. This would need to be strengthened by both experiments and analysis. For instance, Pol II pausing could be addressed using antibodies against the phosphorylated Serine 2 form of RNA pol II. Moreover, to differentiate pausing from an increase in transcriptional, the authors could compute pausing index (ratio of PolII occupancy on gene body over promoter regions)

5) While the author presents clear evidence for SSB1 condensate formation in vitro, not much is analyzed in vivo. The authors could analyze whether SSB1 mutants that do not display the ability to form droplets in vitro, also abolish foci formation in vivo (as in Fig. 5c)

MINOR COMMENTS

1) In abstract the authors say that mutation in SSB1 IDR provokes pervasive accumulation of R-loops at "a genome-wide level". This refers to Figure 6 m and n, where only a few genes were tested (those bound by SOSS-INTAC). Therefore, the statement at a "genome-wide level" is not appropriate and should be accordingly modified.

2) In the abstract, perhaps change "early termination of transcription" to "early termination of transcription at Pol II paused sites", from that sentence, it seems that the paper will be about what

happens at TTS rather than at TSS. Same hold true for “timely transcriptional termination”, change to “timely transcriptional termination at Pol II paused sites”

3) Fig S1C legend says that SSB1, INIP and INTAC subunits were overexpressed. Yet, there is no panel for INIP. And in the text, there is no mention of INIP overexpression. Please adjust accordingly.

4) In Figures 1g, 2b, 5a, S2c and S3a please do not use the SOSS term, but rather SSB1 (see major comment above).

5) Genome browser examples of panel displaying the KAS-seq, and ChIP seq of SSB1, INTS3 and 5 should be shown. Otherwise, there is not insight into the quality of KAS-seq data, nor its resolution profile which could in theory affect the correlation coefficient.

6) It is confusing for reader for the single SSB2 KO to be called control. Please just name it SSB2 KO. In Figure 2f, please show the western blot panel for SSB2.

8) In Fig 5, it is unclear whether in the text of Figure legend whether it is all INTAC subunits that are used in the condensate experiment, or a subset or one. Please modify accordingly.

Referee #2 (Remarks to the Author):

Multisubunit Integrator-PP2A complex [INTAC] is emerging as a major player in regulation of RNA polymerase II transcription of non-coding and protein coding genes. Endonucleolytic cleavage of the nascent transcript by the Int11 subunit of the INTAC is involved in generating the 3' end of snRNAs, regulation of the expression of protein coding genes via premature termination, quality control by preventing poorly configured Pol II complexes from entering elongation and suppression of the unwanted non-coding transcription. How INTAC is recruited to Pol II to execute its functions in transcription represents one of the key unanswered questions in the field.

In this manuscript, Xu et al., demonstrate that single stranded DNA binding protein SSB1 that was previously demonstrated to interact with Int3 subunit of the Integrator forms a stable complex with the entire INTAC [SOSS-INTAC]. SSB1 shows similar genome wide occupancy to the Integrator subunits and coincides with positions of the ssDNA mapped by KAS-seq. Furthermore, occupancy of the INTAC at the promoter and enhancer regions is reduced upon SSB1/SSB2 KO [DKO]. Loss of either INTAC subunits, Int3 or Int5 leads to increase in the levels of the promoter proximal Pol II based on Pol II ChIP-seq and PRO-seq as well as nascent transcriptome analyses by TT-seq in agreement with the previous reports. Loss of SSB protein shows similar effect on Pol II and nascent transcriptome further supporting role of SSB in mediating recruitment and function of the INTAC. Depletion of these factors [Int3, Int5, SSB] is also associated with increased levels of RNase H sensitive R-loop structures in nuclei [observed by IF]. dTAG depletion of SSB1 also leads to accumulation of R-loops at the promoters based on R-loop genome wide mapping. Accumulation of R-loops upon SSB1/2 is associated with genomic instability and replication defects based on gamma H2AX levels and DNA fiber assay. Ints2 KO also leads to increase in gamma H2AX suggesting that it is likely that the entire SOSS-INTAC functions to maintain genome stability. Interestingly, endonucleolytic activity of the INTAC is required for regulation of R loops. On the other hand, R-loops are required to mediate recruitment of SOSS-INTAC. Finally, authors propose that SSB1 mediates formation of SOSS-INTAC condensates that can be disrupted by cancer associated mutations in IDR of SSB1. Mutations in IDR of SSB1 also show some increase in R loop levels leading to hypothesis that SSB1 suppresses R-loop formation by mediating formation of liquid-like condensates.

Although a link between SSB 1/Ints3 [doi: 10.1074/jbc.M109.039404] as well as interaction of SSB1 with multiple subunits of the Integrator [doi: 10.1083/jcb.200907026] have been previously reported, this work convincingly demonstrates formation of stable SOSS-INTAC complex, establishes functional connection between SSB, INTAC and R-loops providing insights into how INTAC is recruited during transcription and regulation of R-loops and genome stability by the

INTAC, which are interesting novel findings. However, some aspects of the analyses are lacking depths and conclusions can be further strengthened by additional mechanistic insights. The weakest part of the manuscript in my opinion is the part related to condensate formation mediated by SSB1 and its role, I am not convinced whether this indeed has any functionality. I therefore cannot recommend this manuscript for publication in its current form.

Specific comments:

-The observation that the endonucleolytic activity of the Ints11 is required for suppression of R-loops is very interesting and is worth exploring further. How does endonucleolytic cleavage could affect removal of the R-loops? Does this provides an entry for 3' and 5' exonucleases? Which ones?

-Does increase in R loops levels contribute to transcription? Can defects in transcription be reversed by over-expression of RNase H?

-Are R-loop a result of Pol II pausing? How does overexpression of RNase H affect nascent transcriptome?

-dTAG depletion is very long [12-24h]-could it be the reason for such a strong global increase in R loop signal genome wide [Fig 6I] due to pleiotropic effects?

-I think that section related to investigation of the functional contribution of SOSS-INTAC to Pol II transcription [page 9] is a bit sketchy and would benefit from further analyses and more careful interpretations of the data. It was recently reported by the Torben Jensen's lab [Lykke -Andersen et al., 2021] and others that depletion of the Integrator subunits leads to different effect on multi-exonic vs mono-exonic genes. Contribution of the Integrator to transcription is not limited to facilitation of premature termination, it also controls transcriptional output from promoters and prevents premature termination of transcription. For example, multi-exonic genes show increased number of reads at the promoter and acute decrease within the gene body suggesting that Integrator prevents premature termination at these genes. In contrast, monoexonic genes show increase in the promoter proximal transcription as well as gene body signal suggesting that Integrator elicit premature terminations for this group of TUs. Authors should perform more thorough analyses of the data with this in mind and integrate this with the R-loop data.

-I find functional aspects of the condensate formation very unconvincing and this should be downplayed. Additionally, the evidence for condensates formation is limited to in vitro experiments and these were shown to form only in low salt which is not physiological , Fig5d. On the other hand, potential contribution of cancer associated mutation to R-loops accumulation is interesting. How do these mutations affect transcription? Is SSB interaction with the Integrator affected? Is DNA binding compromised by these mutations? It would be good to perform genome wide mapping of R loops in these mutants and analyse links with the DNA damage/replication. Line 321- I would rephrase it as the notion of the involvement in disease development as there is no data to show that this drives cancer, this can be discussed but not in the result section.

- The targeting of SSB1 to ssDNA regions was proposed to represent a part of the mechanism for the INTAC recruitment. It would be informative to generate mutants defective in DNA biding to test whether DNA binding of SSB is important for the INTAC recruitment.

-Page 11, second paragraph-...S9.6 antibody potentially dsRNA...- change for :
...S9.6 antibody potentially can detect dsRNA...

-To account for changes in transcription, ChIP data should also be normalized for Pol II occupancy [fig 2j]

- Fig3c,d – metagenes for the whole gene would help to see changes in global transcription changes.

- EMSA assay -Fig2d (it should be explained what protein concentration was used), similar experiments with dsDNA or ssRNA would confirm the specificity of the interaction.

- SSB2 was also shown to interact with INTS3 and therefore might be redundant, it should be made clear what is the interplay between SSB1/2.

-Figure 4. Overexpressing RNaseH1 I would expect that the fork speed would go up or not change but it is down. Therefore not sure how to fully interpret this result. It would be good to have conditions when R-loop resolution is compromised and SSB1 is also depleted.

-Figure S10a sgCtr DAPI does not seem to correspond with the H2AX signal for DMSO+/DOX-

-The scale difference between heatmaps and box plots (ie. Fig2. J vs Fig2. K, L)-is it not scaled and sum?

-Fig3c log2FC and Fig3g heatmaps have a lot of red which is a bit unexpected as most of the values on the sgCtr/INTS2-KO or CTR/DKO are comparable outside the peak.

-FigS1b – not the same bands are visible in IP and input for INTAC and it does not look like Fig1e.

-Fig5c – negative controls for signal specificity (depletion of SSB1 and untagged INTs)
Extensive in vitro studies would benefit from the estimation of how many molecules are in foci in vivo.

-Fig6m – in text line 353 it is stated that SI mutant looks like WT but the figure shows that it is not the case.

Minor:

In methods there is a description of RNA-seq but not sure where the enrichment analysis is presented.

Figure 1e – the main result in the supplement, marker size missing.

Fig6a – R206Q in IDR is indicated H206 to Y

FigS14g would benefit if would be presented as FigS14f

Referee #3 (Remarks to the Author):

The paper by Xu et al describes the role of SOSS-INTAC complex in genome stability and transcription though its function in formation of condensates and R-loop resolution. Previous literature has already demonstrated the role of SOSS complex in DNA damage and repair and the role of INTAC complex, containing multiple Integrator subunits, in termination of non-productive transcripts. This paper is suggesting that there is formation of a bigger SOSS-INTAC complex in human cells, and it is involved in sensing ssDNA and preventing R-loop formation and genomic instability. The authors suggest that this SOSS-INTAC function is mediated though formation of

liquid condensates.

This paper contains vast amount of data and experimental techniques thrown in, which is impressive on one hand, however, on the other hand it lacks novelty and mechanistic insights required to be considered for publication in Nature journal. I found that this paper very quickly brushes through multiple techniques, often without providing essential controls. There is a lot of generalizations in this paper without looking into details of mechanisms in questions. I am still struggling to understand the novelty of this work, since the authors have not actually demonstrated how condensation function of SSB1 of the SOSS-INTAC complex, required for regulation of R-loops, is actually important for transcription and why.

Major comments:

1. The authors propose that the condensation function of SSB1 important for regulation of promoter-proximal pausing/transcription, however, they have not actually tested this experimentally. The authors need to use their condensation mutants and assess promoter proximal pausing. I do not think that simple CHIP-seq experiments are sufficient here, since the pausing index needs to be calculated.
2. This paper uses a lot of generalizations. Is the mechanism of action for SOSS-INTAC complex the same for all gene categories (i.e. coding vs non-coding, intronless, intron-containing etc respond in a similar way)? What happens with the genes which do not have a strong promoter-proximal pause?
3. The authors claim that R-loops are important for recruitment of SSB1, however this is not supported by the provided data. The authors used DOX-inducible over-expression of RNase H1 to test this. However, this experiment did not result in a significant decrease of R-loops as demonstrated in Fig S8a and c by IF. However, the authors demonstrate a reduction in SSB1 signal by ChIP on 4 specific genes. I would like to see the DRIP-qPCR validation that the R-loops are actually reduced on the genes where the SSB1 signal goes down supporting their claims.
4. Figure 3a shows that SSB1 depletion results in increase of Pol II over the whole JUNB gene. This profile looks quite different compared to RSBN1 gene (where there seems no effect in the body of the gene). Also the authors see an increase in sense and anti-sense transcription. These are rather strange phenotypes and they do not really fit with the idea of affected pausing (since in the pausing defect, the elongation should be affected and less Pol II would be expected in the body of the gene). What is happening with Pol II in the body of the genes – is this also going up? Can the authors provide an explanation to what is going on? I think the authors tried to generalize too much regarding their conclusions in this paper (see also my comment 2 above).
5. I have not found any information related to the number of repeats for genomic experiments, therefore the reproducibility of these data are not clear. This information should be included at least in the methods section. All figures with peak distribution on specific genes should include input tracks, since the read values are extremely low for the pull-down experiments in multiple figures (e.g. Fig 3i, Fig3 a; 2i; Fig4 a, Fig 6j) . Furthermore, these results also need to be validated by qPCR. IN some occasions the authors have this, however, in case of R-loop experiments in Fig. 4 a, this information is missing. The authors need to convince the readers that the IP signals are above the input background and are reproducible in independent biological experiments.
6. Figure 4f should include a demonstration of multiple nuclei and not just one cell. Did the authors do any selection for the cells over-expressing wt and mutant constructs of INT11, since the transfection efficiency is not 100%? Are they quantifying all cells or just cells over-expressed with the constructs? Furthermore, there is no statistics presented with the WT over-expression, however in the text the authors claim an effect of complementation? All the statistics need to be clearly presented.

7. The paper would benefit from a clear model.

Minor comments:

1. This paper presents various pieces of genome-wide analysis, often presented as genome-wide heat maps (Fig 1i, 2j etc). This is OK, however often the information is actually lost in such condensed presentation of the genome-wide data. Therefore, it is important to provide an experimental validation by qPCR.
2. In vitro binding experiments presented in Fig 2 d-e should include concentration range indicated on the figure, so that the readers can easily compare the differences between different protein/complexes analysed.
3. Figures 1a and 1k are repetitive. Only one should be left, for example 1k (with all corresponding labels) instead of 1a, b. Currently there is no description of Figure 1k in the figure legends. Furthermore, it is not immediately clear what A and C in green mean on the Fig 1A, possibly more complete names are required on the figure. Some of the subunits are not on the figure (i.e. INT 12), why is that- this needs to be stated in the legends.

Referee #4 (Remarks to the Author):

The authors report the genome stability regulator SOSS associates with the transcription regulator INTAC to terminate transcription and reduce R-loop formation from paused Pol II. The authors propose that, mechanistically, ssDNA generated from R-loop at active transcription sites recruits INTAC through the ssDNA binding protein SSB1 in the SOSS complex to reduce R-loop via RNA cleavage in a manner that depends on SSB1 phase separation. This work provides an important link between transcription and genome stability regulation and showcases the possible functional importance of protein phase separation. My major concern is on the mechanism of SOSS-INTAC regulation of R-loop formation. The author used immunofluorescence (IF) and a R-loop CUT&Tag technique to assess R-loop reduction for SSB1 knockdown or fast degradation. I find the role of SSB1 in R-loop regulation is well supported. However, to establish role of INTAC in R-loop regulation, the authors only used IF with a S9.6 antibody and GFP-tagged catalytical-dead RNase H1 for R-loop detection. I find this is less convincing because the elevated R-loops under the regulation of SOSS-INTAC should correspond to active transcription sites and elevated level of DNA damage stained by gamma H2Ax. However, IF stains of elevated R-loops do not show patterns similar to SOSS-INTAC foci or gamma H2Ax foci. Instead, the staining for S9.6 looks more like nucleoli and that of dRNaseH1 looks like regions excluded of DAPI staining, maybe due to strong non-specific binding in those regions. Therefore, I would be more convinced with data from the more sensitive R-loop CUT&Tag for INTAC knockdown and rescue. Another reason is a previous study (<https://doi.org/10.1111/mmi.14529>) shows that bacterial SSB can direct RNase HI foci to replication sites to reduce R-loop, which might provide an alternative explanation for SSB1 regulation of R-loop in this case. I think the authors should check this possibility to either rule out this pathway. Maybe SSB1 recruits INTAC and RNase H to work synergistically to reduce R-loop formation. Indeed, a previous study ([doi:10.1073/pnas.2000761117](https://doi.org/10.1073/pnas.2000761117)) on bacterial SSB phase separation shows SSB condensates can enrich many weak interacting partners. Other minor comments are:

1. The importance of IDR in SSB1 phase separation is well-supported with existing data. However, IDR is barely the driver for phase separation at physiological conditions (e.g., the authors used concentrations on the order of 50uM and low salt of 30-50mM). A previous study ([doi:10.1073/pnas.2000761117](https://doi.org/10.1073/pnas.2000761117)) on bacterial SSB has established several contributions of multivalence to SSB phase separation including oligomerization, DNA /RNA binding. The domain

analysis in this work does not show significant contribution from non-IDR regions. Is human SSB1 not a tetramer? If it's not, authors should discuss the differences in phase behavior and not make it look like IDR is the major driver of SSB1 phase separation without considering other sources of valance.

2. Since phase separation is based on in vitro data, the functional relevance of phase separation is implied and not directly proved. Therefore, I would advise the authors to make this distinction clear in the manuscript unless they want to provide direct evidence for SSB1 phase separation in cells at physiological conditions, which I think it's beyond the scope of this work. For the same reason, equal care should be taken when interpreting the mutation results in the IDR region. IDRs can have non-phase separating functions. For example, Wolak 2020 (<https://doi.org/10.1111/mmi.14529>) suggested "Stimulation requires docking of the intrinsically disordered C-terminus of SSB (SSB-Ct) into a binding pocket on RNase HI".

3. Line 262 "SOSS-INTAC forms condensates in cells" is misleading. Results show foci, which may not necessarily reflect concentrating effect required for the definition of condensates due to SOSS-INTAC binding to DNA. Please see distinction demonstrated for 53BP1 and gamma H2Ax in this paper (<https://doi.org/10.15252/embj.2018101379>).

4. Line 295 SSB1 drives the formation of SOSS-INTAC condensates. Data suggest more of the model SSB1 undergoes phase separation and then recruits INTAC. A SOSS-INTAC condensate implies both contribute to valance in phase separation. The authors could map the effect of INTAC on SSB1 phase diagram in vitro to gain a better picture instead of using the one data point the manuscript currently has. In addition, the authors could look at how SSB1 and INTAC affect each other's foci formation and their co-localization in cells to make it clearer/more relevant whether they co-phase separate or one phase separates and then recruits the other. Foci formation in cells in response to IDR deletion and mutations can also be a good assay to further test the importance of IDR revealed in the in vitro data.

5. Figure legends can use more detailed information. For example, Figure 6h: with DMSO or dTAG?

6. Typo Line 240: which is eliminated by DOX-induced expression of GFP-dRNH1 (Fig. 4d). Should be expression of WT RNaseH1?

Author Rebuttals to Initial Comments:

General response to reviewers

We greatly appreciate all of the reviewers' positive feedback, including their finding that our manuscript is "of excellent quality with a wide range of complementary techniques", "of interest for a wide readership", and contains "interesting novel findings", and that it "provides an important link between transcription and genome stability regulation". We have now addressed all of the reviewers' comments and criticisms with the substantial addition of new experimental data and bioinformatic analyses, which we have summarized below:

1) Experiments addressing comments about condensate formation by SOSS-INTAC and its functional relevance.

1. Fluorescence microscopy of endogenous INTAC subunits in wild-type and DKO cells to determine whether SSB1 regulates INTAC puncta formation (Extended Data Fig. 16a).
2. Fluorescence microscopy of endogenous INTAC subunits in INTS11-depleted cells to determine whether INTAC regulates SSB1 puncta formation (Extended Data Fig. 16b, c).
3. Establishing the "optoDroplet" system to determine whether SSB1 forms liquid-like condensates in living cells (Extended Data Fig. 17c-e).
4. Fluorescence recovery after photobleaching (FRAP) experiment in cells to evaluate the mobility of SSB1 foci (Extended Data Fig. 17f).
5. In vitro condensation assay performed by incubating different concentrations of SSB1 with INTAC to determine the interdependency of SSB1 and INTAC regarding their condensate formation (Extended Data Fig. 17h, i).
6. Fluorescence microscopy in dTAG-treated SSB1-dTAG with induced expression of wild-type or mutant SSB1 to determine whether these mutations affect SSB1 puncta formation in cells (Extended Data Fig. 20b).
7. R-loop IF to determine whether cancer-derived mutations of SSB1 affect its regulation of cellular R-loop levels (Extended Data Fig. 20c, d).
8. γ H2AX IF to determine whether cancer-derived mutations of SSB1 affect its regulation of genome stability (Extended Data Fig. 20e, f).

2) Experiments addressing comments about the role of SOSS-INTAC in R-loop resolution.

9. R-loop CUT&Tag in CTR and DKO cells (Fig. 4e).
10. Examining whether GFP-dRNH1- and S9.6-based R-loop measurements are confounded by ssRNA and dsRNA contamination (Extended Data Fig. 11c, d).
11. GFP-dRNH1-based R-loop IF in sgCtr and INTS2-KO cells (Extended Data Fig. 12a, b).
12. R-loop CUT&Tag in INTS11-depleted cells with rescue by expression of wild-type or catalytic-dead INTS11 (Extended Data Fig. 12e).
13. The generation and confirmation of an SSB1 mutant defective in interacting with INTAC to reveal the functional importance of the SSB1-INTAC interaction in resolving R-loops (Extended Data Fig. 12f).
14. Co-IP of SSB1 and SSB2 to determine whether they interact with RNase H1 (Review Fig. 1a, b).
15. Pulldown assays to confirm the loss of SSB1-RNase H1 interaction during evolution (Review Fig. 1c-f).

3) Experiments addressing comments about the contributions of SOSS-INTAC to RNAPII pausing.

16. Calculation of the pausing index after depleting SSB1 or INTS2 to corroborate the regulation of promoter-proximal Pol II pausing by SOSS-INTAC (Extended Data Fig. 6j, k).
17. ChIP-Rx of Pol II phosphorylated at CTD Serine 5 (pSer5) in CTR and DKO cells to confirm that SSB1 functions in pausing (Extended Data Fig. 7c).
18. Pol II ChIP-qPCR in dTAG-treated SSB1-dTAG with induced expression of wild-type or mutant SSB1 to determine whether these mutations affect SSB1's role in regulating Pol II occupancy at promoters (Extended Data Fig. 21a).
19. Metagene analysis of Pol II occupancy in CTR and DKO cells in different categories of genes (mono-exonic vs. multi-exonic, short vs. long, and protein-coding vs. non-coding) to demonstrate the regulation of transcription by SSB1 (Review Fig. 4a-c).
20. Quantification of the alterations of the pausing index for genes with different levels of pausing (Review Fig. 4d).

4) Experiments demonstrating mechanisms by which the endonucleolytic activity of INTAC leads to R-loop resolution.

21. Revealing the cooperation of the 5' exonuclease XRN2, 3' exonuclease exosome, and nuclear exosome-targeting (NEXT) complex with SOSS-INTAC in regulating R-loop levels (Extended Data Fig. 13a-c).
22. Measuring the occupancy of XRN2, the exosome, and the NEXT complex in CTR and DKO cells to determine whether SSB1 regulates their genomic binding (Extended Data Fig. 13d).

5) Experiments addressing comments about the formation of the SOSS-INTAC complex.

23. Gel filtration, Co-IP and ChIP experiments to determine whether SSB1 alone or SOSS forms a complex with INTAC (Fig. 1e, Extended Data Fig. 1d, Extended Data Fig. 2d).
24. ChIP-qPCR to verify the genomic binding of SOSS-INTAC subunits (Extended Data Fig. 2d).
25. The generation and confirmation of an SSB1 mutant that is impaired in ssDNA binding to reveal the functional importance of SSB1-mediated ssDNA recognition in the chromatin recruitment of SOSS-INTAC (Extended Data Fig. 5d, f).
26. The generation and validation of an SSB1 mutant defective in interacting with INTAC to uncover the functional importance of the SSB1-INTAC interaction for subsequent INTAC recruitment (Extended Data Fig. 5e, f).

6) Experiments addressing comments about the role of SOSS-INTAC in genome stability.

27. γ H2AX CUT&Tag in CTR and DKO cells to confirm the promoter-associated induction of γ H2AX upon SSB1 loss (Extended Data Fig. 14a).
28. γ H2AX CUT&Tag in sgCtr and INTS2-KO cells to confirm the promoter-associated induction of γ H2AX upon INTS2 depletion (Extended Data Fig. 14d).
29. Flow cytometry analysis following Propidium Iodide labeling and γ H2AX staining to reveal the induction of γ H2AX in both G1 and S phases upon SSB1 loss (Extended Data Fig. 14e).

7) Experiments addressing other comments.

30. EMSA assay to confirm that SSB1 preferentially binds to ssDNA over dsDNA or ssRNA (Extended Data Fig. 3a).
31. SSB2 Co-IP (Review Fig. 1b) and cell growth assay (Extended Data Fig. 4f) to identify a functional redundancy between SSB1 and SSB2.
32. ChIP-qPCR in cells with induced expression of RNase H1 to confirm the reduction of SSB1 at promoters (Extended Data Fig. 8c).
33. R-loop CUT&Tag in cells with induced expression of RNase H1 to confirm the specificity of R-loop signals and provide necessary controls for various R-loop-related experiments (Extended Data Fig. 8d, e).
34. ChIP-qPCR in cells with induced expression of RNase H1 to confirm the reduction of INTS3 (Extended Data Fig. 9b).
35. Determining whether GFP-dRNH1- and S9.6-based R-loop IF are confounded by ssRNA and dsRNA contaminations using the pretreatment of ssRNA endonuclease RNase T1 and dsRNA endonuclease RNase III (Extended Data Fig. 11c, d).
36. EMSA and Co-IP assays to determine whether cancer-derived mutations of SSB1 affect its interaction with ssDNA (Extended Data Fig. 18g) and INTAC (Extended Data Fig. 18h).
37. Pol II ChIP-Rx in CTR and DKO cells with induced expression of wild-type RNase H1 to elucidate whether RNase H1 can reverse the transcriptional defects in DKO cells (Review Fig. 2).
38. R-loop CUT&Tag in cells treated with CDK9/P-TEFb inhibitor Flavopiridol (FP) or XPB/TFIIH inhibitor Triptolide (TPL) to determine whether Pol II pausing regulates R-loop levels (Review Fig. 3a).
39. Transient transcriptome sequencing (TT-seq) in cells with induced expression of wild-type RNase H1 to reveal the impact of RNase H1 overexpression on nascent transcriptome (Review Fig. 3b).
40. Estimation of protein concentration of SSB1 foci in cells (Review Fig. 6).
41. In vitro condensation assay using an SSB1 mutant defective in tetramerization (Review Fig. 7).

In the following pages, please find our point-by-point response to all specific comments of the reviewers. The reviewers' comments are in **black**, and our responses to them are in **blue**. Please note that the comments from the reviewers cover several common issues and, as a result, some of the queries are similar. If questions overlap, we answered each one individually rather than referring the reviewer to another reply. Thus there is some redundancy among the answers below, but our intent is to make our responses to each question as clear as possible. Moreover, we have highlighted the changes described below in the revised manuscript and extended data figures in yellow to further facilitate review. The Review Figures and legends are attached at the end of the letter.

We sincerely thank all of the reviewers for their valuable and insightful comments and suggestions that helped us make substantial improvements to the revised manuscript.

Referees' comments:

Referee #1 (Remarks to the Author):

Xu and colleagues begin their manuscript by uncovering a novel interaction between a recently discovered complex INTAC (INTS1-14 + PP2A core enzymes) and the SOSS complex (SSB1, INIP, INTS3, the latter protein being common in the 2 complexes) and further examine the mechanisms of recruitment of this complex and its function. They show that SOSS-INTAC complex is located at promoters and enhancers (ChIP-seq SSB1, INTS3, INTS5) and that SSB1 is required for INTS3 and INTS5 recruitment (using a Double KO cell line for SSB1 and SSB2, “DKO”, to avoid potential SSB2 redundancy in the absence of SSB1). The manner SSB1 is recruited to chromatin is likely through its intrinsic ssDNA binding capacity (in vitro by EMSA and in vivo by correlation between SOSS-INTAC occupancy and ssDNA presence (KAS-seq). In line with the known INTAC function, the authors propose a role for SOSS-INTAC complex in stimulating early termination at paused sites by preventing aberrant accumulation of paused Polymerase II (here demonstrated in the SSB1/2 DKO context). Since R-loops structure contains a displaced ssDNA strand, the authors investigated relationship between SOSS-INTAC and R-loops. On the one hand they showed that R-loop presence is required for SSB1 and INTS3 binding at promoters, and on the other hand that SSB1 and INTS2 and 11 are also needed to avoid the aberrant presence of R-loops, which causes increased DNA damage and delayed replication fork progression. Lastly, the authors investigate the capacity of SSB1 to mediate its function via liquid-like condensate formation. In vitro, purified human SSB1 form droplets that mediates SOSS-INTAC condensate formation. The authors identified the domain and residues involved in condensation formation and that correspond to residues mutated in cancer. The authors further show that the condensation capacity of SSB1 is necessary to avoid unwanted R-loops levels in vivo. Altogether, the SOSS-INTAC complex is presented as a gatekeeper of genomic integrity by limiting aberrant R-loop formation at the transcriptional pause step through its capacity to form liquid droplets.

Overall, the paper reads very nicely, is presented in a coherent flow, and the data presented are of excellent quality with a wide range of complementary techniques. This work follows up on previously published data of the authors on the INTAC complex (2020) and greatly extends it by presenting novel insights into the manner in which this INTAC complex is recruited to chromatin. This study should be of interest for a wide readership since it establishes that the liquid condensation properties of SSB1, a protein 70x smaller than INTAC, enables INTAC recruitment to chromatin. Moreover, they show that that cancer-derived mutations abolish this condensation capacity and SSB1 intrinsic physical condensation properties are required for suppressing R-loop at promoters, a process important to avoid transcription-replication conflicts and undesired genomic instability. Overall, the authors connect RNA pol II pausing with genome stability maintenance. Some of the data presented in this study nevertheless need to be reinforced to better support the conclusions made (see comments below), mainly on the four main points of the paper, i.e., R-loop formation, Pol II pausing, genome instability, and LLPS.

We greatly appreciate and are encouraged that this reviewer found our work to be of excellent quality and of interest to a wide readership. Following this reviewer’s suggestions, we have conducted a large number of additional experiments and analyses that we have summarized above in our “General response to reviewers”, and which we believe have led to a substantially improved manuscript. Below, please see our point-by-point response to this reviewer’s comments and criticisms.

MAJOR COMMENTS

1) The authors show convincing evidence of IP-MS in nuclear extracts that SSB1 and INIP can independently interact with the INTAC complex and that GST-pull down of purified SSB1 also precipitates reconstituted INTAC and purified INIP. The evidence suggesting that SOSS-INTAC form a super complex only comes from gel filtration experiments, but importantly lack the INIP panel. Thus, at this stage the

evidence for the presence of this SOSS-INTAC super complex at promoter/enhancers is weak as one could easily argue that it is SSB1 alone, and not as part of the SOSS, that forms this super complex with INTAC. To be able to keep this statement “SOSS-INTAC” throughout the text, the authors need to reinforce their data (if not this should be modified to SSB1-INTAC complex). This could be done by showing INIP panel in gel filtration, or/and by performing endogenous INIP ChIP-qPCR to verify that its presence at few target SOSS-INTAC sites.

The reviewer raises an important point regarding whether SSB1 alone or the larger SOSS complex associates with INTAC. As suggested, we conducted INIP western blotting of gel filtration samples. The results show that, similar to SSB1, INIP co-migrates with INTAC subunits (Fig. 1e).

Fig. 1e. Gradient centrifugation using endogenous HEK Expi293 nuclear extracts. The fractionated samples were subjected to SDS-PAGE followed by western blotting.

To further confirm the incorporation of INIP in the SOSS-INTAC complex, we overexpressed Flag-tagged INIP in cells and performed Co-IP using Flag antibody. Western blotting results indicate that INIP interacts with INTAC subunits, supporting our conclusion that SOSS and INTAC can physically interact with each other (Extended Data Fig. 1d).

Extended Data Fig. 1d. Flag IP in cells stably expressing Flag-tagged INIP followed by western blotting. IgG was used as a control for non-specific binding.

We have tested several commercially available INIP antibodies, but unfortunately, none of them were suitable for IP or ChIP experiment. As an alternative approach to interrogate the chromatin occupancy of INIP, we conducted ChIP using Flag antibody in cells stably expressing Flag-tagged INIP. Our data suggest that INIP is enriched at promoters bound by SSB1 and INTAC (Extended Data Fig. 2d).

Extended Data Fig. 2d. ChIP-qPCR experiments using SSB1 (red), INTS3 (blue) and INTS5 (purple) antibodies in DLD-1 cells. Flag ChIP-qPCR was conducted in DLD-1 cells with overexpression of Flag-tagged INIP. Values are mean \pm SD (n = 3).

2) To show that the genomic instability observed with γ H2AX in DKO cells is directly connected with the role of SSB1 at promoters/enhancers and thus with Pol II pausing, the authors could perform γ H2AX by ChIP (or ChIP-seq) to address whether damage occurs at promoters/enhancer. In other words, is genomic instability caused at positions of SOSS-INTAC upon its depletion? Moreover, to differentiate whether it is mainly caused by replication conflicts, the authors could assess whether γ H2AX presence is only visible in S-phase cells. If not, the authors should slightly modify their statement.

We thank the reviewer for this suggestion. We measured genome-wide occupancy of γ H2AX in both CTR and DKO cells and observed a pervasive elevation of γ H2AX at active promoters in DKO cells (Extended Data Fig. 14a). Consistently, INTS2 knockout induces a comparable increase in γ H2AX levels at promoters (Extended Data Fig. 14d).

Extended Data Fig. 14a (left). Heatmaps showing γ H2AX occupancy in CTR and DKO cells. The peaks were centered on TSS of SOSS-INTAC target genes.

Extended Data Fig. 14d (right). Heatmaps showing γ H2AX occupancy in sgCtr and INTS2-KO cells. The peaks were centered on TSS of SOSS-INTAC target genes.

To evaluate the induction of γ H2AX at different cell cycle stages as suggested, we labeled the CTR and DKO with Propidium Iodide followed by γ H2AX staining and flow cytometry analysis. Our results indicate that the elevation of γ H2AX levels in DKO cells is present in both G1 and S phases, highlighting the broad impact of SOSS-INTAC on genome stability (Extended Data Fig. 14e).

e

Extended Data Fig. 14e. Flow cytometry analysis following Propidium Iodide labeling and γ H2AX staining in CTR and DKO cells. Propidium Iodide signal was used to separate cells into G1, S, and G2/M phases. Values are mean \pm SD (n = 3).

We have modified our statement, as indicated in the excerpt below:

Results:

“Flow cytometry analysis following Propidium Iodide labeling and γ H2AX staining revealed the induction of γ H2AX in both G1 and S phases (Extended Data Fig. 14e). We thus posited that SSB1 loss induces genome instability in part through impeding replication fork progression.”

3) Regarding the link with R-loops. Here are several suggestions

a. Figure 4: Analyses of R-loops by imaging is not appropriate without all the necessary controls (which are missing here) (see Smolka et al, J Cell Biol, 2020, PMID: 33830170 “S9.6 is subject to pervasive artifacts without pretreatments and controls that mitigate its promiscuous recognition of cellular RNAs” or Chedin et al, EMBO J, 2021, PMID 33411340 “S9.6 IF microscopy approaches should probably be avoided whenever possible. We also encourage investigators to make efforts to eliminate other confounding variables such as modifications to the nascent transcriptome. It is important to note that those recommendations apply equally if imaging were to be performed using dRNaseH1-based reagents instead of the S9.6 antibody”). R-loop IF data should be interpreted with extreme caution. S9.6 Cut&Tag could come to complement this analysis in DKO (as performed with SSB1 dTag Figure 6).

We fully agree with the reviewer that R-loop imaging data should be interpreted with extreme caution, which we had initially attempted to address by employing both S9.6- and GFP-dRNH1-based R-loop assays. Following the reviewer’s suggestion, we have performed experiments to exclude the possible confounding effects caused by single-stranded or double-stranded RNA contamination during R-loop measurements. Specifically, we compared R-loop IF with and without pretreatment of ssRNA endonuclease RNase T1 and dsRNA endonuclease RNase III (Chedin et al., 2021; Crossley et al., 2021; Smolka et al., 2021). Our results suggest that, although the pretreatment greatly eliminates the signals detected by S9.6, it has no discernable impact on the signals gauged by the GFP-dRNH1, suggesting that the R-loop signal we measured with GFP-dRNH1 is specific and unlikely to arise from ssRNA and dsRNA contamination (Extended Data Fig. 11c, d). We therefore used the GFP-dRNH1-based R-loop IF to quantify cellular R-loop levels and to complement our R-loop CUT&Tag results.

Extended Data Fig. 11c-d. S9.6-based (c) and GFP-dRNH1 (d) R-loop IF pretreated with ssRNA endonuclease RNase T1 and dsRNA endonuclease RNase III.

Following the reviewer's suggestion, we conducted R-loop CUT&Tag in CTR and DKO to confirm the accumulation of R-loops in DKO as observed by immunofluorescence. Indeed, DKO cells exhibited higher levels of R-loops than CTR cells (Fig. 4e), which is consistent with the genome-wide increase in R-loop levels resulting from SSB1 degradation in SSB1-dTAG cells.

Fig. 4e. Heatmaps of R-loop CUT&Tag signals over 6 kb regions centered on TSS of SOSS-INTAC target genes in CTR and DKO cells. CTR cells were treated with RNase H1 during CUT&Tag (4th lane) or incubated with IgG but not S9.6 (5th lane) to confirm the specificity of detected R-loop signals.

b. For the Cut&Tag the IgG control should be included. It is important to have this control since Cut&Tag has a preference for open regions such as those detected here (promoter/enhancer). It is thus critical to determine what is the real R-loop Cut&Tag profile from unspecific tagmentation in open regions.

We thank the reviewer for raising this important point. As requested, we have included IgG controls for all R-loop CUT&Tag experiments (Fig. 4e, 6g, Extended Data Fig. 8e, 12e, 12f, 13c-d, 14a, 14d).

4) The authors interpret increased Pol II occupancy and increased PRO-seq at promoter and 5' end, as a sign of Pol II pausing. This would need to be strengthened by both experiments and analysis. For instance, Pol II pausing could be addressed using antibodies against the phosphorylated Serine 2 form of RNA pol II. Moreover, to differentiate pausing from an increase in transcriptional, the authors could compute pausing index (ratio of Pol II occupancy on gene body over promoter regions).

As the reviewer suggested, we conducted ChIP-Rx using an antibody against the phosphorylated Pol II at Serine 5 (pSer5), which was used to define Pol II pausing sites (Chen et al., 2015; Liu et al., 2015; Rahl et al., 2010). In line an increase of total Pol II levels at promoters, pSer5 levels were substantially induced upon the loss of SSB1 in DKO cells (Extended Data Fig. 7c).

Extended Data Fig. 7c. Heatmaps showing the occupancy of Pol II phosphorylated at CTD Serine 5 (pSer5) on SOSS-INTAC target genes in CTR and DKO cells. The peaks are centered on TSS and ranked by decreasing occupancy in CTR cells.

Moreover, we calculated the pausing index by comparing the ratio of Pol II at promoters versus gene bodies as suggested. As shown below, the pausing index is evidently increased upon the loss of SSB1 or INTS2, suggesting increased Pol II pausing under these conditions (Extended Data Fig. 6j, k).

Extended Data Fig. 6j-k. Empirical cumulative distribution function plot of the pausing index (PI) distribution in CTR and DKO (j), or in sgCtr and INTS2-KO cells (k).

5) While the author presents clear evidence for SSB1 condensate formation *in vitro*, not much is analyzed *in vivo*. The authors could analyze whether SSB1 mutants that do not display the ability to form droplets *in vitro*, also abolish foci formation *in vivo* (as in Fig. 5c).

We thank the reviewer for this important suggestion. In the revised manuscript, we have strengthened our conclusions regarding condensate formation of SSB1 or SOSS-INTAC using *in vivo* experiments that are summarized below:

(1) To determine the interdependency of SSB1 and INTAC in puncta formation in cells, we first depleted SSB1 and SSB2 and conducted IF of endogenous INTAC subunits. Notably, loss of SSB1/2 abolishes

puncta formation of INTAC subunits (Extended Data Fig. 16a). However, induced degradation of INTS11 exerted no noticeable impact on the formation of SSB1 puncta (Extended Data Fig. 16b, c). Supported by *in vitro* analysis of SOSS-INTAC (Fig. 5k, l, Extended Data Fig. 17h, i), we conclude that SSB1 drives the puncta formation of INTAC in cells, but not vice versa.

Extended Data Fig. 16. (a) The immunofluorescent images of SSB1 (red) and INTAC subunits (green) in wild-type and DKO cells. (b) The establishment of INTS11-dTAG cells and verification by dTAG treatment.

(c) The immunofluorescent images of SSB1 (red) and INTS11 (green) in DMSO- or dTAG-treated INTS11-dTAG cells.

(2) To analyze the puncta formation of SSB1 mutants in vivo, we induced the expression of wild-type or mutant SSB1 in dTAG-treated SSB1-dTAG cells. Consistent with the condensation results in vitro (Fig. 6b, c), the puncta formation capacity of Δ IDR and RY is abolished and that of SI and R206Q is severely impaired compared with wild-type SSB1 (Extended Data Fig. 20b).

Extended Data Fig. 20b. Representative images of SSB1 immunofluorescent signals in dTAG-treated SSB1-dTAG cells with overexpression of wild-type or mutant SSB1.

(3) To examine whether SSB1 can form liquid-like condensates in cells, we used the “optoDroplet” system by fusing SSB1 with mCherry-labeled Arabidopsis photoreceptor cryptochrome 2 (CRY2) (Kilic et al., 2019; Shin et al., 2017; Taslimi et al., 2014). This reveals that the droplet formation of SSB1, but not the control, is substantially boosted upon light induction (Extended Data Fig. 17c). Moreover, SSB1 puncta undergo frequent fusion and fission events (Extended Data Fig. 17d, e). We also find that the fluorescent signals of foci recover readily after photobleaching (Extended Data Fig. 17f), which is indicative of droplet-like behavior.

Extended Data Fig. 17c-f. (c) The establishment of the “optoDroplet” system by fusing SSB1 with mCherry-labeled Arabidopsis photoreceptor cryptochrome 2 (CRY2). Representative images of SSB1 and control are shown before and after light induction. (d-e) Time-lapse imaging demonstrating spontaneous fusions (d) and fissions (e), as indicated by the arrows, of SSB1 condensates in cells. (f) Representative micrographs of SSB1 puncta before and after photobleaching.

MINOR COMMENTS

1) *In abstract the authors say that mutation in SSB1 IDR provokes pervasive accumulation of R-loops at “a genome-wide level”. This refers to Figure 6 m and n, where only a few genes were tested (those bound by SOSS-INTAC). Therefore, the statement at a “genome-wide level” is not appropriate and should be accordingly modified.*

We apologize for the inaccuracy of the statement. We have removed “genome-wide level” in the abstract.

2) *In the abstract, perhaps change “early termination of transcription” to “early termination of transcription at Pol II paused sites”, from that sentence, it seems that the paper will be about what happens at TTS rather than at TSS. Same hold true for “timely transcriptional termination”, change to “timely transcriptional termination at Pol II paused sites”*

We thank the reviewer for the great suggestion. To differentiate the early termination at Pol II paused sites from the normal termination at TTS, we now describe it as “early termination of transcription at Pol II paused sites” or “promoter-proximal termination” as the reviewer suggested.

3) *Fig SIC legend says that SSB1, INIP and INTAC subunits were overexpressed. Yet, there is no panel for INIP. And in the text, there is no mention of INIP overexpression. Please adjust accordingly.*

We apologize for the mistake. As we described in the text, only SSB1 and all INTAC subunits, but not INIP, were overexpressed. We have corrected the figure legends.

4) *In Figures 1g, 2b, 5a, S2c and S3a please do not use the SOSS term, but rather SSB1 (see major comment above).*

We thank the reviewer for the comments. For the sake of being accurate as suggested, we now use “SSB1” instead for Fig. 2b, Fig. 5a, and Extended Data Fig. 3c, d (originally S3a). Given that the newly added data confirmed the co-migration of INIP with SSB1 and INTAC by gel filtration (Fig. 1e), the interaction of INIP with INTAC (Extended Data Fig. 1d), and the enrichment of INIP on SSB1/INTAC-bound promoters (Extended Data Fig. 2d), we kindly ask the reviewer to let us stick to “SOSS” term for Fig. 1f (originally Fig. 1g), Fig. 1g (originally Fig. S2c).

5) *Genome browser examples of panel displaying the KAS-seq, and CHIP seq of SSB1, INTS3 and 5 should be shown. Otherwise, there is not insight into the quality of KAS-seq data, nor its resolution profile which could in theory affect the correlation coefficient.*

We agree with this point and have now added the KAS-seq track examples as suggested (Extended Data Fig. 3b).

Extended Data Fig. 3b. Representative browser tracks showing KAS-seq signals compared with the genomic occupancy of SOSS-INTAC subunits.

6) It is confusing for reader for the single SSB2 KO to be called control. Please just name it SSB2 KO.

We agree about the potential for confusion about our use of “CTR” to represent SSB2 KO cells and we are amenable to any suggested changes to the text that help readers better understand our study. Indeed, this very issue had been much debated among our authors, with various suggestions, such as the use of “CTR(SSB2-KO)” or “CTR^{ΔSSB2}” instead of CTR. The consensus decision of all of the authors was to use CTR, and we have kept this nomenclature for the revised manuscript. However, we have tried to clarify our definition of CTR in additional places in the text and figure legends to minimize any possibility of confusion. Aside from these textual concerns, during the revision process we have obtained additional biochemical and biological insights into the potential redundancy of SSB1 and SSB2. As shown by proteomics (Extended Data Fig. 1c) and Co-IP experiments (Review Fig. 1b), we have confirmed that SSB2 also interacts with the INTAC complex. Moreover, we found that induced expression of either SSB1 or SSB2 rescues the growth defects in DKO cells, revealing the redundancy of these two paralogs (Extended Data Fig. 4f). We have also added a western blot panel for SSB2 in Fig. 2f to further clarify that “CTR” cells lack SSB2 protein.

c

		IP:	Empty	SSB1	SSB2	INIP
SOSS	SSB1		0	273840000	852520	17760000
	SSB2		0	0	1853500000	104750
	INIP		0	42252000	59843000	25135000000
INTAC	INTS1		0	5323100	1167700	1683500
	INTS2		0	2415500	1097400	802220
	INTS3		32938	32919000	27619000	14708000
	INTS4		0	4979300	1601300	2851100
	INTS5		0	5258400	1420700	1424600
	INTS6		0	4455200	984270	1628500
	INTS7		0	5953000	1991700	1864400
	INTS8		0	3451200	967020	904530
	INTS9		0	4118800	2740100	1701200
	INTS10		0	4311300	1235400	712390
	INTS11		796810	8124700	4270500	2675200
	INTS12		0	6709300	2174800	5149900
	INTS13		0	5088200	1633900	1854400
	INTS14		0	3817100	0	0
	PP2A-A		21400	16122110	17100000	9556090

Extended Data Fig. 1c. Mass spectrometry analyses of Protein A-tagged SSB1, SSB2 and INIP immunoprecipitation. The values are iBAQ intensity for SOSS, INTAC and Pol II subunits. IgG was used as the binding control.

Review Fig. 1b (left panel). V5 Co-IP in cells stably expressing V5-tagged SSB2, followed by western blotting of INTAC subunits and RNase H1.

Extended Data Fig. 4f (right panel): Growth curves of CTR and DKO cells with or without overexpression of SSB1 or SSB2. Data are mean ± SD from 4 independent experiments.

7) In Figure 2f, please show the western blot panel for SSB2.

We have added the western blot panel for SSB2 in Fig. 2f as suggested by the reviewer.

Fig. 2f. Western blotting of whole-cell extracts from CTR (control, SSB2 knockout) and DKO (SSB1 and SSB2 double knockout) DLD-1 cells.

8) In Fig 5, it is unclear whether in the text of Figure legend whether it is all INTAC subunits that are used in the condensate experiment, or a subset or one. Please modify accordingly.

We apologize for the lack for clarity. The purified INTAC protein used in Fig. 5 and Fig. 6 contains all subunits as Fig. 1c. We have now modified the figure legend as suggested by the reviewer.

Referee #2 (Remarks to the Author):

Multisubunit Integrator-PP2A complex [INTAC] is emerging as a major player in regulation of RNA polymerase II transcription of non-coding and protein coding genes. Endonucleolytic cleavage of the nascent transcript by the Int11 subunit of the INTAC is involved in generating the 3' end of snRNAs, regulation of the expression of protein coding genes via premature termination, quality control by preventing poorly configured Pol II complexes from entering elongation and suppression of the unwanted non-coding transcription. How INTAC is recruited to Pol II to execute its functions in transcription represents one of the key unanswered questions in the field.

In this manuscript, Xu et al., demonstrate that single stranded DNA binding protein SSB1 that was previously demonstrated to interact with Int3 subunit of the Integrator forms a stable complex with the entire INTAC [SOSS-INTAC]. SSB1 shows similar genome wide occupancy to the Integrator subunits and coincides with positions of the ssDNA mapped by KAS-seq. Furthermore, occupancy of the INTAC at the promoter and enhancer regions is reduced upon SSB1/SSB2 KO [DKO]. Loss of either INTAC subunits, Int3 or Int5 leads to increase in the levels of the promoter proximal Pol II based on Pol II ChIP-seq and PRO-seq as well as nascent transcriptome analyses by TT-seq in agreement with the previous reports. Loss of SSB protein shows similar effect on Pol II and nascent transcriptome further supporting role of SSB in mediating recruitment and function of the INTAC. Depletion of these factors [Int3, Int5, SSB] is also associated with increased levels of RNase H sensitive R-loop structures in nuclei [observed by IF]. dTAG depletion of SSB1 also leads to accumulation of R-loops at the promoters based on R-loop genome wide mapping. Accumulation of R-loops upon SSB1/2 is associated with genomic instability and replication defects based on gamma H2AX levels and DNA fiber assay. Ints2 KO also leads to increase in gamma H2AX suggesting that it is likely that the entire SOSS-INTAC functions to maintain genome stability. Interestingly, endonucleolytic activity of the INTAC is required for regulation of R loops. On the other hand, R-loops are required to mediate recruitment of SOSS-INTAC.

Finally, authors propose that SSB1 mediates formation of SOSS-INTAC condensates that can be disrupted by cancer associated mutations in IDR of SSB1. Mutations in IDR of SSB1 also show some increase in R loop levels leading to hypothesis that SSB1 suppresses R-loop formation by mediating formation of liquid-like condensates.

Although a link between SSB 1/Ints3 [doi: 10.1074/jbc.M109.039404] as well as interaction of SSB1 with multiple subunits of the Integrator [doi: 10.1083/jcb.200907026] have been previously reported, this work convincingly demonstrates formation of stable SOSS-INTAC complex, establishes functional connection between SSB, INTAC and R-loops providing insights into how INTAC is recruited during transcription and regulation of R-loops and genome stability by the INTAC, which are interesting novel findings. However, some aspects of the analyses are lacking depths and conclusions can be further strengthened by additional mechanistic insights. The weakest part of the manuscript in my opinion is the part related to condensate formation mediated by SSB1 and its role, I am not convinced whether this indeed has any functionality. I therefore cannot recommend this manuscript for publication in its current form.

We greatly appreciate the reviewer for the positive feedback and for highlighting our interesting novel findings. Following the reviewer's suggestions, we have conducted a series of additional experiments and provide more in-depth bioinformatic analysis to strengthen our conclusions (summarized in "General response to reviewers"). Please see our point-by-point response below:

Specific comments:

-The observation that the endonucleolytic activity of the Ints11 is required for suppression of R-loops is very interesting and is worth exploring further. How does endonucleolytic cleavage could affect removal of the R-loops? Does this provides an entry for 3' and 5' exonucleases? Which ones?

We thank the reviewer for raising this critical point. The major 5' and 3' exonucleases responsible for RNA degradation in the nucleus are thought to be XRN2 and the exosome complex. As the reviewer suggested, we examined whether either or both contribute to R-loop attenuation. We thus depleted XRN2, two catalytic subunits of the exosome (DIS3 and EXOSC10), and the helicase subunit of the nuclear exosome-targeting (NEXT) complex MTR4 that unwinds structured RNA substrates for exosomal degradation (Extended Data Fig. 13a, b). As shown by R-loop CUT&Tag-qPCR on example genes, depletion of XRN2, DIS3 and MTR4 induces a slight but significant upregulation of R-loops at promoters in CTR cells. Simultaneous loss of XRN2 and DIS3 leads to a greater increase in R-loops, indicating that both XRN2 and the exosome contribute to R-loop attenuation (Extended Data Fig. 13c, gray). However, although the loss of SSB1 in DKO cells leads to an apparent upregulation of R-loops, additional disruption of XRN2 and the exosome fails to augment this observed change (Extended Data Fig. 13c, purple). Therefore, it is tempting to speculate that SOSS-INTAC loss-induced R-loop accumulation is epistatic to that caused by disrupting XRN2 and the exosome, which could potentially be explained by the notion that the endonucleolytic cleavage of RNA by SOSS-INTAC is required to create exposed 5' and 3' end for exonucleolytic digestion by XRN2 and/or the exosome. Furthermore, we examined whether the recruitment of XRN2 and the exosome subunits are regulated by SOSS-INTAC. Notably, the occupancy of XRN2, but not subunits of the exosome or the NEXT complex, at promoters is evidently compromised in DKO cells (Extended Data Fig. 13d).

Extended Data Fig. 13. (a-b) Quantitative reverse transcription PCR (RT-qPCR) (a) and western blotting (b) to determine the knockdown efficiency of XRN2, DIS3, EXOSC10, and MTR4. (c) R-loop CUT&Tag in CTR and DKO cells with knockdown of XRN2, DIS3, EXOSC10, and MTR4. Values are mean \pm SD (n = 3). (d) Heatmaps showing the occupancy of XRN2, DIS3, EXOSC10, and MTR4 in CTR and DKO cells.

-Does increase in R loops levels contribute to transcription? Can defects in transcription be reversed by over-expression of RNase H?

We thank the reviewer for this interesting question. As requested, we conducted Pol II ChIP-Rx in CTR and DKO cells with and without induced expression of RNase H1. Our results show that the accumulation of Pol II at promoters in DKO cells is rescued by RNase H overexpression (Review Fig. 2). We thus

conclude that RNase H-mediated R-loop attenuation can at least partially reverse the transcriptional defects resulting from SOSS-INTAC disruption.

Review Fig. 2. (a-b) Heatmaps (a) and metaplots (b) showing Pol II ChIP-Rx signals over 6 kb regions centered on TSS of SOSS-INTAC target genes ranked by decreasing occupancy in cells with DOX-inducible RNase H1 expression.

-Are R-loop a result of Pol II pausing? How does overexpression of RNase H affect nascent transcriptome?

In response to this interesting question, we conducted R-loop CUT&Tag in cells treated with either CDK9/P-TEFb inhibitor flavopiridol (FP), which blocks Pol II release to stimulate pausing, or XPB/TFIIH inhibitor triptolide (TPL), which inhibits accumulation of promoter-proximally paused Pol II by preventing open complex formation. As shown by heatmap analysis, FP treatment induces R-loop accumulation while TPL treatment attenuates R-loop levels at promoters (Review Fig. 3a). Therefore, this suggests that Pol II pausing contributes to R-loop formation at promoters.

To determine how overexpression of RNase H affects nascent transcriptome as suggested, we conducted transient transcriptome sequencing (TT-seq) to quantify the nascent transcriptome in cells with induced expression of RNase H. We observed a greater number of genes that are downregulated than upregulated (Review Fig. 3b), supporting a major role for R-loops in modulating gene expression.

Review Fig. 3. (a) Heatmaps showing R-loop CUT&Tag signals centered on TSS in cells treated with flavopiridol (FP) or triptolide (TPL). (b) MA plot of differentially expressed genes by comparing DMSO- and DOX-treated cells with inducible expression of RNase H1.

-dTAG depletion is very long [12-24h]-could it be the reason for such a strong global increase in R loop signal genome wide [Fig 6I] due to pleiotropic effects?

As suggested, we performed R-loop CUT&Tag in SSB1-dTAG with 6 hours of dTAG treatment. Albeit to a lesser extent than the observed change after 24-hour dTAG treatment, 6 hours of dTAG-mediated SSB1 degradation already induces a substantial increase in R-loop levels genome-wide (Fig. 6g, h). We thus speculate that SOSS-INTAC pervasively regulates R-loop levels at promoters.

Fig. 6.g-h. (g) Heatmaps of R-loop CUT&Tag signals over 6 kb regions centered on TSS of SOSS-INTAC target genes in SSB1-dTAG cells with time-course dTAG treatment. One sample was treated with RNase H1 protein during CUT&Tag to verify the specificity of R-loop signals. (h) Representative browser tracks showing the R-loop signals in SSB1-dTAG cells with time-course dTAG treatment.

-I think that section related to investigation of the functional contribution of SOSS-INTAC to Pol II transcription [page 9] is a bit sketchy and would benefit from further analyses and more careful interpretations of the data. It was recently reported by the Torben Jensen's lab [Lykke –Andersen et al., 2021] and others that depletion of the Integrator subunits leads to different effect on multi-exonic vs mono-exonic genes. Contribution of the Integrator to transcription is not limited to facilitation of premature termination, it also controls transcriptional output from promoters and prevents premature termination of transcription. For example, multi-exonic genes show increased number of reads at the promoter and acute decrease within the gene body suggesting that Integrator prevents premature termination at these genes. In contrast, monoexonic genes show increase in the promoter proximal transcription as well as gene body signal suggesting that Integrator elicit premature terminations for this group of TUs. Authors should perform more thorough analyses of the data with this in mind and integrate this with the R-loop data.

We thank the reviewer for raising this important point. As the reviewer points out, Dr. Torben Jensen's lab found that the depletion of the Integrator subunits exerts different effects on mono- and multi-exonic genes (Lykke-Andersen et al., 2021). We thus separated genes into mono- and multi-exonic groups followed by the analysis of Pol II profiles in CTR and DKO cells. As shown below, although only mono-exonic genes exhibit a substantial elevation of Pol II occupancy in gene bodies, both groups show increased Pol II levels at promoters in DKO cells (Review Fig. 4a). These results are in line with the observation reported by Dr. Jensen's group. Moreover, the pervasive accumulation of promoter-proximal Pol II is consistent with the broad changes in R-loop formation at promoters resulting from loss of SSB1 in DKO cells.

Review Fig. 4a. Metagene analysis of Pol II occupancy in CTR and DKO cells for mono-exonic vs. multi-exonic genes.

Support for a genome-wide role for Integrator was recently provided by a study from Dr. Karen Adelman’s group, where they found that “Integrator endonuclease drives promoter-proximal termination at **all** RNA polymerase II-transcribed loci” (Stein et al., 2022). This is also in accordance with the findings from Torben Jensen’s group and by our studies that most genes, whether mono- or multi-exonic, show an elevation of promoter-proximal Pol II levels following Integrator or SSB1 disruption. Moreover, Adelman and colleagues suggested that shorter genes tend to have greater upregulation of Pol II within gene bodies due to longer genes exhibiting more severe defects in progressive elongation after INTS11 depletion. To determine whether SSB1 depletion results in similar changes as Integrator disruption, we separated genes into quartiles by length and analyzed the Pol II profiles in CTR and DKO cells (Review Fig. 4b). Our results indicate that the shorter genes have greater elevation of Pol II levels within gene bodies in DKO cells, which is in agreement with the observation from Dr. Adelman’s group that INTS11 loss tends to preferentially upregulate shorter genes.

Review Fig. 4b. Metagene analysis of Pol II occupancy in CTR and DKO cells for quartiles of genes classified by gene length.

Collectively, we conclude that disruption of either SSB1 or Integrator/INTAC is sufficient to induce a genome-wide accumulation of Pol II at promoters, while the changes in Pol II profiles within gene bodies can vary according to various gene properties. Specifically, short and mono-exonic genes are more prone to be upregulated compared with long and multi-exonic genes. Importantly, since SSB1 depletion induces a widespread R-loop accumulation at promoter-proximal regions, our results point toward a unified model of how SOSS-INTAC attenuates the accumulation of Pol II and R-loops at promoters.

-I find functional aspects of the condensate formation very unconvincing and this should be downplayed. Additionally, the evidence for condensates formation is limited to in vitro experiments and these were shown to form only in low salt which is not physiological, Fig5d. On the other hand, potential contribution of cancer associated mutation to R-loops accumulation is interesting. How do these mutations affect transcription? Is SSB interaction with the Integrator affected? Is DNA binding compromised by these mutations? It would be good to perform genome wide mapping of R loops in these mutants and analyse

links with the DNA damage/replication. Line 321- I would rephrase it as the notion of the involvement in disease development as there is no data to show that this drives cancer, this can be discussed but not in the result section.

We thank the reviewer for the constructive advice. To address these questions, we first conducted several in vivo data experiments to confirm the condensation capacity of SOSS-INTAC as described below:

(1) To determine the interdependency of SSB1 and INTAC in puncta formation in cells, we first depleted SSB1 and SSB2 and conducted IF of endogenous INTAC subunits. Notably, loss of SSB1/2 abolishes puncta formation of INTAC subunits (Extended Data Fig. 16a). However, induced degradation of INTS11 exerted no noticeable impact on the formation of SSB1 puncta (Extended Data Fig. 16b, c). Supported by in vitro analysis of SOSS-INTAC (Fig. 5k, l, Extended Data Fig. 17h, i), we conclude that SSB1 drives the puncta formation of INTAC in cells, but not vice versa.

Extended Data Fig. 16. Extended Data Fig. 16. (a) The immunofluorescent images of SSB1 (red) and INTAC subunits (green) in wild-type and DKO cells. (b) The establishment of INTS11-dTAG cells and verification by dTAG treatment. (c) The immunofluorescent images of SSB1 (red) and INTS11 (green) in DMSO- or dTAG-treated INTS11-dTAG cells.

(2) To analyze the puncta formation of SSB1 mutants in vivo, we induced the expression of wild-type or mutant SSB1 in dTAG-treated SSB1-dTAG cells. Consistent with the condensation results in vitro (Fig. 6b, c), the puncta formation capacity of Δ IDR and RY is abolished and that of SI and R206Q is severely impaired compared with wild-type SSB1 (Extended Data Fig. 20b).

Extended Data Fig. 20b. Extended Data Fig. 20b. Representative images of SSB1 immunofluorescent signals in dTAG-treated SSB1-dTAG cells with overexpression of wild-type or mutant SSB1.

(3) To examine whether SSB1 can form liquid-like condensates in cells, we used the “optoDroplet” system by fusing SSB1 with mCherry-labeled Arabidopsis photoreceptor cryptochrome 2 (CRY2) (Kilic et al., 2019; Shin et al., 2017; Taslimi et al., 2014). This reveals that the droplet formation of SSB1, but not the control, is substantially boosted upon light induction (Extended Data Fig. 17c). Moreover, SSB1 puncta undergo frequent fusion and fission events (Extended Data Fig. 17d, e). We also find that the fluorescent signals of foci recover readily after photobleaching (Extended Data Fig. 17f), which is indicative of droplet-like behavior.

Extended Data Fig. 17c-f. (c) The establishment of the “optoDroplet” system by fusing SSB1 with mCherry-labeled Arabidopsis photoreceptor cryptochrome 2 (CRY2). Representative images of SSB1 and control are shown before and after light induction. (d-e) Time-lapse imaging demonstrating spontaneous fusions (d) and fissions (e), as indicated by the arrows, of SSB1 condensates in cells. (f) Representative micrographs of SSB1 puncta before and after photobleaching.

To further investigate the contribution of cancer-associated mutations in SSB1 to R-loop formation, we determined whether these mutations affect the ability of SSB1 to recognizing ssDNA and/or INTAC. As shown by EMSA and Co-IP, the cancer-associated mutations impair neither ssDNA recognition nor interaction with INTAC (Extended Data Fig. 18g, h). This is in stark contrast to our findings using mutants bearing compromised ssDNA binding (W55A/F78A) or disruption of the interaction of SSB1 with INTAC(E97A/F98A) (Extended Data Fig. 5d, e).

Extended Data Fig. 18g-h. (g) EMSA assays using Cy3-labeled oligo (dT)48 incubated with wild-type SSB1, S172P/H173L, or R206Q. (h) V5 Co-IP in cells overexpressed with V5-tagged wild-type SSB1, W55A/F78A, or E97A/F98A followed by western blotting of SOSS-INTAC subunits.

Extended Data Fig. 5d-e. (d) EMSA assays using Cy3-labeled oligo (dT)48 incubated with purified wild-type SSB1, W55A/F78A (the mutant defective in binding ssDNA), or E97A/F98A (the mutant defective in interacting with INTS3). (e) V5 Co-IP in cells overexpressed with V5-tagged wild-type SSB1, W55A/F78A, or E97A/F98A.

We further examined whether the cancer-associated mutations affect cellular R-loop levels, genome stability and promoter-proximal termination as suggested. As shown by R-loop and γ H2AX IF, S172P/H173L, but not R206Q, is capable of attenuating cellular R-loop levels and maintaining genome stability in dTAG-treated SSB1-dTAG cells (Extended Data Fig. 20c-f). As shown by ChIP-qPCR, S172P/H173L, but not the R206Q substitution, restores the proper regulation of promoter-proximal termination (Extended Data Fig. 21a), which is consistent with their different roles in condensate formation and R-loop regulation.

Extended Data Fig. 20c-f. (c-d) R-loop IF in dTAG-treated SSB1-dTAG cells with overexpression of wild-type SSB1 or cancer-derived mutants (S172P/H173L and R206Q) (c). Quantification of the nuclear R-loop signals is shown in (d) (n = 150). (e-f) Immunostaining of γ H2AX signal in dTAG-treated SSB1-dTAG cells with overexpression of wild-type SSB1 or cancer-derived mutants (S172P/H173L and R206Q) (e). Quantification of the nuclear γ H2AX foci number is shown in (f). (n = 150).

Extended Data Fig. 21a. Pol II ChIP-qPCR in DMSO- or dTAG-treated SSB1-dTAG cells with overexpression of wild-type or mutant SSB1.

Moreover, and as suggested by the reviewer, we have moved the discussion regarding the potential involvement of cancer-associated mutations in disease development into the Discussion section, as indicated in the excerpt below:

Discussion:

“Moreover, cancer-derived mutations in SSB1 disrupting the condensation capacity of SOSS-INTAC also compromise its role in regulating R-loops and genome stability, which could potentially contribute to an oncogenic program.”

Also following the reviewer’s suggestions, we have toned down our conclusions about the potential biological significance of condensate formation on SOSS-INTAC function, as indicated in the excerpt below:

Discussion:

“However, it is important to note that the IDR of SSB1 could possess condensation-independent functions, such that mutations disrupting condensation may also introduce additional impacts yet to be identified. Meanwhile, non-IDR regions of SSB1 may also contribute to condensation through different mechanisms such as self-oligomerization and/or interactions with nucleic acids or proteins. Therefore, future studies are warranted to systematically investigate the biophysical properties of SOSS-INTAC and their contributions to transcription, R-loop regulation, and genome stability, and the degree to which the condensation ability of SOSS-INTAC contributes to these processes.”

- The targeting of SSB1 to ssDNA regions was proposed to represent a part of the mechanism for the INTAC recruitment. It would be informative to generate mutants defective in DNA binding to test whether DNA binding of SSB is important for the INTAC recruitment.

We agree with the reviewer that it would be informative to have an SSB1 mutant defective in DNA binding. Based on available structural information (Ren et al., 2014), we have generated a SSB1 mutant (W55A/F78A) that has compromised ssDNA recognition capacity but preserves its interaction with INTAC (Extended Data Fig. 5d, e). As expected, this SSB1 mutant exhibits impaired binding at the promoters of SOSS-INTAC targets, as analyzed by ChIP-qPCR (Extended Data Fig. 5f). Moreover, the recruitment of INTAC subunits INTS3 and INTS5 to these sites is also abolished in cells reconstituted with this SSB1 mutant, suggesting that DNA binding of SSB1 is important for the INTAC recruitment (Extended Data Fig. 5f).

Extended Data Fig. 5d-f. (d) EMSA assays using Cy3-labeled oligo (dT)48 incubated with purified wild-type SSB1, W55A/F78A (the mutant defective in binding ssDNA), or E97A/F98A (the mutant defective in interacting with INTS3). (e) V5 Co-IP in cells overexpressed with V5-tagged wild-type SSB1, W55A/F78A, or E97A/F98A. (f) ChIP-qPCR of SSB1, INTS3 and INTS5 in CTR and DKO cells with overexpression of wild-type SSB1, W55A/F78A, or E97A/F98A. Values are mean \pm SD (n = 3).

-Page 11, second paragraph-...S9.6 antibody potentially dsRNA...- change for :
 ...S9.6 antibody potentially can detect dsRNA...

We thank the reviewer for pointing out this mistake. We have corrected it as suggested.

-To account for changes in transcription, ChIP data should also be normalized for Pol II occupancy [fig 2j]

Following this suggestion, we used Pol II occupancy to normalize the ChIP data for SSB1, INTS3, and INTS5 in both CTR and DKO cells. Our results indicate that the occupancy of SSB1, INTS3, and INTS5 are pronouncedly decreased in DKO cells (Extended Data Fig. 6d-f).

Extended Data Fig. 6d-f. Boxplots showing the relative occupancies of SSB1 (d), INTS3 (e) and INTS5 (f) compared with Pol II in CTR and DKO cells. ****: $p < 0.0001$.

- Fig3c,d – metagenes for the whole gene would help to see changes in global transcription changes.

As suggested, we now show metagene analyses for gene groups divided by exon numbers (mono-exonic vs multi-exonic), protein-coding properties (protein-coding vs non-coding), and gene lengths to demonstrate the global transcription changes. As discussed above, most genes have increased levels of Pol II at promoters after disrupting SOSS-INTAC, while only mono-exonic, non-coding, and shorter genes exhibit a substantial elevation of Pol II occupancy within gene bodies (Review Fig. 4a-c).

Review Fig. 4a-c. (a-c) Metagene analysis of Pol II occupancy in CTR and DKO cells for mono-exonic vs. multi-exonic genes (a), quartiles of genes classified by gene length (b), and protein-coding vs. non-coding genes (c).

- EMSA assay -Fig2d (it should be explained what protein concentration was used), similar experiments with dsDNA or ssRNA would confirm the specificity of the interaction.

We have specified the protein concentrations used for all EMSA as requested. Moreover, and as suggested, we have compared the interaction of SSB1 with ssDNA, dsDNA, and ssRNA by EMSA, which reveals that SSB1 preferentially binds ssDNA (Extended Data Fig. 3a).

Extended Data Fig. 3a. EMSA assays using Cy3-labeled ssDNA, dsDNA and ssRNA incubated with SSB1.

- SSB2 was also shown to interact with INTS3 and therefore might be redundant, it should be made clear what is the interplay between SSB1/2.

We agree with the reviewer that SSB1 and SSB2 potentially have redundant roles. As shown by our proteomics data (Extended Data Fig. 1c) and Co-IP experiment (Review Fig. 1b), we find that SSB2 also interacts with the INTAC complex.

Extended Data Fig. 1c (left panel). Mass spectrometry analyses of Protein A-tagged SSB1, SSB2 and INIP immunoprecipitation. The values are iBAQ intensity for SOSS, INTAC and Pol II subunits. IgG was used as the binding control.

Review Fig. 1b (right panel). V5 Co-IP in cells stably expressing V5-tagged SSB2, followed by western blotting of INTAC subunits and RNase H1.

We also examined whether SSB1 and SSB2 could rescue the growth defects in DKO cells. Notably, induced expression of either SSB1 or SSB2 rescues the growth defects in DKO cells, revealing a redundancy in these paralogs (Extended Data Fig. 4f).

Extended Data Fig. 4f. Growth curves of CTR and DKO cells with or without overexpression of SSB1 or SSB2. Data are mean \pm SD from 4 independent experiments.

-Figure 4. Overexpressing RNaseH1 I would expect that the fork speed would go up or not change but it is down. Therefore not sure how to fully interpret this result. It would be good to have conditions when R-loop resolution is compromised and SSB1 is also depleted.

We are glad that the reviewer brought up this interesting point. Our observation that overexpression of RNaseH1 slightly slows replication fork speed recapitulates findings from previous studies as exemplified below:

Figure 2b from Bayona-Feliu et al. (Bayona-Feliu et al., 2021). (Please compare 1st and 2nd lanes)

Figure 1f from Prendergast et al. (Prendergast et al., 2020). (Please compare 1st and 2nd lanes)

Figure 2f from Pérez-Calero et al. (Perez-Calero et al., 2020). (Please compare 1st and 2nd lanes)

It is unfortunate, perhaps, that these prior studies failed to address why overexpression of RNase H1 alone is sufficient to lead to deceleration of fork speed as the reviewer pointed out. We speculate that although it is well-established that ectopic/unscheduled accumulation of R-loops impedes the progression of replication forks, physiological R-loops, presumably maintained at basal levels by factors including SOSS-INTAC, could have beneficial roles in multiple processes such as chromatin compaction and gene expression. Therefore, it is plausible that perturbations of these processes owing to RNaseH1 overexpression-induced R-loop clearance could slightly perturb DNA replication. In fact, and as discussed in several excellent reviews, elucidating and distinguishing between detrimental and beneficial roles of R-loop formation are currently under intense investigation (Crossley et al., 2019; Garcia-Muse and Aguilera, 2019; Petermann et al., 2022).

-Figure S10a sgCtr DAPI does not seem to correspond with the H2AX signal for DMSO+/DOX-

We apologize for this mistake made during figure organization, which has now been corrected.

-The scale difference between heatmaps and box plots (ie. Fig2. J vs Fig2. K, L)-is it not scaled and sum?

We apologize for the lack of clarity in describing the definition of the scales. The value in heatmaps represents the enrichment on each bin (the bin size of all heatmaps is 10 bp), while the value in boxplots represents the enrichment over the entire promoter region and, thus, is not affected by the bin size. We have now clarified this in the figure legends.

-Fig3c log2FC and Fig3g heatmaps have a lot of red which is a bit unexpected as most of the values on the sgCtr/INTS2-KO or CTR/DKO are comparable outside the peak.

We used a relatively small window to emphasize changes in promoter-proximal regions. When larger windows were used for analysis, we continued to observe a predominant change at promoters (Review Fig. 5).

Review Fig. 5. (a-b) Heatmaps showing Pol II ChIP-Rx signals over 20 kb (a) and 40 kb (b) regions centered on TSS of SOSS-INTAC target genes in sgCtr and INTS2-KO cells.

-FigS1b – not the same bands are visible in IP and input for INTAC and it does not look like Fig1e.

We apologize for the confusion caused by the use of different gel recipes in Fig. 1c (originally Fig. 1e) and Extended Data Fig. 1e (originally Fig. S1b). In Fig. 1c, we used gradient SDS-PAGE gels to resolve all INTAC subunits with a broad range of sizes. On the other hand, in Extended Data Fig. 1e, we used 7.5% gels to specifically resolve larger subunits of INTAC, which allows for a more detailed view of the larger subunits and confirms their presence in the complex. We have now re-run the gel and labeled the subunits for further clarity (Extended Data Fig. 1e).

Extended Data Fig. 1e. Immobilized GST or GST-SSB1 were incubated with purified INTAC in the presence or absence of INIP. The input and bound proteins were analyzed by Coomassie blue staining.

-Fig5c – negative controls for signal specificity (depletion of SSB1 and untagged INTs)

Extensive in vitro studies would benefit from the estimation of how many molecules are in foci in vivo.

In the revised manuscript, we have conducted IF of endogenous SSB1 and INTAC subunits in cells with simultaneous depletion of SSB1 and SSB2. We found that the loss of SSB1/2 leads not only to depletion of SSB1 signal but also abolishes puncta formation of INTAC subunits (Extended Data Fig. 16a). However, efficient degradation of INTS11, as confirmed by INTS11 WB and IF, exerted no noticeable impact on the formation of SSB1 puncta (Extended Data Fig. 16b, c). These results confirm not only the signal specificity of SOSS-INTAC puncta, but also reveal the crucial role of SSB1 in driving SOSS-INTAC puncta formation.

Extended Data Fig. 16. Extended Data Fig. 16. (a) The immunofluorescent images of SSB1 (red) and INTAC subunits (green) in wild-type and DKO cells. (b) The establishment of INTS11-dTAG cells and verification by dTAG treatment. (c) The immunofluorescent images of SSB1 (red) and INTS11 (green) in DMSO- or dTAG-treated INTS11-dTAG cells.

To estimate the amount of SSB1 protein in foci *in vivo* as suggested, we generated a stable cell line with the expression of Flag-EGFP-tagged SSB1 comparable to the endogenous levels (Review Fig. 6a). Using purified EGFP-SSB1 to calibrate the image intensities collected by quantitative microscopy, we estimated the average foci concentration of SSB1 to be 0.286 μM (Review Fig. 6b). It is worth noting that protein condensation in cells can be influenced by multiple factors, including post-translational modifications and the presence of other client proteins and nuclear acids. Therefore, the concentration of a protein required to form liquid droplets *in vitro*, in the absence of all the other modulators, may be quite different from its concentration in phase-separated condensates within cells.

Review Fig. 6. (a) Establishing a stable cell line expressing GFP-SSB1 to a level comparable to endogenous SSB1. (b) Estimation of average protein concentration of SSB1 foci in cells using purified EGFP-SSB1 to calibrate the image intensities.

-Fig6m – in text line 353 it is stated that SI mutant looks like WT but the figure shows that it is not the case.

We apologize for the inaccuracy in describing the data. As the reviewer pointed out, the SI mutant fails to fully rescue R-loop levels as seen with wild-type SSB1. We have now rephrased the description, as indicated in the excerpt below:

Results:

“We next conducted R-loop CUT&Tag-qPCR under the same conditions. As shown on example genes, induced expression of SSB1-WT, but not ΔIDR or empty vector, restores R-loop levels to the control condition (Fig. 6k). We then tested all mutant constructs described above and found that RY and SI fail to fully rescue R-loop levels comparable to SSB1-WT (Fig. 6k). Moreover, cancer-derived mutations SI72P/HI73L harboring LLPS capacity, but not R206Q that has impaired droplet formation ability, restricted R-loops to basal levels as shown at example SOSS-INTAC targets (Fig. 6k).”

Minor:

In methods there is a description of RNA-seq but not sure where the enrichment analysis is presented.

We apologize for this oversight. We do not have RNA-seq data in this study. We have carefully reviewed and updated the experimental methods for the revised manuscript.

Figure 1e – the main result in the supplement, marker size missing.

We have added the marker sizes for Fig. 1c (originally Fig. 1e) and Extended Data Fig. 1e (originally Extended Data Fig. 1b).

Fig. 1c. Coomassie staining of reconstituted human INTAC complex purified from HEK Expi293 cells, and GST-tagged human SSB1 and Strep-tagged human INIP proteins purified from *E. coli*.

Extended Data Fig. 1e. Immobilized GST or GST-SSB1 were incubated with purified INTAC in the presence or absence of INIP. The input and bound proteins were analyzed by Coomassie blue staining.

Fig6a – R206Q in IDR is indicated H206 to Y

We apologize for this error and have corrected it.

FigS14g would benefit if would be presented as FigS14f

Following the reviewer's suggestion, we now present Extended Data Fig. 21b (originally FigS14g) in the same way as Extended Data Fig. 20a (originally FigS14f).

Extended Data Fig. 20a.

Referee #3 (Remarks to the Author):

The paper by Xu et al describes the role of SOSS-INTAC complex in genome stability and transcription though its function in formation of condensates and R-loop resolution. Previous literature has already demonstrated the role of SOSS complex in DNA damage and repair and the role of INTAC complex, containing multiple Integrator subunits, in termination of non-productive transcripts. This paper is suggesting that there is formation of a bigger SOSS-INTAC complex in human cells, and it is involved in sensing ssDNA and preventing R-loop formation and genomic instability. The authors suggest that this SOSS-INTAC function is mediated though formation of liquid condensates.

This papers contains vast amount of data and experimental techniques thrown in, which is impressive on one hand, however, on the other hand it lacks novelty and mechanistic insights required to be considered for publication in Nature journal. I found that this paper very quickly brushes through multiple techniques, often without providing essential controls. There is a lot of generalizations in this paper without looking into details of mechanisms in questions. I am still struggling to understand the novelty of this work, since the authors have not actually demonstrated how condensation function of SSB1 of the SOSS-INTAC complex, required for regulation of R-loops, is actually important for transcription and why.

Major comments:

1. The authors propose that the condensation function of SSB1 important for regulation of promoter-proximal pausing/transcription, however, they have not actually tested this experimentally. The authors need to use their condensation mutants and assess promoter proximal pausing. I do not think that simple CHIP-seq experiments are sufficient here, since the pausing index needs to be calculated.

We thank the reviewer for the helpful comments. As suggested, we calculated the pausing index by comparing the ratio of Pol II at promoters versus gene bodies. As shown below, the pausing index is evidently increased upon the loss of SSB1 or INTS2, suggesting that SOSS-INTAC disruption induces the accumulation of Pol II pausing (Extended Data Fig. 6j, k).

Extended Data Fig. 6j-k. Empirical cumulative distribution function plot of the pausing index (PI) distribution in CTR and DKO (j), or in sgCtr and INTS2-KO cells (k).

To examine the regulation of promoter-proximal pausing by SSB1 mutants as suggested, we quantified the Pol II occupancy at promoters of example genes by ChIP. Our results suggest that SSB1 mutants that are competent for condensation (HY and S172P/H173L) can restore Pol II levels at promoter-proximal regions,

while SSB1 mutants that are defective in condensation (Δ IDR, SI, RY, R206Q) fail to rescue Pol II levels at these regions (Extended Data Fig. 21a).

Extended Data Fig. 21a. Pol II ChIP-qPCR in DMSO- or dTAG-treated SSB1-dTAG cells with overexpression of wild-type or mutant SSB1.

2. This paper uses a lot of generalizations. Is the mechanism of action for SOSS-INTAC complex the same for all gene categories (i.e. coding vs non-coding, intronless, intron-containing etc respond in a similar way)? What happens with the genes which do not have a strong promoter-proximal pause?

We thank the reviewer for bringing up this point. As the reviewer suggested, we compared intronless (mono-exonic) vs. intron-containing (multi-exonic) genes, and coding vs non-coding genes. Our analysis suggests that all groups of genes, regardless of intron numbers and/or protein-coding capacities, exhibit promoter-proximal Pol II accumulation (Review Fig. 4a, c). However, mono-exonic and non-coding genes exhibit a greater upregulation of Pol II levels within gene bodies (Review Fig. 4a, c), which is consistent with the observation reported by Dr. Torben Jensen’s group (Lykke-Andersen et al., 2021).

A genome-wide effect of Integrator was recently confirmed by a study from Dr. Karen Adelman’s group, where they revealed that “Integrator endonuclease drives promoter-proximal termination at **all** RNA polymerase II-transcribed loci” (Stein et al., 2022). This is also in accordance with the findings from Torben Jensen’s group and by our studies that most genes, whether mono- or multi-exonic, show an elevation of promoter-proximal Pol II levels following Integrator or SSB1 disruption. Moreover, Adelman and colleagues suggested that shorter genes tend to have greater upregulation of Pol II within gene bodies due to longer genes exhibiting more severe defects in progressive elongation after INTS11 depletion. To determine whether SSB1 depletion results in similar changes as Integrator disruption, we separated genes into quartiles by length and analyzed the Pol II profiles in CTR and DKO cells (Review Fig. 4b). Our results indicate that the shorter genes have greater elevation of Pol II levels within gene bodies in DKO cells, which is in agreement with the observation from Dr. Adelman’s group that INTS11 loss tends to preferentially upregulate shorter genes.

Review Fig. 4a-c. (a-c) Metagenome analysis of Pol II occupancy in CTR and DKO cells for mono-exonic vs. multi-exonic genes (a), quartiles of genes classified by gene length (b), and protein-coding vs. non-coding genes (c).

We further divided genes into quartiles by degree of pausing (calculated by pausing index in the control condition) and analyzed the change of pausing resulting from SSB1 depletion. All four groups of genes show an overall higher degree of pausing in DKO than in CTR cells (Review Fig. 4d), consistent with the above-mentioned notion that SOSS-INTAC drives promoter-proximal termination in a genome-wide manner.

Review Fig. 4d. Boxplots quantifying the pausing index in CTR and DKO cells for all four quartiles of genes classified by pausing index in CTR cells.

Collectively, we conclude that disruption of either SSB1 or Integrator/INTAC is sufficient to induce a genome-wide accumulation of Pol II at promoters, while the changes in Pol II profiles within gene bodies can vary according to various gene properties. Specifically, short and mono-exonic genes are more prone to be upregulated compared with long and multi-exonic genes. Importantly, since SSB1 depletion induces a widespread R-loop accumulation at promoter-proximal regions, our results point toward a unified model of how SOSS-INTAC attenuates the accumulation of Pol II and R-loops at promoters.

3. The authors claim that R-loops are important for recruitment of SSB1, however this is not supported by the provided data. The authors used DOX-inducible over-expression of RNase H1 to test this. However, this experiment did not result in a significant decrease of R-loops as demonstrated in Fig S8a and c by IF.

However, the authors demonstrate a reduction in SSB1 signal by ChIP on 4 specific genes. I would like to see the DRIP-qPCR validation that the R-loops are actually reduced on the genes where the SSB1 signal goes down supporting their claims.

We thank the reviewer for raising this interesting point. As the reviewer pointed out, RNase H1 overexpression induces a moderate but consistent decline in cellular R-loops in control cells as measured by either GFP-dRNH1-based (Fig. 4c, d, Extended Data Fig. 12a, b) or S9.6-based (Extended Data Fig. 10a-d) IF assays. We speculate that the reason for the observed moderate changes in R-loop formation is a consequence of preexisting physiological R-loops being maintained at basal levels. Following the reviewer's suggestion, we have conducted R-loop CUT&Tag to quantify R-loop changes on example genes (Extended Data Fig. 8e), which confirm a decrease in R-loops.

Extended Data Fig. 8e. R-loop CUT&Tag-qPCR in cells with DOX-inducible RNase H1 expression. DMSO-treated cells were incubated with IgG but not S9.6 (3rd lane) or treated with RNase H1 during CUT&Tag (4th lane) to confirm the specificity of detected R-loop signals. Values are mean \pm SD (n = 3).

We have also used ChIP-qPCR to validate the reduction in the occupancy of SSB1 and INTS3 on promoters of these genes (Extended Data Fig. 8c, 9b).

Extended Data Fig. 8c (left). SSB1 ChIP-qPCR on promoters of example genes in cells with DOX-inducible RNase H1 expression. Values are mean \pm SD (n = 3).

Extended Data Fig. 9b (right). INTS3 ChIP-qPCR on promoters of example genes in cells with DOX-inducible RNase H1 expression. Values are mean \pm SD (n = 3).

4. Figure 3a shows that *SSB1* depletion results in increase of Pol II over the whole *JUNB* (*JUN* was shown in the initial manuscript) gene. This profile looks quite different compared to *RSBN1* gene (where there seems no effect in the body of the gene). Also the authors see an increase in sense and anti-sense transcription. These are rather strange phenotypes and they do not really fit with the idea of affected pausing (since in the pausing defect, the elongation should be affected and less Pol II would be expected in the body of the gene). What is happening with Pol II in the body of the genes – is this also going up? Can the authors provide an explanation to what is going on? I think the authors tried to generalize too much regarding their conclusions in this paper (see also my comment 2 above).

As we demonstrated above (comment 2), results from both our studies and those of Dr. Karen Adelman's group (Stein et al., 2022) have shown that, although *SSB1* and *INTAC/Integrator* loss induces genome-wide Pol II accumulation at promoters, the change of Pol II within gene bodies varies for genes of different lengths. Specifically, shorter genes tend to have greater upregulation of Pol II within gene bodies (Review Fig. 4b). Therefore, the observation that the Pol II occupancy at gene bodies increased much more pronouncedly in *JUN* (~3 kb, Q1 group) than *RSBN1* (~51 kb, Q3 group) is in accordance with our global analysis.

Review Fig. 4b. Metagenome analysis of Pol II occupancy in CTR and DKO cells for quartiles of genes classified by gene length.

Notably, the upregulation of *JUN* upon loss of *INTAC* was also confirmed by the Adelman group, where they used *JUN* as an example gene to demonstrate the upregulation of expression of shorter genes.

Figure 6b-c from Chad B. Stein et al. (Stein et al., 2022).

We are grateful that the reviewer noticed that *SSB1* loss induces an increase in both sense and anti-sense transcription. This observation is in agreement with several previous studies. For example, the pervasive upregulation of Pol II in the sense direction upon *INTAC/Integrator* loss was shown by the groups of Drs. Karen Adelman (Stein et al., 2022), Ramin Shiekhattar (Beckedorff et al., 2020), and by us (Wang et al., 2022; Zheng et al., 2020); the broad role of *INTAC/Integrator* in attenuating non-coding transcripts, including anti-sense transcripts (or *PROMPTS*) and eRNAs, was demonstrated by the groups of Drs. Torben

Heick Jensen (Lykke-Andersen et al., 2021), Mo Chen (Liu et al., 2022), Ramin Shiekhatter (Lai et al., 2015). These studies collectively highlight the pervasive role of INTAC/Integrator in mediating promoter-proximal termination of different classes of transcription units (TUs), despite that the defects in promoter-proximal termination potentially result in disparate transcriptional outcomes for TUs of different or even the same classes.

5. I have not found any information related to the number of repeats for genomic experiments, therefore the reproducibility of these data are not clear. This information should be included at least in the methods section. All figures with peak distribution on specific genes should include input tracks, since the read values are extremely low for the pull-down experiments in multiple figures (e.g. Fig 3i, Fig3 a; 2i; Fig4 a, Fig 6j). Furthermore, these results also need to be validated by qPCR. IN some occasions the authors have this, however, in case of R-loop experiments in Fig. 4 a, this information is missing. The authors need to convince the readers that the IP signals are above the input background and are reproducible in independent biological experiments.

We thank the reviewer for the constructive comments. We apologize for a mistake introduced during ATAC-seq data normalization that led to low values, which has now been corrected (Fig. 3g-i, Extended Data Fig. 7d-e). All of the genomic experiments have at least two biological replicates in this study. We now include detailed information of genomic experiments in Supplementary Table 1. As suggested, we have included input tracks for all gene examples shown in figures (Fig. 2i, 3a, 3i, 6h, Extended Data Fig. 3b, 5a, 6a, 7e, 8b, 9a, 19f). Moreover, we have conducted qPCR experiments to confirm the enrichment of signals by ChIP-seq or CUT&Tag (Fig. 2h, 6j, Extended Data Fig. 2d, 4e, 5f, 7f, 8c, 8e, 9b, 12f).

6. Figure 4f should include a demonstration of multiple nuclei and not just one cell. Did the authors do any selection for the cells over-expressing wt and mutant constructs of INT11, since the transfection efficiency is not 100%? Are they quantifying all cells or just cells over-expressed with the constructs? Furthermore, there is no statistics presented with the WT over-expression, however in the text the authors claim an effect of complementation? All the statistics need to be clearly presented.

As suggested by the reviewer, we have updated Fig. 4f by presenting images with more than one nucleus. We also clarify that we used lentiviral vectors expressing the blasticidin-resistance gene as well as the wild-type or catalytic-dead INTS11 followed by approximately a week of blasticidin selection. We also included cells not expressing the blasticidin-resistance gene to evaluate antibiotic selection efficiency. Therefore, given that all surviving cells express ORFs in the vector, we quantified all selected cells for R-loop quantification. Moreover, we have conducted statistical analysis regarding wild-type INTS11 overexpression as the reviewer suggested (Fig. 4f, g).

Fig. 4f. IF of R-loop signals in cells with shRNA targeting INTS11 (NonT shRNA as the control) and overexpression of wild-type (WT) or catalytic-dead (E203Q) INTS11 (empty vector as the control).

7. The paper would benefit from a clear model.

We thank the reviewer for this helpful suggestion and have added a working model as shown below (Extended Data Fig. 21c). This model could potentially be incorporated into the main Fig. 6, if this reviewer finds this model to be sufficiently informative and not overtly guilty of over-interpretation of data.

c

Extended Data Fig. 21c. A working model demonstrating the proposed mechanism by which SOSS-INTAC attenuates R-loop accumulation and maintains genome stability. The left side represents wild-type cells, where the SSB1 subunit of SOSS interacts with the single stranded DNA (ssDNA) to recruit the SOSS-INTAC complex to promoters and drives condensate formation. RNA cleavage by SOSS-INTAC condensates permits degradation by a combination of XRN2 and the exosome complex, leading to premature promoter-proximal termination by RNA Pol II and R-loop attenuation. In the context of cancer-associated mutations of SSB1 that impair condensation, or disruption of SOSS-INTAC functions, the loss of premature promoter-proximal Pol II termination leads to aberrant accumulation of R-loops, with potential adverse consequences such as DNA damage.

Minor comments:

1. This paper presents various pieces of genome-wide analysis, often presented as genome-wide heat maps (Fig 1i, 2j etc). This is OK, however often the information is actually lost in such condensed presentation of the genome-wide data. Therefore, it is important to provide an experimental validation by qPCR.

We agree with the reviewer and have provided qPCR data to validate our sequencing data of ChIP or CUT&Tag (Fig. 2h, 6j, Extended Data Fig. 2d, 4e, 5f, 7f, 8c, 8e, 9b, 12f).

2. *In vitro* binding experiments presented in Fig 2 d-e should include concentration range indicated on the figure, so that the readers can easily compare the differences between different protein/complexes analysed.

We appreciate these constructive comments and have now specified the protein concentrations used for all EMSA as requested.

3. Figures 1a and 1k are repetitive. Only one should be left, for example 1k (with all corresponding labels) instead of 1a, b. Currently there is no description of Figure 1k in the figure legends. Furthermore, it is not immediately clear what A and C in green mean on the Fig 1A, possibly more complete names are required on the figure. Some of the subunits are not on the figure (i.e. INT 12), why is that- this needs to be stated in the legends.

We thank the reviewer for the helpful comments. We have moved the original Fig. 1a and 1b to Extended Data Fig. 1a and 1b. Moreover, we have updated the figure legends of Fig. 1j (original Fig. 1k) with a more detailed description of the points raised by the reviewer, as shown below:

Figure legends of Fig. 1j:

“(j) Schematic of the SOSS-INTAC complex. Based on structural and biochemical information (Jia et al., 2021; Pfleiderer and Galej, 2021; Ren et al., 2014; Sabath et al., 2020; Zheng et al., 2020), the complex can be divided into 6 modules, including backbone (INTS1, INTS2 and INTS7), shoulder (INTS5 and INTS8), endonuclease (INTS4, INTS9 and INTS11), phosphatase (INTS6, PP2A-A and PP2A-C), INTS10-13-14, and SOSS (INTS3, SSB1/2, INIP) modules. The structural organization of the backbone, shoulder, endonuclease and phosphatase modules are illustrated based on the structure of INTAC (Zheng et al., 2020). The organization of the SOSS module was placed according to the structures of SOSS (Ren et al., 2014) and INTS3-INTS6 (Jia et al., 2021). The organization of the INTS10-13-14 was estimated based on structural and biochemical information of INTS10-13-14 (Sabath et al., 2020). The structural placement of INTS12 is currently unclear.”

Referee #4 (Remarks to the Author):

The authors report the genome stability regulator SOSS associates with the transcription regulator INTAC to terminate transcription and reduce R-loop formation from paused Pol II. The authors propose that, mechanistically, ssDNA generated from R-loop at active transcription sites recruits INTAC through the ssDNA binding protein SSB1 in the SOSS complex to reduce R-loop via RNA cleavage in a manner that depends on SSB1 phase separation. This work provides an important link between transcription and genome stability regulation and showcases the possible functional importance of protein phase separation. My major concern is on the mechanism of SOSS-INTAC regulation of R-loop formation. The author used immunofluorescence (IF) and a R-loop CUT&Tag technique to assess R-loop reduction for SSB1 knockdown or fast degradation. I find the role of SSB1 in R-loop regulation is well supported. However, to establish role of INTAC in R-loop regulation, the authors only used IF with a S9.6 antibody and GFP-tagged catalytic-dead RNase H1 for R-loop detection. I find this is less convincing because the elevated R-loops under the regulation of SOSS-INTAC should correspond to active transcription sites and elevated level of DNA damage stained by gamma H2Ax. However, IF stains of elevated R-loops do not show patterns similar to SOSS-INTAC foci or gamma H2Ax foci. Instead, the staining for S9.6 looks more like nucleoli and that of dRNaseH1 looks like regions excluded of DAPI staining, maybe due to strong non-specific binding in those regions. Therefore, I would be more convinced with data from the more sensitive R-loop CUT&Tag for INTAC knockdown and rescue.

We greatly appreciate the positive comments and recognition that our study provides an important link between the regulation of transcription and genome stability. To address the reviewer's concern regarding the role of INTAC in R-loop regulation, we depleted INTS11, the catalytic subunit of the endonuclease module, and rescued its depletion with ectopic INTS11 expression (Extended Data Fig. 12c, d). As assessed by R-loop IF and CUT&Tag, INTS11 depletion alone is sufficient to induce a strong accumulation of R-loops (Fig. 4f, g, Extended Data Fig. 12e). Importantly, this accumulation is rescued by wild-type but not catalytic-dead INTS11 (E203Q) expression, with simultaneous INTS11 knockdown and catalytic-dead INTS11 expression giving rise to the most pronounced R-loop enrichment (Fig. 4f, g, Extended Data Fig. 12e). These results suggest an important role for INTAC in preventing abnormal R-loop accumulation.

Fig. 4f-g. (f) IF of R-loop signals in cells with shRNA targeting INTS11 (NonT shRNA as the control) and overexpression of wild-type (WT) or catalytic-dead (E203Q) INTS11 (empty vector as the control). (g) Quantification of the nuclear R-loop signals for (f). ****: $p < 0.0001$. Statistical analyses were performed using two-tailed unpaired t-test ($n = 180$).

Extended Data Fig. 12e. Heatmaps of R-loop CUT&Tag signals over 6 kb regions centered on TSS of SOSS-INTAC target genes in cells with INTS11 knockdown and overexpression of wild-type or E203Q INTS11.

We also conducted R-loop IF and γ H2AX CUT&Tag in sgCtr and INTS2-KO cells. As shown below, INTS2 loss induces an apparent induction of cellular R-loop levels (Extended Data Fig. 12a, b) and promoter-proximal γ H2AX signals (Extended Data Fig. 14b-d), supporting the role of INTAC in regulating R-loops and genome stability.

Extended Data Fig. 12a-b. GFP-dRNH1-based IF of R-loops in sgCtr and INTS2-KO cells with DOX-inducible RNase H1 expression (a) and the quantification of the nuclear R-loop signals (b). *: $p < 0.05$, ****: $p < 0.0001$. Statistical analyses were performed using a two-tailed unpaired t-test ($n = 110$).

Extended Data Fig. 14d. Heatmaps showing γ H2AX occupancy in sgCtr and INTS2-KO cells. The peaks were centered on TSS of SOSS-INTAC target genes.

To establish a direct link between SSB1 and INTAC in regard to the regulation of R-loops, we generated a mutant SSB1, E97A/F98A, that is defective in interacting with INTAC as confirmed by IP and western blotting (Extended Data Fig. 5e). We next induced the expression of wild-type SSB1 or E97A/F98A in DKO cells followed by R-loop CUT&Tag. This reveals that wild-type SSB1, but not E97A/F98A, restored the R-loop levels to the control condition (Extended Data Fig. 12f), indicating that the physical interaction between SSB1 and INTAC is crucial for INTAC-mediated R-loop regulation.

Extended Data Fig. 12f. R-loop CUT&Tag-qPCR on example genes in CTR or DKO cells overexpressed with empty vector, wild-type SSB1 or E97A/F98A, the mutant defective in interacting with INTS3. Values are mean \pm SD (n = 3).

(Continued from above) Another reason is a previous study (<https://doi.org/10.1111/mmi.14529>) shows that bacterial SSB can direct RNase HI foci to replication sites to reduce R-loop, which might provide an alternative explanation for SSB1 regulation of R-loop in this case. I think the authors should check this possibility to either rule out this pathway. Maybe SSB1 recruits INTAC and RNase H1 to work synergistically to reduce R-loop formation. Indeed, a previous study ([doi:10.1073/pnas.2000761117](https://doi.org/10.1073/pnas.2000761117)) on bacterial SSB phase separation shows SSB condensates can enrich many weak interacting partners.

We thank the reviewer for raising this crucial point. To examine whether SSB1/2 interacts with RNase H1 as does the bacterial SSB, we first conducted Co-IP of SSB1/2, which failed to detect an interaction between SSB1/2 and RNase H1 (Review Fig. 1a, b).

Based on the crystal structure of bacterial SSB and RNase HI, a 9 amino acid peptide at the C-terminus of SSB mediates the binding of RNase HI (Petzold et al., 2015). However, the corresponding sequences for human SSB1 and SSB2 are substantially different from that of bacterial SSB (Review Fig. 1c). Moreover, the C terminus of SSB has an overall negative charge while the C termini of SSB1 and SSB2 are more positively charged (Review Fig. 1c). In contrast to the difference between SSB and SSB1/2, structure and sequence comparisons reveal that many amino acid residues crucial for bacterial RNase HI to recognize SSB are conserved (Review Fig. 1d, e). Therefore, we posited that the lack of SSB1/2-RNase H1 interaction in human primarily results from the evolutionary changes in SSB1/2 at its C terminus.

To further explore this hypothesis, we generated a chimeric construct by replacing the C-terminal 9 residues of human SSB1 with the corresponding region of bacterial SSB. V5-tagged human SSB1, bacterial SSB, or the chimeric construct was overexpressed with Flag-tagged RNase H1 in cells followed by V5 tag IP. We find that bacterial SSB, but not human SSB1, interacts with human RNase H1 (Review Fig. 1f, 7th and 8th lanes). Notably, the chimeric construct gains the capacity of binding RNase H1 (Review Fig. 1f, 9th lane), confirming that the C terminus of bacterial SSB is crucial for its interaction with RNase H1.

Review Fig. 1. (a-b) V5 Co-IP in cells stably expressing V5-tagged SSB1 (a) and SSB2 (b), followed by western blotting of INTAC subunits and RNase H1. (c) Sequence comparison of the C-termini of bacterial SSB and human SSB1/2. (d) Illustration of the bacterial SSB–RNase HI structure. (e) Sequence comparison of bacterial RNase HI and human RNase H1. The evolutionarily conserved amino acid residues of bacterial RNase HI potentially important for its interaction with SSB are highlighted in red boxes. (f) V5 Co-IP and western blotting in cells stably expressing V5-tagged human SSB1, bacterial SSB, or the chimeric construct (replacing the C-terminal 9 residues of human SSB1 with that of bacterial SSB) in addition to Flag-tagged human RNase H1.

Other minor comments are:

1. The importance of IDR in SSB1 phase separation is well-supported with existing data. However, IDR is barely the driver for phase separation at physiological conditions (e.g., the authors used concentrations on the order of 50uM and low salt of 30-50mM). A previous study (doi:10.1073/pnas.2000761117) on bacterial SSB has established several contributions of multivalence to SSB phase separation including oligomerization, DNA /RNA binding. The domain analysis in this work does not show significant contribution from non-IDR regions. Is human SSB1 not a tetramer? If it's not, authors should discuss the

differences in phase behavior and not make it look like IDR is the major driver of SSB1 phase separation without considering other sources of valance.

We completely agree with the reviewer that, although the IDR is required for SSB1 condensation, a plethora of other factors (e.g., interactions with other proteins, tetramerization, and the binding to nucleic acids) contributes to condensation capacity. Given that human SSB1 can indeed form tetramers as the reviewer pointed out, we examined whether tetramerization contributes to droplet formation in vitro. Specifically, we mutated two crucial amino acid residues (N16D and N18D) that mediate tetramerization based on structural information (Touma et al., 2017) (Review Fig. 7a). As shown by droplet formation assay, the condensation capacity of SSB1 is compromised by these two mutations, implying that tetramerization could contribute to SSB1 condensation (Review Fig. 7b, c).

Review Fig. 7. (a) Structural illustration of SSB1-SSB1 interaction interface. (b-c) Wild-type SSB1 and mutant SSB1 protein defective in tetramerization were analyzed using droplet formation assays (b). Quantification of the size of droplets are shown in (c). Red lines indicate the mean in each population.

We have now discussed the potential contributions of non-IDR regions and other elements in SSB1 condensation, as indicated in the excerpt below:

Discussion:

“However, it is important to note that the IDR of SSB1 could possess condensation-independent functions, such that mutations disrupting condensation may also introduce additional impacts yet to be identified. Meanwhile, non-IDR regions of SSB1 may also contribute to condensation through different mechanisms such as self-oligomerization and/or interactions with nucleic acids or proteins. Therefore, future studies are warranted to systematically investigate the biophysical properties of SOSS-INTAC and their contributions to transcription, R-loop regulation, and genome stability, and the degree to which the condensation ability of SOSS-INTAC contributes to these processes.”

2. Since phase separation is based on in vitro data, the functional relevance of phase separation is implied and not directly proved. Therefore, I would advise the authors to make this distinction clear in the manuscript unless they want to provide direct evidence for SSB1 phase separation in cells at physiological conditions, which I think it’s beyond the scope of this work. For the same reason, equal care should be taken when interpreting the mutation results in the IDR region. IDRs can have non-phase separating functions. For example, Wolak 2020 (<https://doi.org/10.1111/mmi.14529>) suggested “Stimulation requires docking of the intrinsically disordered C-terminus of SSB (SSB-Ct) into a binding pocket on RNase H1”.

We deeply appreciate the constructive suggestions. In response to the implications of SSB1 phase separation in cells asked by this and other reviewers, we conducted additional experiments for the revised manuscript as listed below:

(1) To determine the interdependency of SSB1 and INTAC in puncta formation in cells, we first depleted SSB1 and SSB2 and conducted IF of endogenous INTAC subunits. Notably, loss of SSB1/2 abolishes puncta formation of INTAC subunits (Extended Data Fig. 16a). However, induced degradation of INTS11 exerted no noticeable impact on the formation of SSB1 puncta (Extended Data Fig. 16b, c). Supported by *in vitro* analysis of SOSS-INTAC (Fig. 5k, l, Extended Data Fig. 17h, i), we conclude that SSB1 drives the puncta formation of INTAC in cells, but not vice versa.

Extended Data Fig. 16. Extended Data Fig. 16. (a) The immunofluorescent images of SSB1 (red) and INTAC subunits (green) in wild-type and DKO cells. (b) The establishment of INTS11-dTAG cells and verification by dTAG treatment. (c) The immunofluorescent images of SSB1 (red) and INTS11 (green) in DMSO- or dTAG-treated INTS11-dTAG cells.

(2) To analyze the puncta formation of SSB1 mutants in vivo, we induced the expression of wild-type or mutant SSB1 in dTAG-treated SSB1-dTAG cells. Consistent with the condensation results in vitro (Fig. 6b, c), the puncta formation capacity of Δ IDR and RY is abolished and that of SI and R206Q is severely impaired compared with wild-type SSB1 (Extended Data Fig. 20b).

Extended Data Fig. 20b. Extended Data Fig. 20b. Representative images of SSB1 immunofluorescent signals in dTAG-treated SSB1-dTAG cells with overexpression of wild-type or mutant SSB1.

(3) To examine whether SSB1 can form liquid-like condensates in cells, we used the “optoDroplet” system by fusing SSB1 with mCherry-labeled Arabidopsis photoreceptor cryptochrome 2 (CRY2) (Kilic et al., 2019; Shin et al., 2017; Taslimi et al., 2014). This reveals that the droplet formation of SSB1, but not the control, is substantially boosted upon light induction (Extended Data Fig. 17c). Moreover, SSB1 puncta undergo frequent fusion and fission events (Extended Data Fig. 17d, e). We also find that the fluorescent signals of foci recover readily after photobleaching (Extended Data Fig. 17f), which is indicative of droplet-like behavior.

Extended Data Fig. 17c-f. (c) The establishment of the “optoDroplet” system by fusing SSB1 with mCherry-labeled Arabidopsis photoreceptor cryptochrome 2 (CRY2). Representative images of SSB1 and control are shown before and after light induction. (d-e) Time-lapse imaging demonstrating spontaneous fusions (d) and fissions (e), as indicated by the arrows, of SSB1 condensates in cells. (f) Representative micrographs of SSB1 puncta before and after photobleaching.

Collectively, we believe that the newly added data corroborate the involvement of SSB1's condensation in R-loop regulation. That said, we have paid extra caution in the functional relevance of phase separation for SSB1-mediated regulation of transcription and R-loops throughout the manuscript.

In addition, we fully agree with the reviewer that IDRs can have non-phase separating functions. Although the interaction between the SSB C-terminus and RNase HI is unlikely to be conserved in human as demonstrated above (Review Fig. 1), it is plausible that the disordered C-terminus of SSB1 harbors (or has gained during evolution) other non-phase separating functions yet to be discovered. As such, mutations that impair SSB1's condensation capacity, even by changing one single amino acid residue, could potentially perturb other underlying non-phase separating functions. We have discussed this important point in the manuscript, as indicated in the excerpt below:

Discussion:

“However, it is important to note that the IDR of SSB1 could possess condensation-independent functions, such that mutations disrupting condensation may also introduce additional impacts yet to be identified. Meanwhile, non-IDR regions of SSB1 may also contribute to condensation through different mechanisms such as self-oligomerization and/or interactions with nucleic acids or proteins. Therefore, future studies are warranted to systematically investigate the biophysical properties of SOSS-INTAC and their contributions to transcription, R-loop regulation, and genome stability, and the degree to which the condensation ability of SOSS-INTAC contributes to these processes.”

3. Line 262 “SOSS-INTAC forms condensates in cells” is misleading. Results show foci, which may not necessarily reflect concentrating effect required for the definition of condensates due to SOSS-INTAC binding to DNA. Please see distinction demonstrated for 53BP1 and gamma H2Ax in this paper (<https://doi.org/10.15252/embj.2018101379>).

We thank the reviewer for raising this critical point. In the paper quoted by the reviewer (Kilic et al., 2019), the authors used the optoDroplet assay to differentiate the biophysical properties of 53BP1 and γ H2AX foci. Their results showed that 53BP1, but not MDC1— an adaptor protein that binds γ H2AX, forms liquid-like droplets upon light induction. To determine whether the SSB1 foci are more similar to that of 53BP1 or MDC1, we also used the optoDroplet system to examine its droplet formation capacity in cells. As described in the response to point 2 above, we found that the droplet formation of SSB1, but not the control, is substantially boosted upon light induction (Extended Data Fig. 17c). Moreover, SSB1 puncta undergo frequent fusion and fission events (Extended Data Fig. 17d, e). We also found that the fluorescent signals of foci recover readily after photobleaching (Extended Data Fig. 17f), which is indicative of liquid-like behavior.

Despite these additional lines of evidence of condensate formation, we have changed the heading for this section to “SOSS-INTAC forms nuclear puncta in cells” to more accurately summarize our findings as suggested.

4. Line 295 SSB1 drives the formation of SOSS-INTAC condensates. Data suggest more of the model SSB1 undergoes phase separation and then recruits INTAC. A SOSS-INTAC condensate implies both contribute to valance in phase separation. The authors could map the effect of INTAC on SSB1 phase diagram in vitro to gain a better picture instead of using the one data point the manuscript currently has.

We appreciate the constructive comments. As suggested, to get a better picture of how SSB1 and INTAC affects each other regarding condensation, we incubated different concentrations of SSB1 with INTAC and quantified the droplet signals. Although formation of INTAC condensates requires the presence of SSB1,

the condensation capacity of SSB1 is at most marginally affected by the presence of INTAC (Extended Data Fig. 17h, i). These data suggest that SSB1 drives the condensation of SOSS-INTAC.

Extended Data Fig. 17h-i. Different concentrations of GFP-SSB1 were mixed with Alx568-labeled INTAC and analyzed using droplets formation assay (h), followed by the quantification of the relative GFP intensity per droplet (i). Red lines indicate the mean in each group.

(Continued from above) In addition, the authors could look at how SSB1 and INTAC affect each other's foci formation and their co-localization in cells to make it clearer/more relevant whether they co-phase separate or one phase separates and then recruits the other.

As suggested by the reviewer and as described in the response to point 2 above, we determined how SSB1 and INTAC affect each other's foci formation in cells by depleting SSB1 or INTAC subunit INTS11. This reveals that SSB1 is essential for the puncta formation of INTAC (Extended Data Fig. 16a), but not vice versa (Extended Data Fig. 16b, c), supporting the key role of SSB1 in SOSS-INTAC condensation.

(Continued from above) Foci formation in cells in response to IDR deletion and mutations can also be a good assay to further test the importance of IDR revealed in the in vitro data.

As described in the response to point 2 above, we also examined foci formation in cells in response to IDR deletion and site-specific mutations as the reviewer suggested. As shown above, we induced the expression of wild-type or mutant SSB1 in dTAG-treated SSB1-dTAG cells. Consistent with the in vitro condensation results (Fig. 6b, c), the puncta formation capacity of ΔIDR and RY is abolished and that of SI and R206Q are severely impaired compared with wild-type SSB1 (Extended Data Fig. 20b).

5. Figure legends can use more detailed information. For example, Figure 6h: with DMSO or dTAG?

In this case, it is with DMSO in Fig. 6f (originally Fig. 6h). We apologize for the lack for clarity and have updated all of the figure legends with more detailed descriptions.

6. Typo Line 240: which is eliminated by DOX-induced expression of GFP-dRNH1 (Fig. 4d). Should be expression of WT RNaseH1?

Yes, it should be the “expression of wild-type RNase H1”. We apologize for this typo and have now corrected it.

Reference:

- Bayona-Feliu, A., Barroso, S., Munoz, S., and Aguilera, A. (2021). The SWI/SNF chromatin remodeling complex helps resolve R-loop-mediated transcription-replication conflicts. *Nat Genet* 53, 1050-1063.
- Beckedorff, F., Blumenthal, E., daSilva, L.F., Aoi, Y., Cingaram, P.R., Yue, J., Zhang, A., Dokaneheifard, S., Valencia, M.G., Gaidosh, G., *et al.* (2020). The Human Integrator Complex Facilitates Transcriptional Elongation by Endonucleolytic Cleavage of Nascent Transcripts. *Cell Rep* 32, 107917.
- Chedin, F., Hartono, S.R., Sanz, L.A., and Vanoosthuyse, V. (2021). Best practices for the visualization, mapping, and manipulation of R-loops. *EMBO J* 40, e106394.
- Chen, F.X., Woodfin, A.R., Gardini, A., Rickels, R.A., Marshall, S.A., Smith, E.R., Shiekhattar, R., and Shilatifard, A. (2015). PAF1, a Molecular Regulator of Promoter-Proximal Pausing by RNA Polymerase II. *Cell* 162, 1003-1015.
- Crossley, M.P., Bocek, M., and Cimprich, K.A. (2019). R-Loops as Cellular Regulators and Genomic Threats. *Mol Cell* 73, 398-411.
- Crossley, M.P., Brickner, J.R., Song, C., Zar, S.M.T., Maw, S.S., Chedin, F., Tsai, M.S., and Cimprich, K.A. (2021). Catalytically inactive, purified RNase H1: A specific and sensitive probe for RNA-DNA hybrid imaging. *J Cell Biol* 220.
- Garcia-Muse, T., and Aguilera, A. (2019). R Loops: From Physiological to Pathological Roles. *Cell* 179, 604-618.
- Jia, Y., Cheng, Z., Bharath, S.R., Sun, Q., Su, N., Huang, J., and Song, H. (2021). Crystal structure of the INTS3/INTS6 complex reveals the functional importance of INTS3 dimerization in DSB repair. *Cell Discov* 7, 66.
- Kilic, S., Lezaja, A., Gatti, M., Bianco, E., Michelena, J., Imhof, R., and Altmeyer, M. (2019). Phase separation of 53BP1 determines liquid-like behavior of DNA repair compartments. *EMBO J* 38, e101379.
- Lai, F., Gardini, A., Zhang, A., and Shiekhattar, R. (2015). Integrator mediates the biogenesis of enhancer RNAs. *Nature* 525, 399-403.
- Liu, X., Guo, Z., Han, J., Peng, B., Zhang, B., Li, H., Hu, X., David, C.J., and Chen, M. (2022). The PAF1 complex promotes 3' processing of pervasive transcripts. *Cell Rep* 38, 110519.
- Liu, X., Kraus, W.L., and Bai, X. (2015). Ready, pause, go: regulation of RNA polymerase II pausing and release by cellular signaling pathways. *Trends Biochem Sci* 40, 516-525.
- Lykke-Andersen, S., Zumer, K., Molska, E.S., Rouviere, J.O., Wu, G., Demel, C., Schwalb, B., Schmid, M., Cramer, P., and Jensen, T.H. (2021). Integrator is a genome-wide attenuator of non-productive transcription. *Mol Cell* 81, 514-529 e516.
- Perez-Calero, C., Bayona-Feliu, A., Xue, X., Barroso, S.I., Munoz, S., Gonzalez-Basallote, V.M., Sung, P., and Aguilera, A. (2020). UAP56/DDX39B is a major cotranscriptional RNA-DNA helicase that unwinds harmful R loops genome-wide. *Genes Dev* 34, 898-912.
- Petermann, E., Lan, L., and Zou, L. (2022). Sources, resolution and physiological relevance of R-loops and RNA-DNA hybrids. *Nat Rev Mol Cell Biol* 23, 521-540.
- Petzold, C., Marceau, A.H., Miller, K.H., Marqusee, S., and Keck, J.L. (2015). Interaction with Single-stranded DNA-binding Protein Stimulates Escherichia coli Ribonuclease HI Enzymatic Activity. *J Biol Chem* 290, 14626-14636.

Pfleiderer, M.M., and Galej, W.P. (2021). Structure of the catalytic core of the Integrator complex. *Mol Cell* *81*, 1246-1259 e1248.

Prendergast, L., McClurg, U.L., Hristova, R., Berlinguer-Palmini, R., Greener, S., Veitch, K., Hernandez, I., Pasero, P., Rico, D., Higgins, J.M.G., *et al.* (2020). Resolution of R-loops by INO80 promotes DNA replication and maintains cancer cell proliferation and viability. *Nat Commun* *11*, 4534.

Rahl, P.B., Lin, C.Y., Seila, A.C., Flynn, R.A., McCuine, S., Burge, C.B., Sharp, P.A., and Young, R.A. (2010). c-Myc regulates transcriptional pause release. *Cell* *141*, 432-445.

Ren, W., Chen, H., Sun, Q., Tang, X., Lim, S.C., Huang, J., and Song, H. (2014). Structural basis of SOSS1 complex assembly and recognition of ssDNA. *Cell Rep* *6*, 982-991.

Sabath, K., Staubli, M.L., Marti, S., Leitner, A., Moes, M., and Jonas, S. (2020). INTS10-INTS13-INTS14 form a functional module of Integrator that binds nucleic acids and the cleavage module. *Nat Commun* *11*, 3422.

Shin, Y., Berry, J., Pannucci, N., Haataja, M.P., Toettcher, J.E., and Brangwynne, C.P. (2017). Spatiotemporal Control of Intracellular Phase Transitions Using Light-Activated optoDroplets. *Cell* *168*, 159-171 e114.

Smolka, J.A., Sanz, L.A., Hartono, S.R., and Chedin, F. (2021). Recognition of RNA by the S9.6 antibody creates pervasive artifacts when imaging RNA:DNA hybrids. *J Cell Biol* *220*.

Stein, C.B., Field, A.R., Mimoso, C.A., Zhao, C., Huang, K.L., Wagner, E.J., and Adelman, K. (2022). Integrator endonuclease drives promoter-proximal termination at all RNA polymerase II-transcribed loci. *Mol Cell* *82*, 4232-4245 e4211.

Taslimi, A., Vrana, J.D., Chen, D., Borinskaya, S., Mayer, B.J., Kennedy, M.J., and Tucker, C.L. (2014). An optimized optogenetic clustering tool for probing protein interaction and function. *Nat Commun* *5*, 4925.

Touma, C., Adams, M.N., Ashton, N.W., Mizzi, M., El-Kamand, S., Richard, D.J., Cubeddu, L., and Gamsjaeger, R. (2017). A data-driven structural model of hSSB1 (NABP2/OBFC2B) self-oligomerization. *Nucleic Acids Res* *45*, 8609-8620.

Wang, Z., Song, A., Xu, H., Hu, S., Tao, B., Peng, L., Wang, J., Li, J., Yu, J., Wang, L., *et al.* (2022). Coordinated regulation of RNA polymerase II pausing and elongation progression by PAF1. *Sci Adv* *8*, eabm5504.

Zheng, H., Qi, Y., Hu, S., Cao, X., Xu, C., Yin, Z., Chen, X., Li, Y., Liu, W., Li, J., *et al.* (2020). Identification of Integrator-PP2A complex (INTAC), an RNA polymerase II phosphatase. *Science* *370*, eabb5872.

Review Figure 1

Review Fig. 1. (a-b) V5 Co-IP in cells stably expressing V5-tagged SSB1 (a) and SSB2 (b), followed by western blotting of INTAC subunits and RNase H1. (c) Sequence comparison of the C-termini of bacterial SSB and human SSB1/2. (d) Illustration of the bacterial SSB–RNase HI structure. (e) Sequence comparison of bacterial RNase HI and human RNase H1. The evolutionarily conserved amino acid residues of bacterial RNase HI potentially important for its interaction with SSB are highlighted in red boxes. (f) V5 Co-IP and western blotting in cells stably expressing V5-tagged human SSB1, bacterial SSB, or the chimeric construct (replacing the C-terminal 9 residues of human SSB1 with that of bacterial SSB) in addition to Flag-tagged human RNase H1.

Review Figure 2

Review Fig. 2. (a-b) Heatmaps (a) and metaplots (b) showing Pol II ChIP-Rx signals over 6 kb regions centered on TSS of SOSS-INTAC target genes ranked by decreasing FC occupancy in cells with DOX-inducible RNase H1 expression.

Review Figure 3

Review Fig. 3. (a) Heatmaps showing R-loop CUT&Tag signals centered on TSS in cells treated with flavopiridol (FP) or triptolide (TPL). (b) MA plot of differentially expressed genes by comparing DMSO- and DOX-treated cells with inducible expression of RNase H1.

Review Figure 4

Review Fig. 4. (a-c) Metagenome analysis of Pol II occupancy in CTR and DKO cells for mono-exonic vs. multi-exonic genes (a), quartiles of genes classified by gene length (b), and protein-coding vs. non-coding genes (c). (d) Boxplots quantifying the pausing index in CTR and DKO cells for all four quartiles of genes classified by pausing index in CTR cells.

Review Figure 5

Review Fig. 5. (a-b) Heatmaps showing Pol II ChIP-Rx signals over 20 kb (a) and 40 kb (b) regions centered on TSS of SOSS-INTAC target genes in sgCtr and INTS2-KO cells.

Review Figure 6

Review Fig. 6. (a) Establishing a stable cell line expressing GFP-SSB1 to a level comparable to endogenous SSB1. (b) Estimation of average protein concentration of SSB1 foci in cells using purified EGFP-SSB1 to calibrate the image intensities.

Review Figure 7

a

SSB1-SSB1 interaction interface

b

c

Review Fig. 7. (a) Structural illustration of SSB1-SSB1 interaction interface. (b-c) Wild-type SSB1 and mutant SSB1 protein defective in tetramerization were analyzed using droplet formation assays (b). Quantification of the size of droplets are shown in (c). Red lines indicate the mean in each population.

Reviewer Reports on the First Revision:

Referees' comments:

Referee #1 (Remarks to the Author):

In their revised manuscript, Xu and colleagues addressed all of the suggested comments made, as well as those of other reviewers. They added a considerable amount of new data which strengthened their results. The work provided during the revisions period is impressive and of great standard. As requested, we now agree on the usage of the SOSS term instead of SSB1 as suggested for the figures Fig. 1f and g, since data has been brought forward to support the fact that the SOSS complex is involved. We however suggest several additional changes:

- 1) all novel imaging experiments integrated on SOSS-INTAC nuclear puncta in vivo require quantification with adapted statistics.
- 2) after looking at the materials and methods, we could not find the reference list of antibodies used in this manuscript. Please adjust accordingly.
- 3) Please incorporate the method for the analysis of the pausing index in the materials and methods.
- 4) We could not find the information regarding the plasmid used for RNase H1 overexpression nor the manner it was experimentally used. Please adjust the materials and method/tables accordingly. On this note, perhaps a section in the materials and methods about rescue experiments and the origins of the plasmids also needs to be added.
- 5) In the immunofluorescence section of the Material and methods, the references of the RNase T1 and III need to be added.
- 6) The nature of the break could also be discussed somewhere in the text as it was somewhat surprising to observe such a narrow γ H2AX signal at promoters (here probably spanning 1kb), while at DSB, γ H2AX is rather known to spread on Megabase domains which correspond to the entire damaged TAD, and which are thought to correspond to "foci" visible by microscopy. So here there is a slight discrepancy between the images shown and the γ H2AX Cut&Tag data that therefore deserve some comments.

Once these changes are made and if they still support the experimental data (for comment #1), we believe that the revised manuscript is fitting Nature publication standards and will be of interest to a broad readership in the converging fields of transcription, R-loop and genome stability.

Referee #2 (Remarks to the Author):

Authors have performed a number of experiments that I believe adequately address all the key points raised by all the reviews. The revised version has greatly improved. I have some minor but important points that needs to be addressed in the final version that are summarized below:

1. Manuscript lacks information on the antibodies used (antibodies used for ChIP, IF, pull downs and WB for Integrator subunits, exosome, Xrn2, SSB proteins). This is important to include to ensure that the reported effects can be recapitulated by other researchers. Currently antibody source is not indicated either in Suppl table or Methods section. How shRNA KD and dTAG construction and protein depletion experiments were performed? Authors should clearly indicate what cell line was used for each experiment. The information should be provided to ensure that other labs could reproduce the protocols including the company and the name of the product.
2. Data shown in Reviewer fig 4 should be included in the manuscript and discussed. It is important to distinguish effect on promoter proximal pausing seen at multi-exonic genes from

premature transcription termination (mono-exonic genes) as Integrator recruitment is linked to both processes. I suggest that authors further determine whether genes that show Xrn2 recruitment within gene body are those that show increase in Pol II gene body occupancy in DKO? Is integrator cleavage required for termination or pausing (are mono-exonic genes that are affected in cleavage defective INTS11)?

3. Page 35- replace...‘with S9.6 to cleavage’.. for ‘with S9.6 to perform cleavage’...

Referee #3 (Remarks to the Author):

I appreciate the authors taking time to do additional controls and many re-requested experiments to improve this ms. However, I still have some outstanding comments listed below, which are important to support the validity of proposed mechanism.

In relation to major comment 1:

The authors have not accessed if they observe changes in the Pol II pausing index using condensation mutants, as requested in my previous Major Comment 1. The current experiment in Supplem Fig 21a just shows the level of Pol II at prom regions which does not discriminate between recruitment and pausing of Pol II in this region. The experiment calculating the ratio between proximally paused and active Pol II in the gene body (i.e. pausing index) is required to support the claims that condensation function of SSB1 is important for the regulation of promoter-proximal pausing, as claimed by the authors in this ms. Such analysis needs to be performed properly with the condensation mutants especially considering that authors observe 4 different groups of genes based on their ‘promoter-proximal pausing’ response to SOSS-INTAC depletion (as discussed in the major Point 2).

In relation to comment 3:

The authors need to add statistical significance for their new data and R-loop profiles (genome browser and pPCR validation) for their negative control gene HBB1 (which is also used for SSB1 and INTS3 CHIP analysis). This will support the specificity of the observed effects.

Referee #4 (Remarks to the Author):

The authors have addressed most of my concerns. I appreciate the addition of more data to support R-loop formation and in cell phase separation. Two minor comments:

1. The R-loop IF data still don't look very convincing to me: They do not look like the pattern of SSB1/INTS4 foci or rH2AX foci, so I don't think they can be used to support the effect of SSB1/INTS4 mediated R-loop formation for genome stability control. The added R-loop CUT&Tag and γ H2AX CUT&Tag data are convincing. I would recommend tuning down on the dependence of the IF data and mentioning the discrepancy/possible artifacts when interpreting the IF data.
2. What is the control in c in the optodroplet assay? Please directly state it on the figure or in the legend. Since the paper focuses on the importance of IDR, I think the effect of IDR deletion and mutants in this optodroplet assay, in parallel to what's shown the in vitro assay, would really strengthen the role of IDR in affecting SSB1 function through phase separation.

Author Rebuttals to First Revision:

Referee #1 (Remarks to the Author):

In their revised manuscript, Xu and colleagues addressed all of the suggested comments made, as well as those of other reviewers. They added a considerable amount of new data which strengthened their results. The work provided during the revisions period is impressive and of great standard. As requested, we now agree on the usage of the SOSS term instead of SSB1 as suggested for the figures Fig. 1f and g, since data has been brought forward to support the fact that the SOSS complex is involved. We however suggest several additional changes:

We deeply appreciate the reviewer for the supportive comments and we have implemented the additional suggested changes in the new revised manuscript.

1) all novel imaging experiments integrated on SOSS-INTAC nuclear puncta in vivo require quantification with adapted statistics.

We thank the reviewer for this important point. As suggested, we have performed quantifications for imaging experiments as described below:

(1) Nuclear puncta of SSB1 and INTAC subunits in wild-type and DKO cells. These results verify that SSB1/2 is crucial for the puncta formation of INTAC subunits (Extended Data Fig. 16a).

(2) Nuclear puncta of SSB1 and INTS11 in INTS11-dTAG cells treated with DMSO or dTAG. These results confirm that the depletion of INTS11 exerts no noticeable impact on the formation of SSB1 puncta (Extended Data Fig. 16b).

(3) Nuclear puncta of the stably expressed wild-type and mutant SSB1 in SSB1-dTAG cells. These data corroborate that, compared with wild-type SSB1, the puncta formation capacity of Δ IDR and RY is abolished and that of SI and R206Q is severely impaired (Extended Data Fig. 20b, c).

2) after looking at the materials and methods, we could not find the reference list of antibodies used in this manuscript. Please adjust accordingly.

We apologize for the lack of information regarding materials. In the revised manuscript, we have added a table (Supplementary Table 1) that includes detailed information of materials, including antibodies, used in this study.

3) Please incorporate the method for the analysis of the pausing index in the materials and methods.

As suggested, we have now incorporated the method of analysis of the pausing index in the “Methods” section.

4) We could not find the information regarding the plasmid used for RNase H1 overexpression nor the manner it was experimentally used. Please adjust the materials and method/tables accordingly. On this note, perhaps a section in the materials and methods about rescue experiments and the origins of the plasmids also needs to be added.

As suggested, we have added sequence information of plasmids used in this manuscript in Supplementary Table 2. Moreover, we now describe the RNase H1 overexpression and rescue experiments in detail in the “Methods” section.

5) In the immunofluorescence section of the Material and methods, the references of the RNase T1 and III need to be added.

The references for the RNase T1 and III are now included in Supplementary Table 1.

6) The nature of the break could also be discussed somewhere in the text as it was somewhat surprising to observe such a narrow γ H2AX signal at promoters (here probably spanning 1kb), while at DSB, γ H2AX is rather known to spread on Megabase domains which correspond to the entire damaged TAD, and which are thought to correspond to “foci” visible by microscopy. So here there is a slight discrepancy between the images shown and the γ H2AX Cut&Tag data that therefore deserve some comments.

We thank the reviewer for raising this interesting point. Previous studies have shown both narrow peaks^{1,2} and megabase domains^{3,4} for induced γ H2AX signals. Importantly, the distribution of endogenous γ H2AX we observed here is consistent with published results demonstrating the enrichment of γ H2AX around TSS of actively transcribed genes in cells without external damage⁵. One plausible explanation is that the differences in profiles of induced γ H2AX signals may reflect the varying patterns of DSB occurrence. Supporting this hypothesis, estrogen treatment induced focused enrichment of γ H2AX at ER-bound regions, while H₂O₂ treatment led to a more dispersed elevation of γ H2AX signals².

Figure 4K-L from Periyasamy et al.²

Once these changes are made and if they still support the experimental data (for comment #1), we believe that the revised manuscript is fitting Nature publication standards and will be of interest to a broad readership in the converging fields of transcription, R-loop and genome stability.

We thank the reviewer again for the positive comments. As elaborated in our response to comment #1, the newly added quantification data and the associated statistics for *in vivo* nuclear puncta assays fully support our original conclusions.

Referee #2 (Remarks to the Author):

Authors have performed a number of experiments that I believe adequately address all the key points raised by all the reviews. The revised version has greatly improved. I have some minor but important points that needs to be addressed in the final version that are summarized below:

We appreciate the positive comments and we have addressed all additional points raised here as described below.

1. Manuscript lacks information on the antibodies used (antibodies used for ChIP, IF, pull downs and WB for Integrator subunits, exosome, Xrn2, SSB proteins). This is important to include to ensure that the reported effects can be recapitulated by other researchers. Currently antibody source is not indicated either in Suppl table or Methods section. How shRNA KD and dTAG construction and protein depletion experiments were performed? Authors should clearly indicate what cell line was used for each experiment. The information should be provided to ensure that other labs could reproduce the protocols including the company and the name of the product.

We agree that it is crucial to provide detailed information of materials and methods for the sake of reproducibility and apologize for the lack of detail and clarity in the previous version of the manuscript. In this revised manuscript, we have added a table (Supplementary Table 1) that includes detailed information of materials, including all antibodies used in the study. The DNA sequences for shRNA, primers, and cDNAs, used in this study are now included in the Supplementary Table 2. Experimental details of RNAi, dTAG cell generation, induced protein degradation and rescue assays are included in the “Methods” section. Moreover, we have now specified which cell line was used for each experiment in either the “Results” section or in the appropriate figure legends.

2. Data shown in Reviewer fig 4 should be included in the manuscript and discussed. It is important to distinguish effect on promoter proximal pausing seen at multi-exonic genes from premature transcription termination (mono-exonic genes) as Integrator recruitment is linked to both processes. I suggest that authors further determine whether genes that show Xrn2 recruitment within gene body are those that show increase in Pol II gene body occupancy in DKO? Is integrator cleavage required for termination or pausing (are mono-exonic genes that are affected in cleavage defective INTS11)?

Following the reviewer’s suggestion, we now include and discuss the original Reviewer Fig. 4 in the revised manuscript (Extended Data Fig. 6d-f). Moreover, we analyzed the occupancy of XRN2 as suggested. Plotting XRN2 occupancy on genes of different classes indicated highly similar patterns for XRN2 and Pol II in the control condition, as was previously reported (Extended Data Fig. 6d-f, Extended Data Fig. 13e-g)⁶. Importantly, we found that mono-exonic, noncoding and shorter genes, which showed an increase in gene-body Pol II occupancy (Extended Data Fig. 6d-f), exhibited a decline in XRN2 occupancy at corresponding regions in DKO cells (Extended Data Fig. 13e-g). This implies that the coordination of

SOSS-INTAC endonuclease and XRN2 exonuclease is not limited at promoters and may function at transcription elongation for a subset of genes.

As requested, we next examined whether the RNA endonuclease activity of SOSS-INTAC is required for its regulation of both mono-exonic and multi-exonic genes. As shown by ChIP-qPCR, specific disruption of the endonuclease activity by expressing catalytic-dead, but not wild-type INTS11, in INTS11-depleted cells elicited comparable alterations of Pol II occupancy for both shorter and longer genes (Extended Data Fig. 13h, i) as seen in DKO cells (Extended Data Fig. 6f, Extended Data Fig. 21a). These results corroborate that the INTAC endonuclease functions at both mono-exonic and multi-exonic genes.

3. Page 35- replace... 'with S9.6 to cleavage'.. for 'with S9.6 to perform cleavage'...

We apologize for this error and have corrected it as suggested.

Referee #3 (Remarks to the Author):

I appreciate the authors taking time to do additional controls and many re-requested experiments to improve this ms. However, I still have some outstanding comments listed below, which are important to support the validity of proposed mechanism.

We appreciate this reviewer for positively commenting on our improvements to the manuscript and the additional comments that have helped us further clarify SOSS-INTAC functions in pausing.

In relation to major comment 1:

The authors have not accessed if they observe changes in the Pol II pausing index using condensation mutants, as requested in my previous Major Comment 1. The current experiment in Supplem Fig 21a just shows the level of Pol II at prom regions which does not discriminate between recruitment and pausing of Pol II in this region. The experiment calculating the ratio between proximally paused and active Pol II in the gene body (i.e. pausing index) is required to support the claims that condensation function of SSB1 is important for the regulation of promoter-proximal pausing, as claimed by the authors in this ms. Such analysis needs to be performed properly with the condensation mutants especially considering that authors observe 4 different groups of genes based on their 'promoter-proximal pausing' response to SOSS-INTAC depletion (as discussed in the major Point 2).

Following the reviewer's suggestion, we have now quantified the Pol II occupancy at both promoters and gene bodies and calculated the pausing index for genes of different classes. Specifically, we focused on both shorter and longer genes. For both groups of genes, Pol II occupancy increases in response to SSB1 depletion were rescued by the expression of wild-type SSB1, HY, and S172P/H173L, but not by that of Δ IDR, SI, RY, or R206Q that harbor impaired condensation capacity (Extended Data Fig. 21a). A greater increase in pausing index, the ratio of Pol II occupancy at promoters to gene bodies, was observed on longer genes in dTAG treated cells, which was reversed by the ectopic expression of wild-type SSB1, HY, and S172P/H173L, but not Δ IDR, SI, RY, or R206Q (Extended Data Fig. 21b).

In relation to comment 3:

The authors need to add statistical significance for their new data and R-loop profiles (genome browser and pPCR validation) for their negative control gene HBB1 (which is also used for SSB1 and INTS3 CHiP analysis). This will support the specificity of the observed effects.

We thank the reviewer for this important comment. We have quantified the newly added R-loop CUT&tag data and calculated the statistical significance as suggested (Extended Data Fig. 12a, Extended Data Fig. 19f). Moreover, we now show genome browser examples of genes, include the negative control gene HBB1, as well as qPCR validation for the R-loop CUT&tag data (Extended Data Fig. 19b, c, Extended Data Fig. 19g). Indeed, these results support the specificity of the observed effects.

Referee #4 (Remarks to the Author):

The authors have addressed most of my concerns. I appreciate the addition of more data to support R-loop formation and in cell phase separation. Two minor comments:

We thank the reviewer for encouraging us to examine SSB1 mutants using the optoDroplet assay in addition to conducting a more thorough analysis of the R-loop IF results that have greatly improved the manuscript. Below, we describe how we have addressed the remaining two comments in the revised manuscript.

1. The R-loop IF data still don't look very convincing to me: They do not look like the pattern of SSB1/INTS4 foci or rH2AX foci, so I don't think they can be used to support the effect of SSB1/INTS4 mediated R-loop formation for genome stability control. The added R-loop CUT&Tag and γ H2AX CUT&Tag data are convincing. I would recommend tuning down on the dependence of the IF data and mentioning the discrepancy/possible artifacts when interpreting the IF data.

We agree with the reviewer that the CUT&Tag data are more rigorous and have better resolution compared with the IF data. Therefore, we have tuned down the importance of the IF data and discussed the discrepancy/possible artifacts of the IF data as suggested, as indicated in the excerpt below:

Results:

"However, we noticed that the signals of R-loop foci measured by GFP-dRNH1 in DKO cells are more dispersed compared with the observed pattern for SOSS-INTAC puncta, which might be owing to extended impacts on R-loop accumulation resulting from SSB1 loss or merely reflect limitations of GFP-dRNH1-based R-loop visualization. Accordingly, we mainly relied on R-loop CUT&Tag for determinations of the amount and distribution of R-loops in this study"

2. What is the control in c in the optodroplet assay? Please directly state it on the figure or in the legend. Since the paper focuses on the importance of IDR, I think the effect of IDR deletion and mutants in this optodroplet assay, in parallel to what's shown the in vitro assay, would really strengthen the role of IDR in affecting SSB1 function through phase separation.

We apologize for the lack of clarify in our prior description. The control is the empty mCherry-CRY2 vector. We now label the figure as suggested.

In addition, we have examined whether SSB1 mutation affects its liquid-like condensation capacity in cells using the optoDroplet assay. We focused on the cancer-derived mutations of SSB1. Interestingly, our results indicate that S172P/H173L forms condensates as readily as wild-type SSB1 upon light induction, whereas

R206Q, like Δ IDR, exhibits a severely impaired capacity in condensation (Extended Data Fig. 20d). These results obtained by the optoDroplet assay are consistent with that of the puncta formation assays *in vivo* and *in vitro*.

References:

- 1 Kantidakis, T. *et al.* Mutation of cancer driver MLL2 results in transcription stress and genome instability. *Genes Dev* **30**, 408-420, doi:10.1101/gad.275453.115 (2016).
- 2 Periyasamy, M. *et al.* APOBEC3B-Mediated Cytidine Deamination Is Required for Estrogen Receptor Action in Breast Cancer. *Cell Rep* **13**, 108-121, doi:10.1016/j.celrep.2015.08.066 (2015).
- 3 Promonet, A. *et al.* Topoisomerase 1 prevents replication stress at R-loop-enriched transcription termination sites. *Nat Commun* **11**, 3940, doi:10.1038/s41467-020-17858-2 (2020).
- 4 Lyu, X., Chastain, M. & Chai, W. Genome-wide mapping and profiling of gammaH2AX binding hotspots in response to different replication stress inducers. *BMC Genomics* **20**, 579, doi:10.1186/s12864-019-5934-4 (2019).
- 5 Seo, J. *et al.* Genome-wide profiles of H2AX and gamma-H2AX differentiate endogenous and exogenous DNA damage hotspots in human cells. *Nucleic Acids Res* **40**, 5965-5974, doi:10.1093/nar/gks287 (2012).
- 6 Cortazar, M. A. *et al.* Xrn2 substrate mapping identifies torpedo loading sites and extensive premature termination of RNA pol II transcription. *Genes Dev* **36**, 1062-1078, doi:10.1101/gad.350004.122 (2022).

Reviewer Reports on the Second Revision:

Referees' comments:

Referee #1 (Remarks to the Author):

The authors have fully responded to our previous comments, as well as to other's reviewer comments. We believe the manuscript is now suitable for publication

Referee #2 (Remarks to the Author):

Most comments are addressed, however information on Methods and reagents is still incomplete and should be completed before publication. How ChIP experiments or Xrn2, Integrator and exosome subunits was performed? Did authors use antibodies for Chromatin immunoprecipitation in each of these experiments or used tagged proteins? What antibodies were used in CHIP in each case? Information on antibody for all ChIP experiments should be included together with the conditions for IP/washes and ab manufacturer in Methods and Supplementary table 1. Currently Suppl table one lists only mentions ab for Western blotting.

Referee #3 (Remarks to the Author):

The authors have adequately addressed my comments.

Author Rebuttals to Second Revision:

Referee #1 (Remarks to the Author):

5 *The authors have fully responded to our previous comments, as well as to other's reviewer comments. We believe the manuscript is now suitable for publication.*

We deeply appreciate the reviewer for considering that our manuscript is now suitable for publication.

10

Referee #2 (Remarks to the Author):

15 *Most comments are addressed , however information on Methods and reagents is still incomplete and should be completed before publication. How ChIP experiments or Xrn2, Integrator and exosome subunits was performed? Did authors use antibodies for Chromatin immunoprecipitation in each of these experiments or used tagged proteins? What antibodies were used in CHIP in each case? Information on antibody for all ChIP experiments should be included together with the conditions for IP/washes and ab manufacturer in Methods and Supplementary table 1. Currently Suppl table one lists only mentions ab for Western blotting.*

20

25 Following the reviewer's suggestion, we have now provided more detailed information on Methods and reagents. All ChIP-seq experiments were performed using antibodies against proteins of interest instead of antibodies recognizing protein tags. The detailed information of antibodies used for ChIP-seq and western blotting is included in Supplementary Table 1. The detailed procedures of ChIP experiments are described in Methods section.

25

Referee #3 (Remarks to the Author):

30

The authors have adequately addressed my comments.

We deeply appreciate the reviewer for believing that we have addressed all suggested comments.